# Random Projections for Spectral Algorithms in Mis-specified Setting: Sobolev Norm Learning Rates and Minimax Optimality

## Abstract

Random projections (RP) offer an effective approach to reducing computational and storage costs while preserving the geometric structure of the data. However, existing studies primarily focus on the optimal generalization performance of specific kernel-regularized algorithms with RP in the well-specified setting under restrictive conditions. In this paper, we provide a comprehensive and improved analysis of the generalization performance of RP-based spectral algorithms under general conditions, without increasing computational complexity. By leveraging the embedding property of the RKHS and a refined analysis of the operator similarity, we establish optimal learning rates in Sobolev norms that match the minimax lower bounds up to logarithmic factors. For both randomized sketches and Nyström sub-sampling (uniform or leverage-based), we show that the projection dimension needed for optimality is proportional to the average or maximal effective dimension, yielding a significant reduction in computational cost while maintaining the statistical efficiency. Our results do not rely on the uniform boundedness assumption on the target function and hold for a broad range of source conditions, i.e., $s \geq \alpha - 1/\beta$, where $s, \beta$, and $\alpha$ denote the smoothness index, capacity index, and the embedding index, respectively. In the benign case when $\alpha = 1/\beta$, the optimality holds for all $s \in (0, 2\tau]$ with $\tau$ denoting the quantification index. Experimental results confirm our theoretical findings and demonstrate the practical effectiveness of RP.

## 1 Introduction

In statistical learning theory, the primary objective is to learn a function that approximates the underlying relationship between input and output variables based on a finite set of training data (Vapnik, 1999b;a). Kernel methods, a class of nonparametric learning algorithms, have gained significant attention due to their ability to capture complex data structures and their strong generalization performance (Wahba, 1990; Schölkopf & Smola, 2002). These methods operate in a reproducing kernel Hilbert space (RKHS) associated with a positive definite kernel function $K(\cdot, \cdot)$. To address the ill-posedness of such learning problems, regularization techniques are commonly employed, leading to the development of various kernel-regularized algorithms, or spectral algorithms (Engl et al., 1996). Popular examples include Tikhonov regularization, also known as kernel ridge regression (KRR) (Smale & Zhou, 2007; Caponnetto & De Vito, 2007), spectral cut-off (Blanchard et al., 2007; Dicker et al., 2017), gradient methods (Yao et al., 2007; Lin & Rosasco, 2016). Despite their solid theoretical guarantees, these algorithms often suffer substantial computational costs for large-scale datasets. For instance, KRR typically requires $O(n^3)$ time and $O(n^2)$ space complexity, where $n$ is the number of training samples. To mitigate this issue, various accelerating techniques have been developed, including distributed algorithms (Zhang et al., 2015; Guo et al., 2017), random features (Rahimi & Recht, 2007; Rudi & Rosasco, 2017), randomized sketches (Yang et al., 2017; Zhang & Liao, 2019), and Nyström sub-sampling (Williams & Seeger, 2001; Kumar et al., 2012).

Random projections (RP), including randomized sketches and Nyström sub-sampling as special cases, have emerged as powerful techniques for dimensionality reduction that enable efficient computation, while preserving the geometric structure of data (Vempala, 2005). The core idea of RP is to estimate the regression function within a suitably chosen random subspace of the RKHS, thereby yielding spectral algorithms with RP. Specifically, we consider a projection dimension $m \ll n$ and construct

a random subspace $\mathcal{F} = \overline{\text{span}\{K_{\boldsymbol{x}_{i_j}}\}_{j=1}^m}$, where $\{i_1, \ldots, i_m\} \subset \{1, \ldots, n\}$ are randomly selected indices from the input set $\{\boldsymbol{x}_i\}_{i=1}^n$. Performing the spectral algorithm within this subspace $\mathcal{F}$ yields the Nyström sub-sampling method. More generally, we may consider a random matrix $\mathbf{G} \in \mathbb{R}^{m \times n}$ and define the random subspace as $\mathcal{F} = \overline{\text{span}\{\sum_{j=1}^n \mathbf{G}_{ij} K_{\boldsymbol{x}_j}, i = 1, \ldots, m\}}$, which leads to the randomized sketching method. By choosing $m \ll n$, we can typically achieve substantial reductions in the time and space complexity of spectral algorithms while maintaining their statistical efficiency. (Williams & Seeger, 2001; Kumar et al., 2012; Mahoney et al., 2011).

Theoretical analysis of the generalization properties for kernel methods with specific random projection methods has been extensively studied in the literature. In the fixed design, where the inputs $\{\boldsymbol{x}_i\}_{i=1}^n$ are deterministic, Yang et al. (2017) studied KRR with randomized sketches and established optimal learning rates in the worst case where the target function $f_\rho$ lies in the RKHS. Similarly, sharp generalization error bounds in expectation for the Nyström sub-sampling method were derived in Bach (2013); Alaoui & Mahoney (2015). In the random design setting, where the inputs $\{\boldsymbol{x}_i\}_{i=1}^n$ are drawn i.i.d. from an unknown distribution, Rudi et al. (2015); Myleiko et al. (2019) focused on KRR and general spectral algorithms with Nyström sub-sampling, and established source and capacity dependent learning rates in high probability. However, these results are limited to the well-specified case where $f_\rho$ lies in the RKHS ($s \in [1, 2\tau]$ with $s$ denoting the smoothness index and $\tau$ denoting the quantification index), and the generalization error is measured via the $L_\rho^2$ norm. The most closely related work is Lin & Cevher (2020a), which studied the spectral algorithms with random projections in the mis-specified setting where $f_\rho$ may not lie in the RKHS. They derived optimal learning rates in various norms when $s \geq 1 - 1/\beta$ ($\beta \in [1, \infty)$ with $\beta$ denoting the capacity index), thus covering part of the mis-specified setting ($s \in (0, 1)$). However, their analysis relies on restrictive assumptions, such as the Hölder-type source condition and $f_\rho$ is uniformly bounded, which may not hold in practice. Till now, whether optimal learning rates of the spectral algorithms with random projections can be achieved in the mis-specified setting when $s < 1 - 1/\beta$ remains an open question.

In this paper, we provide a comprehensive and improved analysis of the generalization performance for the spectral algorithms with random projections (SARP) in the mis-specified setting under general conditions, without increasing computational complexity. The following are the main contributions of this paper:

**General conditions.** Our analysis relies on five key assumptions: (i) Bernstein condition on the noise and does not require the uniformly boundedness assumption on $f_\rho$ (Assumption 1); (ii) capacity condition (Assumptions 2 and 5) and embedding condition (Assumption 3); (iii) the regression function $f_\rho$ follows a general source condition (Assumption 4), where the Hölder source condition is a special case. Compared to those in the literature of spectral algorithms with RP (Yang et al., 2017; Rudi et al., 2015; Lin & Cevher, 2020a), our assumptions are more general or easier to satisfy in practice.

**Comprehensive and improved learning rates.** Under these more general assumptions, we first establish the minimax lower bounds in Sobolev norms for all $s \in (0, 2\tau]$ in Theorem 1. For the upper bounds, we combine the integral-operator technique with the embedding property of the RKHS to derive sharp learning rates for general SARP in Theorem 2. By choosing the regularization parameter $\lambda$ appropriately, we show that the upper bound is minimax optimal once the projection error term is of smaller order than the typical approximation and empirical error terms. We further prove this is achievable for randomized sketches and Nyström sub-sampling in Corollary 2, when the projection dimension $m$ is proportional to the average or maximal effective dimension (up to logarithmic factors). Compared to the state-of-the-art results in Lin & Cevher (2020a), we do not require the uniformly boundedness of $f_\rho$, and the theory holds for a general and broader range of source conditions, i.e., $s \geq \alpha - 1/\beta$. In the benign case when $\alpha = 1/\beta$, the optimality holds for all $s \in (0, 2\tau]$. Note that our bounds do not improve the minimax exponents compared to the best existing results in overlapping regimes, but significantly broaden the scope and practicality of minimax-optimal results for SARP.

**Efficient computational improvement.** From a computational perspective, our results show that the projection dimension $m$ needed to achieve the optimal learning rates is no larger (and can even be smaller in some cases) than those required in Yang et al. (2017); Rudi et al. (2015); Lin & Cevher (2020a). This leads to a substantial reduction in computational cost while preserving statistical

efficiency in a broader source range. The improvement stems from our more refined analysis of operator similarity and sample variance control. See Remark 3 for more details.

For a better understanding of our contribution and novelty, we present a detailed comparison with existing methods in Appendix B. We also report numerical experiments that validate our theoretical findings and illustrate the practical effectiveness of SARP in Appendix F.

## 2 PRELIMINARIES

Let $(\mathcal{X}, \mathcal{B})$ be a Borel measure space as the input space, $\mathcal{Y} \subset \mathbb{R}$ be the output space. Let $\rho$ be the joint distribution on $\mathcal{X} \times \mathcal{Y}$, which can be decomposed as $\rho(\boldsymbol{x}, y) = \rho_{\mathcal{X}}(\boldsymbol{x})\rho(y|\boldsymbol{x})$, here $\rho(y|\boldsymbol{x})$ be the conditional distribution of $y$ given $\boldsymbol{x}$. Consider a standard supervised learning framework, we observe a set of training samples $\mathbf{z} = \{(\boldsymbol{x}_i, y_i)\}_{i=1}^n$ drawn i.i.d. from the unknown distribution $\rho$. The objective is to find a function $f$ that minimizes the expected risk over the space of all the measurable functions, which is defined as

$$\mathcal{E}(f) = \int_{\mathcal{X} \times \mathcal{Y}} \left(f(\boldsymbol{x}) - y\right)^2 d\rho(\boldsymbol{x}, y). \tag{1}$$

The minimizer of the expected risk is the regression function $f_\rho(\boldsymbol{x}) = \int_{\mathcal{Y}} y d\rho(y|\boldsymbol{x})$. Consequently, we can define the following nonparametric regression model:

$$y = f_\rho(x) + \varepsilon, \tag{2}$$

where $\varepsilon$ is the noise satisfying that $\mathbb{E}(\varepsilon|\boldsymbol{x}) = 0$.

**Assumption 1.** *There exist $\sigma \geq 0$ and $M \geq 0$ such that for all $p \geq 2$, $\mathbb{E}(|\varepsilon|^p|\boldsymbol{x}) \leq \frac{1}{2}p!\sigma^2 M^{p-2}$.*

This assumption is the classical Bernstein condition on the noise, which is widely used in kernel learning theory (Caponnetto & De Vito, 2007; Rastogi & Sampath, 2017). It is satisfied, for example, by bounded, Gaussian, and sub-Gaussian noise. In contrast, the most related work (Lin & Cevher, 2020a) imposes the Bernstein condition directly on the response $y$, which in turn ensures that the regression function $f_\rho$ is uniformly bounded. Such a requirement can be overly restrictive, and a uniform bound on $f\_\rho$ is very hard to verify from finite data in practice. Assumption 1 instead places the Bernstein condition solely on the noise $\varepsilon$ and does not impose uniform boundedness on $f_\rho$. We refer to Appendix C.6 of for a detailed discussion. In our synthetic example in Appendix F, the regression function diverges when $s = 0.5$, so $f_\rho$ is not uniformly bounded, yet SARP still achieves the optimal Sobolev learning rates. This further illustrates that the uniform boundedness assumption is not necessary for deriving the optimal learning rates of SARP.

### 2.1 RKHS AND OPERATORS

Consider a bounded kernel $K : \mathcal{X} \times \mathcal{X} \to \mathbb{R}$, which is symmetric and positive definite, i.e., $K(\boldsymbol{x}, \boldsymbol{x}') = K(\boldsymbol{x}', \boldsymbol{x})$ for $\boldsymbol{x}, \boldsymbol{x}' \in \mathcal{X}$ and $\sum_{i,j=1}^n c_i c_j K(\boldsymbol{x}_i, \boldsymbol{x}_j) \geq 0$ for any $n \in \mathbb{N}$, $\{c_i\}_{i=1}^n \subset \mathbb{R}$, and $\{\boldsymbol{x}_i\}_{i=1}^n \subset \mathcal{X}$. This kernel is associated with a unique reproducing kernel Hilbert space (RKHS) $\mathcal{H}$ equipped with inner product $\langle \cdot, \cdot \rangle_{\mathcal{H}}$ and the norm $\| \cdot \|_{\mathcal{H}}$. The RKHS is the completion of the space of finite linear combinations of the kernel functions $\{K_{\boldsymbol{x}} = K(\boldsymbol{x}, \cdot) : \boldsymbol{x} \in \mathcal{X}\}$ with respect to the inner product $\langle K_{\boldsymbol{x}}, K_{\boldsymbol{x}'} \rangle_{\mathcal{H}} = K(\boldsymbol{x}, \boldsymbol{x}')$. For any $f \in \mathcal{H}_K$, there holds that

$$f(\boldsymbol{x}) = \langle f, K_{\boldsymbol{x}} \rangle_{\mathcal{H}}, \quad \forall \boldsymbol{x} \in \mathcal{X}.$$

Since $K$ is bounded, we have $\sup_{\boldsymbol{x} \in \mathcal{X}} K(\boldsymbol{x}, \boldsymbol{x}) \leq \kappa^2$ for some $\kappa > 0$. This implies that $K$ is square-integrable w.r.t. the measure $\rho_{\mathcal{X}}$. Since this condition can already guarantee the spectral decomposition of $K$, we do not impose the standard compactness of $\mathcal{X}$ and Mercer kernel conditions.

Let $L^2_{\rho_{\mathcal{X}}}$ be the space of square integral functions from $\mathcal{X}$ to $\mathbb{R}$ with respect to $\rho_{\mathcal{X}}$, with its norm given by $\|f\|_\rho = \sqrt{\int_{\mathcal{X}} f^2(\boldsymbol{x}) d\rho_{\mathcal{X}}(\boldsymbol{x})}$. We denote the inclusion operator $S_K : \mathcal{H}_K \to L^2_{\rho_{\mathcal{X}}}$ as $S_K f(\boldsymbol{x}) = \langle f, K_{\boldsymbol{x}} \rangle_{\mathcal{H}}$, for any $f \in \mathcal{H}_K$ and $\boldsymbol{x} \in \mathcal{X}$. Its adjoint operator is denoted by $S_K^* : L^2_{\rho_{\mathcal{X}}} \to \mathcal{H}_K$, $S_K^* g = \int_{\mathcal{X}} K_{\boldsymbol{x}} g(\boldsymbol{x}) d\rho_{\mathcal{X}}(\boldsymbol{x})$, for any $g \in L^2_{\rho_{\mathcal{X}}}$ and $\boldsymbol{x} \in \mathcal{X}$. Then $S_K$ and $S_K^*$ are Hilbert-Schmidt

operators, and thus compact operators. Furthermore, we can define the integral operator $L_K$ and the covariance operator $C_K$ as follows:

$$L_K : L^2_{\rho_\mathcal{X}} \to L^2_{\rho_\mathcal{X}}, \quad L_K g = \int_\mathcal{X} K(\boldsymbol{x}, \cdot) g(\boldsymbol{x}) d\rho_\mathcal{X}(\boldsymbol{x}),$$

$$C_K : \mathcal{H}_K \to \mathcal{H}_K, \quad C_K = \int_\mathcal{X} \langle \cdot, K_{\boldsymbol{x}} \rangle_\mathcal{H} K_{\boldsymbol{x}} d\rho_\mathcal{X}(\boldsymbol{x}) = \int_\mathcal{X} K_{\boldsymbol{x}} \otimes K_{\boldsymbol{x}} d\rho_\mathcal{X}(\boldsymbol{x}). \quad (3)$$

Since $K$ is bounded on $\mathcal{X}$, the operators $L_K$ and $C_K$ are positive, self-adjoint, and trace-class (hence Hilbert–Schmidt and compact) (Steinwart & Christmann, 2008). They satisfy the identities $L_K = S_K S_K^*$ and $C_K = S_K^* S_K$. Moreover, as $\sup_{\boldsymbol{x} \in \mathcal{X}} K(\boldsymbol{x}, \boldsymbol{x}) \leq \kappa^2$, we have $\|S_K\| = \|S_K^*\| \leq \kappa$ and $\|L_K\| = \|C_K\| \leq \kappa^2$. For any $f \in \mathcal{H}_K$, the following isometry property holds (Steinwart & Christmann, 2008):

$$\|f\|_\rho = \|S_K f\|_\rho = \|C_K^{1/2} f\|_\mathcal{H}. \quad (4)$$

In addition, by the polar decomposition of $S_K$, $\|L_K^{-1/2} S_K f\|_\rho \leq \|f\|_\mathcal{H}$.

By the spectral theorem, there exists a sequence of non-negative and non-increasing eigenvalues $\{\mu_i\}_{i \geq 1}$, orthonormal basis $\{e_i\}_{i \geq 1}$ of $L^2_\rho(\mathcal{X})$, and orthonormal basis $\{\mu_i^{1/2} e_i\}_{i \geq 1}$ of $\mathcal{H}_K$ such that

$$L_K = \sum_{i=1}^\infty \mu_i \langle \cdot, e_i \rangle_\rho e_i, \quad C_K = \sum_{i=1}^\infty \mu_i \langle \cdot, \mu_i^{1/2} e_i \rangle_\mathcal{H} \mu_i^{1/2} e_i. \quad (5)$$

Moreover, the kernel $K$ can be decomposed as

$$K(\boldsymbol{x}, \boldsymbol{x}') = \sum_{i=1}^\infty \mu_i e_i(\boldsymbol{x}) e_i(\boldsymbol{x}'), \quad \forall \boldsymbol{x}, \boldsymbol{x}' \in \mathcal{X},$$

where the convergence is uniform and absolute. Note that the eigenvalues $\{\mu_i\}_{i \geq 1}$ and eigenfunctions $\{e_i\}_{i \geq 1}$ depend on the kernel $K$ and the marginal distribution $\rho_\mathcal{X}$.

**Definition 1** (**Effective dimension**). *For $\lambda > 0$, we define the random variable $\mathcal{N}_{\boldsymbol{x}}(\lambda) = \langle K_{\boldsymbol{x}}, (C_K + \lambda I)^{-1} K_{\boldsymbol{x}} \rangle_\mathcal{H}$ with $\boldsymbol{x} \in \mathcal{X}$ drawn from $\rho_\mathcal{X}$, then the average effective dimension and maximal effective dimension of RKHS $\mathcal{H}_K$ are defined as $\mathcal{N}(\lambda) = \mathbb{E}\mathcal{N}_{\boldsymbol{x}}(\lambda)$ and $\mathcal{N}_\infty(\lambda) = \sup_{\boldsymbol{x} \in \mathcal{X}} \mathcal{N}_{\boldsymbol{x}}(\lambda)$.*

The effective dimension characterizes the complexity of the RKHS $\mathcal{H}_K$ and plays a crucial role in learning theory (Caponnetto & De Vito, 2007; Steinwart et al., 2009; Guo et al., 2017). It is easy to see that $\mathcal{N}(\lambda) = \text{Tr}((C_K + \lambda I)^{-1} C_K) = \sum_{i=1}^\infty \frac{\mu_i}{\mu_i + \lambda}$ and $\mathcal{N}(\lambda) \leq \mathcal{N}_\infty(\lambda) \leq \kappa^2 \lambda^{-1}$.

**Assumption 2.** *For $\lambda > 0$, there exists $Q > 0$, $\beta \in [1, \infty]$, such that $\mathcal{N}(\lambda) \leq Q^2 \lambda^{-\frac{1}{\beta}}$.*

This assumption is known as the capacity condition, which always holds when $\beta = 1$ by taking $Q = \text{Tr}(L_K) \leq \kappa^2$. It can also be verified by the decay rate of the eigenvalues $\{\mu_i\}_{i \geq 1}$. For example, if $\mu_i \leq ci^{-\beta}$ for some $c > 0$ and $\beta \geq 1$, then $\mathcal{N}(\lambda) \leq c_\beta \lambda^{-\frac{1}{\beta}}$ for some constant $c_\beta > 0$. More examples and discussions on Assumption 2 can be found in Appendix C.5.

## 2.2 INTERPOLATION SPACE OF RKHS

In this section, we introduce the interpolation space (or power space) of the RKHS $\mathcal{H}_K$. For any $t \in \mathbb{R}$, we define the fractional integral operator $L_K^t$ as $L_K^t = \sum_{i=1}^\infty \mu_i^t \langle \cdot, e_i \rangle_\rho e_i$.

**Definition 2.** *Given the RKHS $\mathcal{H}$ associated with the kernel $K$ and the integral operator $L_K$, for any $t \in \mathbb{R}$, we define the interpolation space $[\mathcal{H}]^t_\rho$ as*

$$[\mathcal{H}]^t_\rho := Ran L_K^{t/2} = \left\{ \sum_{i=1}^\infty a_i \mu_i^{t/2} e_i : \sum_{i=1}^\infty a_i^2 < \infty \right\} \subset L^2_\rho(\mathcal{X}). \quad (6)$$

*Let $f, g \in [\mathcal{H}]^t_\rho$ and $f = \sum_{i=1}^\infty a_i \mu_i^{t/2} e_i, g = \sum_{i=1}^\infty b_i \mu_i^{t/2} e_i$, define the inner product and norm of $[\mathcal{H}]^t_\rho$ as $\langle f, g \rangle_{[\mathcal{H}]^t_\rho} = \langle L_K^{-t/2} f, L_K^{-t/2} g \rangle_\rho = \sum_{i=1}^\infty a_i b_i$, $\|f\|_{[\mathcal{H}]^t_\rho} = \langle f, f \rangle^{1/2}_{[\mathcal{H}]^t_\rho} = (\sum_{i=1}^\infty a_i^2)^{1/2}$.*

The interpolation space $[\mathcal{H}]_\rho^t$ is also a separable Hilbert space with orthonormal basis $\{\mu_i^{t/2} e_i\}_{i \geq 1}$. It is easy to see that $[\mathcal{H}]_\rho^1 = [\mathcal{H}]_\rho = \mathcal{H}$ and $[\mathcal{H}]_\rho^0 = L_\rho^2(\mathcal{X})$. For $0 \leq t_1 \leq t_2$, we have the continuous embedding $[\mathcal{H}]_\rho^{t_2} \hookrightarrow [\mathcal{H}]_\rho^{t_1} \hookrightarrow [\mathcal{H}]_\rho^0$ (Fischer & Steinwart, 2020). The interpolation space $[\mathcal{H}]_\rho^s$ can be used to characterize the regularity of the target function $f_\rho$. For example, if $f_\rho \in [\mathcal{H}]_\rho^t$ for some $t > 1$, then $f_\rho$ is smoother than the functions in the RKHS $\mathcal{H}$, and if $f_\rho \in [\mathcal{H}]_\rho^s$ for some $0 < t < 1$, then $f_\rho$ is less smooth than the functions in $\mathcal{H}$.

For $0 < t < 1$, the interpolation space $[\mathcal{H}]_\rho^t$ can be characterized by the real interpolation method (Tsybakov, 2008; Steinwart & Scovel, 2012; Bitzer & Steinwart, 2025). Specifically, we have

$$[\mathcal{H}]_\rho^t \cong [L_\rho^2(\mathcal{X}), \mathcal{H}]_{t,2}, \tag{7}$$

where $\cong$ denotes the isomorphism between the two spaces and $[\cdot, \cdot]_{s,2}$ is the real interpolation functor of two function spaces (see Appendix C.7 for more details).

**Assumption 3.** *Assume that there exists an order $\alpha \in [1/\beta, 1]$ and a constant $A > 0$, such that the canonical inclusion map $i^\alpha : [\mathcal{H}]_\rho^\alpha \hookrightarrow L_\rho^\infty(\mathcal{X}), i^\alpha(f) = f$, is bounded, i.e.*

$$\|f\|_\infty \leq A \|f\|_{[\mathcal{H}]_\rho^\alpha}, \quad \forall f \in [\mathcal{H}]_\rho^\alpha.$$

*Equivalently, $[\mathcal{H}]_\rho^\alpha$ is continuously embedded into $L_\rho^\infty(\mathcal{X})$ and the operator norm of the embedding satisfies $\|i^\alpha\| \leq A$.*

The embedding property indicates that the interpolation space $[\mathcal{H}]_\rho^\alpha$ can be continuously embedded into $L_\rho^\infty(\mathcal{X})$, with operator norm bounded by $A$. A larger $\alpha$ corresponds to a weaker embedding. Notably, the condition is always satisfied when $\alpha = 1$ since the kernel $K$ is bounded. The case $\alpha = 1/\beta$ corresponds to the benign RKHSs discussed in Appendix C.3.

## 2.3 GENERAL SOURCE CONDITION

In this section, we introduce a general source condition which characterizes the regularity of the target function $f_\rho$ via the index function. For $s \in (0, 2\tau]$, and any operator $A$ and projection operator $P$, we first define the index function set as

$$\mathcal{C}_s = \Big\{ \phi : [0, \kappa^2] \to \mathbb{R}^+ \big| \phi \text{ is non-decreasing}, \phi(0) = 0, \phi(\kappa^2) < \infty,$$

$$\text{and } \frac{\phi^2}{t^s} \text{ is non-decreasing for some } s \in (0, 2\tau] \Big\}.$$

**Assumption 4.** *For $R \geq 0$, assume the smoothness condition can be expressed as*

$$f_\rho \in \Omega_{\phi, s, R} := \big\{ f \in L_{\rho_\mathcal{X}}^2 : f = \phi(L_K) g, \|g\|_\rho \leq R, \phi \in \mathcal{C}_s \big\}, \tag{8}$$

*For $\lambda \in [0, \kappa^2]$, we further assume that there exists $c > 0$ such that $c\lambda^\tau / \phi(\lambda) \leq \inf_{t \in [\lambda, \kappa^2]} t^\tau / \phi(t)$, where $\tau$ is the qualification parameter will be defined in Definition 3.*

A typical example is the Hölder source condition, i.e., $\phi(t) = t^{s/2}$ for some $s \in (0, 2\tau]$ (see proof in Appendix C.4). In this case, $f_\rho \in \Omega_{\phi, s, R}$ implies that $f_\rho \in [\mathcal{H}]_\rho^s$ and $\|f_\rho\|_{[\mathcal{H}]_\rho^s} \leq R$, as also considered in Fischer & Steinwart (2020); Zhang et al. (2023; 2024). The parameter $s$ quantifies the smoothness of $f_\rho$ relative to the RKHS $\mathcal{H}_K$. Specifically, if $s > 1$, then $f_\rho$ is smoother than the functions in $\mathcal{H}_K$ ($f_\rho \in \mathcal{H}_K$), while if $0 < s < 1$, then $f_\rho$ is less smooth than the functions in $\mathcal{H}_K$ ($f_\rho \notin \mathcal{H}_K$). Practically, for example, we use a very smooth kernel (Gaussian or high-smoothness Matérn kernel) while the true regression function has only finite Sobolev regularity. Such "kernel too smooth, truth less smooth" situations are common in practice when a Gaussian kernel is used as a default and still performs well. Our theory will show that this is reasonable since SARP can still achieve the optimal rates under this mis-specified setting. We refer to Appendix C.4 for discussions on this general source condition. In Theorem 3, we further generalize the source condition into three different cases.

## 2.4 MINIMAX LOWER BOUND FOR SOBOLEV NORM LEARNING

In this section, we present a minimax lower bound for the learning rate in Sobolev norms under general source conditions, offering the fundamental learning limits of SARP. In order to establish

the lower bound, we need the following additional assumption which can be viewed as an enhanced version of Assumption 2 (Proof can be found in Appendix C.5). A similar assumption is also conducted in Fischer & Steinwart (2020); Zhang et al. (2024).

**Assumption 5.** *Assume that there exists some positive constant $0 < q_1 \leq q_2$ and $\beta \in [1, \infty)$ that*

$$q_1 k^{-\beta} \leq \mu_k \leq q_2 k^{-\beta}, \quad \forall k \geq 1, \tag{9}$$

*where $\mu_k$ is the $k$-th eigenvalue of the integral operator $L_K$. Under (9), there also holds that $\mathcal{N}(\lambda) \asymp \lambda^{-\frac{1}{\beta}}$.*

**Theorem 1.** *Let the constants $\sigma, M, R, Q$ be fixed, then given $s \in (0, 2\tau]$, $\beta \in [1, \infty)$, let $\mathcal{P} = \mathcal{P}(\phi, s, \beta)$ be the set of all probability distributions $\rho$ such that Assumptions 1, 4, 5 hold, then for any $\gamma \in [0, s \wedge 1]$ and $\delta \in (0, 3/8)$, and all possible estimators $\widehat{f}$ based on the $n$ i.i.d. sample set $\mathcal{D}$, there exists a probability measure $\rho^* \in \mathcal{P}$ such that, with probability at least $1 - \delta$, there holds*

$$\|\widehat{f} - f_{\rho^*}\|_{[\mathcal{H}]_{\rho^*}^\gamma}^2 \geq C\delta\phi^2(\Phi^{-1}(n^{-1}))(\Phi^{-1}(n^{-1}))^{-\gamma}, \tag{10}$$

*where $\Phi(t) = (\phi(t)/\phi(1))^2 t^{1/\beta}$ and $\Phi^{-1}(\cdot)$ denotes the functional inverse, and $C$ is a positive constant independent of $\lambda$, $n$, $\delta$, and given explicitly in the proof.*

Theorem 1 establishes a minimax lower bound for the learning rate in Sobolev norms under general source conditions. Compared to the existing lower bound in the literature, our result is more general by considering a general source condition and a broader range of $s \in (0, 2\tau]$. For example, Caponnetto & De Vito (2007); Rastogi & Sampath (2017) only consider the well-specified case with smoothing index $s \in (1, 2\tau]$, and establish the lower bound for the $L_\rho^2$ norm or $\mathcal{H}$ norm. Recently, Fischer & Steinwart (2020); Zhang et al. (2023; 2024) consider the mis-specified case and establish the lower bound for the Sobolev norm with $s \in (0, 2\tau]$. Our results generalize these existing results by considering a general source condition (see detailed comparison in Remark 4 of the Appendix). When $\phi(t) = t^{s/2}$ for some $s \in (0, 2\tau]$, the lower bound reduces to those in Fischer & Steinwart (2020); Zhang et al. (2023; 2024), as the following corollary shows.

**Corollary 1.** *Under the same conditions of Theorem 1, if $\phi(t) = t^{s/2}$ for some $s \in (0, 2\tau]$, then with probability at least $1 - \delta$, there holds that*

$$\|\widehat{f} - f_{\rho^*}\|_{[\mathcal{H}]_{\rho^*}^\gamma}^2 \geq C\delta n^{-\frac{(s-\gamma)\beta}{s\beta+1}}. \tag{11}$$

## 3 SPECTRAL ALGORITHMS AND RANDOM PROJECTIONS

Spectral algorithms (SA) comprise a family of regularization methods that leverage the spectral operator through filter functions to stabilize learning problems (Engl et al., 1996). Initially developed for ill-posed linear inverse problems (De Vito et al., 2005), they have been subsequently extended to nonparametric regression via the connection between learning theory and inverse problems (Rosasco et al., 2005; De Vito et al., 2006; Guo et al., 2017).

Suppose that we have a training dataset $\{(\boldsymbol{x}_i, y_i)\}_{i=1}^n$ drawn i.i.d. from an unknown distribution $\rho$. We denote by $S_{K,n} : \mathcal{H} \to \mathbb{R}^n$ the sampling operator as

$$S_{K,n} : \mathcal{H}_K \to \mathbb{R}^n, \quad S_{K,n}f = \frac{1}{\sqrt{n}}(\langle f, K_{\boldsymbol{x}_1}\rangle_{\mathcal{H}}, \ldots, \langle f, K_{\boldsymbol{x}_n}\rangle_{\mathcal{H}}),$$

and its adjoint operator is

$$S_{K,n}^* : \mathbb{R}^n \to \mathcal{H}_K, \quad S_{K,n}^*\boldsymbol{\alpha} = \frac{1}{\sqrt{n}}\sum_{i=1}^n \alpha_i K_{\boldsymbol{x}_i}, \text{ for } \boldsymbol{\alpha} = (\alpha_1, \alpha_2, \ldots, \alpha_n)^T \in \mathbb{R}^n.$$

We also define $S_{K,n}^\diamond : L_{\rho_\mathcal{X}}^2 \to \mathcal{H}_K, S_{K,n}^\diamond g = \frac{1}{n}\sum_{i=1}^n g(\boldsymbol{x}_i)K_{\boldsymbol{x}_i}$. The empirical covariance operator is defined as

$$C_{K,n} : \mathcal{H}_K \to \mathcal{H}_K, \quad C_{K,n} = S_{K,n}^* S_{K,n} = S_{K,n}^\diamond S_{K,n} = \frac{1}{n}\sum_{i=1}^n K_{\boldsymbol{x}_i} \otimes K_{\boldsymbol{x}_i}.$$

Our objective is to learn an estimator $\hat{f} \in \mathcal{H}_K$ from the training data that approximates the regression function $f_\rho$ defined in (2), i.e. the empirical risk minimizer, $\hat{f} = \operatorname{argmin}_{f \in \mathcal{H}_K} \frac{1}{n} \sum_{i=1}^n (f(\boldsymbol{x}_i) - y_i)^2$. Using the operator notations introduced above, the minimizer should satisfy the normal equation $C_{K,n}\hat{f} = S_{K,n}^* \mathbf{y}$, where $\mathbf{y} = \frac{1}{\sqrt{n}}(y_1, y_2, \ldots, y_n)^T$. However, directly solving the normal equation may lead to overfitting, and the operator $C_{K,n}$ may be singular.

To address these issues, spectral algorithms introduce a regularization mechanism by applying a filter function to the empirical covariance operator. This approach helps to stabilize the solution and mitigates overfitting by the regularization effect of the filter function. Specifically, a spectral algorithm constructs the estimator $f_{n,\lambda}$ as

$$f_{n,\lambda} = g_\lambda(C_{K,n})S_{K,n}^* \mathbf{y}, \tag{12}$$

where $\lambda > 0$ is a regularization parameter and $g_\lambda : [0, \kappa^2] \to \mathbb{R}$ is a filter function satisfying certain conditions (see Definition 3 below).

**Definition 3.** *A function $g_\lambda(\cdot) : [0, \kappa^2] \to \mathbb{R}^+$ indexed by $\lambda \in \Lambda \subset \mathbb{R}^+$ is called a filter function with qualification $\tau \geq 1$, if it satisfies the following conditions:*

$$\sup_{\nu \in [0,1]} \sup_{\lambda \in \Lambda} \sup_{t \in [0,\kappa^2]} |t^\nu g_\lambda(t)| \lambda^{1-\nu} \leq E, \quad \sup_{\nu \in [0,\tau]} \sup_{\lambda \in \Lambda} \sup_{t \in [0,\kappa^2]} |(1 - tg_\lambda(t))| t^\nu \lambda^{-\nu} \leq F_\tau, \tag{13}$$

*where $E$ and $F_\tau$ are positive constants independent of $\lambda$ and $t$.*

Different choices of the filter function $g_\lambda$ yield different spectral regularization schemes. We list some common spectral algorithms and their corresponding filter functions, qualifications, and constants in Appendix C.1.

Despite the theoretical appeal of spectral algorithms, their practical implementation can be computationally intensive, especially for large-scale datasets. The main computational bottleneck arises from computing and storing the empirical covariance operator $C_{K,n}$, which involves operations on an $n \times n$ kernel matrix (i.e., $n \times n$ matrix inverse in KRR). This can lead to significant time and space complexity (i.e., $O(n^3)$ and $O(n^2)$ for KRR), making it infeasible for large $n$. To address this challenge, various approximation techniques have been proposed to reduce the computational burden. One such technique is the use of random projections, which can effectively reduce the dimensionality of the data while preserving its essential structure (Vempala, 2005). The core idea of random projections is to restrict the original solution space to a lower-dimensional subspace using a projection operator $P$, satisfying $P^2 = P$. Incorporating random projections into spectral algorithms leads to the following estimator:

$$f_{n,\lambda}^{rp} = Pg_\lambda(PC_{K,n}P)PS_{K,n}^* \mathbf{y}. \tag{14}$$

This approach can significantly reduce the computational complexity of spectral algorithms, making them more scalable and efficient for large datasets. In the next subsections, we will discuss some specific implementations of SARP, focusing on randomized sketches and Nyström sub-sampling.

### 3.1 RANDOMIZED SKETCHES FOR SPECTRAL ALGORITHMS

Randomized sketches involve a sketch matrix $\mathbf{G} \in \mathbb{R}^{m \times n}$, where $m \ll n$, and the range of the projection operator $P$ is defined as the closure of the span of the rows of $\mathbf{G}S_{K,n}$, i.e., $\text{Range}(P) = \overline{\text{span}\{S_{K,n}^* \mathbf{G}^T\}}$. The sketch matrix $\mathbf{G}$ is typically chosen to be a random matrix with certain properties to ensure the preservation of the data structure after projection, i.e., for any $\mathbf{a} \in \mathbb{R}^n$ and $t \in (0,1)$, there exist constants $C > 0$ and $b \geq 0$ such that

$$\mathbb{P}\left(\|\mathbf{G}\mathbf{a}\|_2^2 \geq (1+t)\|\mathbf{a}\|_2^2\right) \geq 2\exp(-Cmt^2/(\log n)^b). \tag{15}$$

Common choices for $\mathbf{G}$ include sub-Gaussian random sketches, randomized orthogonal system (ROS) sketches, and sub-sampling sketches (Yang et al., 2017; Lin & Cevher, 2020a). We refer to Appendix C.2 for more details on these sketching methods.

### 3.2 NYSTRÖM SUB-SAMPLING FOR SPECTRAL ALGORITHMS

**Plain Nyström.** Nyström sub-sampling is equivalent to sub-sampling sketch using matrix $\mathbf{G}$, where the range of the projection operator $P$ is defined as the closure of the span of a subset of the

training data points. Specifically, we randomly and uniformly select a subset of $m$ data points $\{\boldsymbol{x}_{i_j}\}_{j=1}^m$ from the training dataset $\{\boldsymbol{x}_i\}_{i=1}^n$ without replacement, and define the range of $P$ as $\text{Range}(P) = \overline{\text{span}\{K_{\boldsymbol{x}_{i_j}}\}_{j=1}^m}$.

**Approximate leverage score (ALS) Nyström.** Alaoui & Mahoney (2015) proposed the leverage score sampling for Nyström sub-sampling, which can be viewed as a non-uniform sub-sampling sketch. The leverage scores measure the importance of each data point in the dataset:

$$(l_i(\lambda))_{i=1}^n, \quad l_i(\lambda) = \left(\mathbf{K}(\mathbf{K} + \lambda I)^{-1}\right)_{ii}, \quad i = 1, 2, \ldots, n.$$

where $\mathbf{K} = S_{K,n} S_{K,n}^*$ denotes the Gram matrix. In practice, the exact leverage scores are often approximated using fast algorithms (Musco & Musco, 2017; Calandriello et al., 2017; Rudi et al., 2018; Chen & Yang, 2021). The approximate leverage scores $\{\hat{l}_i(\lambda)\}_{i=1}^n$ satisfy $\hat{l}_i(\lambda)/L \leq l_i(\lambda) \leq L\hat{l}_i(\lambda)$ for some $L \geq 1$. The ALS Nyström method selects $m$ data points $\{\widetilde{\boldsymbol{x}}_{i_j}\}_{j=1}^m$ from the training dataset $\{\boldsymbol{x}_i\}_{i=1}^n$ according to the probability distribution $\{q_i\}_{i=1}^n$, where $q_i = \hat{l}_i(\lambda)/\sum_{j=1}^n \hat{l}_j(\lambda)$. The range of the projection operator $P$ is defined as $\text{Range}(P) = \overline{\text{span}\{K_{\widetilde{\boldsymbol{x}}_{i_j}}\}_{j=1}^m}$.

We discuss the time and space complexity of SARP with different techniques, and the comparison with related work in Appendix B.

## 4 MAIN RESULTS

In this section, we present our main theoretical results on the learning rates of SARP. For clarity, we introduce a projection operator term $\mathcal{A}_{K,P} = \|C_K^{1/2}(I - P)\|$ and the projection error $\Delta$, which will be given explicitly in the following theorem. We assume $s \in (0, 2\tau]$ throughout the analysis.

**Theorem 2.** *Under Assumptions 1-4, for any $\gamma \in [0, 1 \wedge s]$, $\lambda \in [0, 1]$ and $\delta \in (0, 1)$, if $n \gtrsim \lambda^{-\alpha} \log(8/\delta) \vee (\lambda^\alpha \phi(\lambda))^{-\frac{\alpha}{s+\alpha}}$, then with probability at least $1 - \delta$, the following holds:*

*(1) if $\phi : [0, \kappa^2] \to \mathbb{R}^+$ is non-decreasing, and $\phi(0) = 0$, $\phi(\kappa^2) < \infty$, or $\phi : [0, \kappa^2] \to \mathbb{R}^+$ is Lipschitz continuous with constant 1 and satisfies that $P\phi(A)P = \phi(PAP)$ for any operator $A$ and projection operator $P$, then*

$$\|f_{n,\lambda}^{rp} - f_\rho\|_{[\mathcal{H}]_\rho^\gamma} \lesssim \lambda^{-\gamma} \left[\phi(\lambda) + \frac{1}{n\lambda^{\max\{\frac{\alpha}{2}, \alpha - \frac{s}{2}\}}} + \frac{1}{\sqrt{n}\lambda^{\frac{1}{2\beta}}} + \frac{\phi(\lambda)}{\sqrt{n}\lambda^{\frac{\alpha}{2}}} + \Delta\right] \log\frac{8}{\delta}.$$

*(2) if $\phi = \psi\vartheta$, where $\psi : [0, \kappa^2] \to \mathbb{R}^+$ is non-decreasing, and $\psi(0) = 0$, $\psi(\kappa^2) < \infty$, and $\vartheta : [0, \kappa^2] \to \mathbb{R}^+$ is non-decreasing and Lipschitz continuous with constant 1 and satisfies that $P\phi(A)P = \phi(PAP)$ for any operator $A$ and projection operator $P$, and $\vartheta(0) = 0$, and $\forall\lambda \in [0, \kappa^2]$, there exists $c' > 0$ such that $c'\frac{\lambda^\tau}{\vartheta(\lambda)\lambda^{1/2}} \leq \inf_{t \in [\lambda, \kappa^2]} \frac{t^\tau}{\vartheta(t)t^{1/2}}$, then*

$$\|f_{n,\lambda}^{rp} - f_\rho\|_{[\mathcal{H}]_\rho^\gamma} \lesssim \lambda^{-\gamma} \left[\phi(\lambda) + \frac{1}{n\lambda^{\max\{\frac{\alpha}{2}, \alpha - \frac{s}{2}\}}} + \frac{1}{\sqrt{n}\lambda^{\frac{1}{2\beta}}} + \frac{\phi(\lambda)}{\sqrt{n}\lambda^{\frac{\alpha}{2}}} + \vartheta(\lambda)\psi(\frac{1}{\sqrt{n}}) + \Delta\right] \log\frac{8}{\delta}.$$

*Here the projection error $\Delta$ is defined as* $\Delta = \begin{cases} \lambda^{\frac{s}{2}-1}(\mathcal{A}_{K,P}^2 + \lambda^{\frac{1}{2}}\mathcal{A}_{K,P} + \lambda^{\frac{\gamma}{2}}\mathcal{A}_{K,P}^{2-\gamma}), & \text{if } s < 2; \\ \mathcal{A}_{K,P}^2 + n^{-\frac{1}{4}}\mathcal{A}_{K,P} + \lambda^{\frac{\gamma}{2}}\mathcal{A}_{K,P}^{2-\gamma}, & \text{if } s \geq 2. \end{cases}$

Theorem 2 establishes high-probability Sobolev norm error bounds for the SARP under general source conditions. Here we consider three types of source conditions: (1) non-decreasing index function $\phi$; (2) Lipschitz continuous index function $\phi$; (3) composite index function $\phi = \psi\vartheta$, which also appear in Bauer et al. (2007); Lin & Cevher (2020b). The derived learning rates can be decomposed into several terms reflecting different aspects of the learning process. The term $\lambda^{-\frac{\gamma}{2}}\phi(\lambda)$ represents the approximation error, which depends on the regularization parameter $\lambda$ and the smoothness of the target function $f_\rho$ characterized by the index function $\phi$. The intermediate terms involving $n, \lambda, \beta$, and $\alpha$ represent the empirical error, which captures the effect of finite sample variance. The last term $\Delta$ accounts for the impact of random projections on learning performance. When $P = I$ (so that $\Delta = 0$), our result recovers the existing bounds in Lin & Cevher (2020b) for spectral algorithms without RP. To achieve the optimal trade-off between the error terms, we can choose the regularization parameter $\lambda$ appropriately, which leads to the following theorem.

**Theorem 3.** *Under the same assumptions and conditions as Theorem 2, let $s \geq \alpha - 1/\beta$, and $\lambda = \Phi^{-1}(n^{-1})$ with $\Phi(t) = (\phi(t)/\phi(1))^2 t^{1/\beta}$ and $\Phi^{-1}(\cdot)$ denoting the functional inverse, for case (2), we additionally let $\lambda \geq n^{-1/2}$, then with probability at least $1 - \delta$, there holds that*

$$\|f_{n,\lambda}^{rp} - f_\rho\|_{[\mathcal{H}]_\rho^\gamma}^2 \lesssim (\phi(\Phi^{-1}(n^{-1})) + \Delta)^2 (\Phi^{-1}(n^{-1}))^{-\gamma} \log^2 \frac{8}{\delta},$$

*where the projection error $\Delta$ is defined in Theorem 2.*

Compared to the lower bound in Theorem 1, Theorem 3 shows that the SARP attains the minimax-optimal learning rates up to a logarithmic factor whenever the projection error $\Delta$ is of a smaller order than $\phi(\Phi^{-1}(n^{-1}))$. This condition can be satisfied by choosing an appropriate projection dimension $m$ for different types of random projections, as discussed in the next section. When $P = I$ (so that $\Delta = 0$), the derived learning rates align with those in Bauer et al. (2007) and Lin & Cevher (2020b) for spectral algorithms under general source conditions. However, our results are more general as they encompass a broader range of source index $s \geq \alpha - 1/\beta$ and Sobolev norms $\gamma \in [0, 1 \wedge s]$, while the previous works primarily focus on the well-specified case with $s \geq 1$ (Bauer et al., 2007) and part of the mis-specified case $s \geq 1 - 1/\beta$ (Lin & Cevher, 2020b).

### 4.1 Learning Rates for Specific Random Projections

In this section, we apply Theorem 3 to three specific types of random projections introduced earlier: randomized sketches, plain, and ALS Nyström sub-sampling. We show how to choose the projection dimension $m$ to effectively control the projection error $\Delta$, thereby achieving optimality. For simplicity, we focus on the Hölder source condition with $\phi(t) = t^{s/2}$ in the following corollary.

**Corollary 2.** *Under the same assumptions and conditions as Theorem 2, let $\phi(t) = \kappa^{-(s-2)+} t^{\frac{s}{2}}$ for some $s \in (0, 2\tau]$, we consider the following three cases of $s$:*

*(1) When $s < \alpha - \frac{1}{\beta}$, let $\lambda = n^{-\frac{1}{\alpha}}$, $n \geq n_0$ for some $n_0 \geq 0$, and the projection dimension $m$ satisfies*

$$m \gtrsim \begin{cases} n^{\frac{1}{\alpha\beta}}, & \text{for Randomized Sketches and ALS Nyström;} \\ n, & \text{for Plain Nyström.} \end{cases}$$

*Then with probability at least $1 - \delta$, there holds that*

$$\|f_{n,\lambda}^{rp} - f_\rho\|_{[\mathcal{H}]_\rho^\gamma}^2 \lesssim n^{-\frac{s-\gamma}{\alpha}} \log^2 \frac{10}{\delta}.$$

*(2) When $s \geq \alpha - \frac{1}{\beta}$ and $s < 2$, $n \geq n_0$ for some $n_0 \geq 0$, let $\lambda = n^{-\frac{\beta}{s\beta+1}}$, and the projection dimension $m$ satisfies that*

$$m \gtrsim \begin{cases} n^{\frac{1}{s\beta+1}}, & \text{for Randomized Sketches and ALS Nyström;} \\ n^{\frac{\alpha\beta}{s\beta+1}}, & \text{for Plain Nyström.} \end{cases}$$

*Then with probability at least $1 - \delta$, there holds that*

$$\|f_{n,\lambda}^{rp} - f_\rho\|_{[\mathcal{H}]_\rho^\gamma}^2 \lesssim n^{-\frac{(s-\gamma)\beta}{s\beta+1}} \log^2 \frac{10}{\delta}.$$

*(3) When $s \geq 2$, let $\lambda = n^{-\frac{\beta}{s\beta+1}}$, $n \geq n_0$ for some $n_0 \geq 0$, and the projection dimension $m$ satisfies that*

$$m \gtrsim \begin{cases} n^{\frac{s-\gamma}{(2-\gamma)(s\beta+1)}}, & \text{for Randomized Sketches and ALS Nyström;} \\ n^{\frac{\alpha\beta(s-\gamma)}{(2-\gamma)(s\beta+1)}}, & \text{for Plain Nyström.} \end{cases}$$

*Then with probability at least $1 - \delta$, there holds that*

$$\|f_{n,\lambda}^{rp} - f_\rho\|_{[\mathcal{H}]_\rho^\gamma}^2 \lesssim n^{-\frac{(s-\gamma)\beta}{s\beta+1}} \log^2 \frac{10}{\delta}.$$

Corollary 2 provides capacity-dependent learning rates for spectral algorithms with different types of random projections under the Hölder source condition. The results demonstrate that by selecting an appropriate projection dimension $m$ (proportional to the average or maximal effective dimension, see Remark 6 in the Appendix for the details), the projection error $\Delta$ can be controlled to be of a smaller order than the approximation and empirical error, thereby achieving the minimax optimal learning rates $n^{-(s-\gamma)\beta/(s\beta+1)}$ up to a logarithmic factor when $s \geq \alpha - 1/\beta$. The learning rates degrade to $n^{-(s-\gamma)/\alpha}$ when $s < \alpha - 1/\beta$. In the most benign case when $\alpha = 1/\beta$, the optimal learning rates are achieved for all $s \in (0, 2\tau]$. Examples of such benign RKHSs are provided in Appendix C.3.

**Remark 1.** *Compared to the existing results for spectral algorithms (or KRR) with randomized random projection under the Hölder source condition, our guarantees extend to the mis-specified case with $s < \alpha - 1/\beta$ and Sobolev norms $\gamma \in [0, 1 \wedge s]$. For example, Yang et al. (2017) only considered the worst case with $s = 1$ and $\gamma = 0$ under fix design, Rudi et al. (2015) established the optimal learning rates on the well-specified case with $s \geq 1$ and $\gamma = 0$, the state-of-the-art results Lin & Cevher (2020a) focused on various norm of learning rates under part of the mis-specified case with $s \geq 1 - 1/\beta$. This extension comes from tighter estimates of (i) the similarity between the covariance operator $C_K$ and the sample covariance operator $C_{K,n}$, leading to the condition on $n$ from $n \gtrsim \lambda^{-1}$ to $n \gtrsim \lambda^{-\alpha}$, and (ii) the sample-variance term, improving it roughly from $\mathcal{O}(n^{-1}\lambda^{\min\{-1/2, s/2-1\}} + n^{1/2}\lambda^{(s-1)/2})$ to $\mathcal{O}(n^{-1}\lambda^{\min\{-\alpha/2, s/2-\alpha\}} + n^{1/2}\lambda^{(s-\alpha)/2})$. Different from the analysis in Lin & Cevher (2020a), we further do not require the uniform boundedness of $f_\rho$, which is discussed in Assumption 1.*

**Remark 2.** *For the case when $s > 2\tau$, a saturation effect occurs, where the learning rates do not depend on the source index $s$ anymore and are the same as the case when $s = 2\tau + \gamma$. This phenomenon is well known in the theory of inverse problems (Bauer et al., 2007) and recently has been proved in the context of KRR ($\tau = 1$) in (Li et al., 2023b) and spectral algorithms in (Lu et al., 2024). Thus, we only focus on $s \in (0, 2\tau]$ in this paper.*

**Remark 3.** *Computationally, for randomized sketches and ALS Nyström, the required projection dimension $m$ is significantly smaller than the sample size $n$, leading to substantial computational savings. In contrast, plain Nyström requires a larger projection dimension, which may limit its computational efficiency in certain scenarios ($s < \alpha - 1/\beta$). Compared to the optimal projection dimension derived in Lin & Cevher (2020a), the required projection dimension for the optimal rates remains the same as in Lin & Cevher (2020a) for randomized sketches and ALS Nyström, while the source range extends from $s \geq 1 - 1/\beta$ to $s \geq \alpha - 1/\beta$. For plain Nyström, our results reduce the required projection dimension from $m \gtrsim n^{\beta/(s\beta+1)}$ when $s \in [1 - 1/\beta, 2)$ to $m \gtrsim n^{\alpha\beta/(s\beta+1)}$ when $s \in [\alpha - 1/\beta, 2)$, from $m \gtrsim n^{\frac{\beta(s-\gamma)}{(2-\gamma)(s\beta+1)}}$ to $m \gtrsim n^{\frac{\alpha\beta(s-\gamma)}{(2-\gamma)(s\beta+1)}}$ when $s \geq 2$. In the most benign case when $\alpha = 1/\beta$, the required projection dimension further reduces to those in randomized sketches and ALS Nyström. These improvements are particularly due to our refined analysis on the operator similarly and the sample variance control, see Appendix E.4 for more details.*

For a more detailed comparison on the learning rates, the corresponding conditions, computational and storage aspects with existing results in the literature, we refer to Table 2 in the Appendix.

# 5 CONCLUSION

In this paper, we have established high-probability Sobolev norm error bounds for SARP under general source conditions. Our results demonstrate that, by selecting an appropriate projection dimension and regularization parameter, these algorithms can achieve minimax optimal learning rates up to a logarithmic factor. Notably, our analysis extends to the mis-specified case with $s \geq \alpha - 1/\beta$ and Sobolev norms $\gamma \in [0, 1 \wedge s]$, which were not fully addressed in previous works. Our findings contribute to a deeper understanding of the theoretical properties of SARP and provide practical insights for its implementation in large-scale learning tasks. Future research may proceed in several directions. First, our analysis techniques may apply to other scalable kernel learning algorithms, such as random features Rudi & Rosasco (2017); Wang & Feng (2024) and stochastic gradient methods Lin & Rosasco (2016). Second, it is natural to extend our analysis to vector-valued settings, for example, in connection with conditional mean embeddings Li et al. (2022). Third, one may relax the noise assumption to the heavy-tailed case and combine our SARP framework with robust concentration tools, which have recently been developed by Mollenhauer et al. (2025). Finally, a fully general information-theoretic lower bound on the projection dimension $m$ for SARP is also an open problem.

## ETHICS STATEMENT

This paper presents work whose goal is to advance the field of learning theory. There is no potential violations of the ICLR Code of Ethics. None of potential societal consequences of our work should be specifically highlighted here.

## REPRODUCIBILITY STATEMENT

The main text and the Appendix present the theoretical results, provide clear explanations of all assumptions, and include complete proofs of the claims. Because the primary contribution is theoretical, the experiments serve only to validate the theoretical findings. These experiments are straightforward to implement using standard Python libraries, and the real-world LIBSVM datasets used are publicly available.

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

APPENDIX CONTENTS

## THE USE OF LARGE LANGUAGE MODELS (LLMS)

In this paper, we used Large Language Models (LLMs) to assist with and refine the writing, improving grammar without altering the technical content or ideas presented. We have carefully reviewed all material and take full responsibility for the final version of the paper.

## A  NOTATIONS

Let $\Omega$ be a subset of a Euclidean space and $\mathcal{P}(\Omega)$ be the set of all probability measures on $\Omega$. For any $\rho \in \mathcal{P}(\Omega)$, we denote by $L_\rho^p(\Omega)$ the space of all $p$-th power integrable functions $f : \Omega \to \mathbb{R}$. For $p = 2$ and $f, g \in L_\rho^2(\Omega)$, we define the inner product and norm as $\langle f, g \rangle_\rho = \int f(\boldsymbol{x})g(\boldsymbol{x})d\rho(\boldsymbol{x})$ and $\|f\|_\rho = \langle f, f \rangle_\rho^{1/2}$, respectively. Additionally, we denote by $L_\rho^\infty(\Omega)$ the space of all essentially bounded functions $f : \Omega \to \mathbb{R}$ with the norm $\|f\|_{L_\rho^\infty(\Omega)} = \operatorname{ess\,sup}_{\boldsymbol{x}\in\Omega}|f(\boldsymbol{x})|$. For two Banach spaces $\mathcal{B}_1$ and $\mathcal{B}_2$ and $\mathcal{B}_1 \subset \mathcal{B}_2$, we denote by $\mathcal{B}_1 \hookrightarrow \mathcal{B}_2$ if $\mathcal{B}_1$ is continuously embedded in $\mathcal{B}_2$. We use $\|\mathcal{B}_1 \hookrightarrow \mathcal{B}_2\| = \sup_{u\in\mathcal{B}_1} \|u\|_{\mathcal{B}_2}$ to denote the embedding norm. For two Hilbert spaces $\mathcal{H}_1$ and $\mathcal{H}_2$, and a linear operator $A : \mathcal{H}_1 \to \mathcal{H}_2$, we denote the operator norm as $\|A\| = \sup_{u\in\mathcal{H}_1, u\neq 0} \frac{\|Au\|_{\mathcal{H}_2}}{\|u\|_{\mathcal{H}_1}}$, here we omit the dependence of $A$ on $\mathcal{H}_1$ and $\mathcal{H}_2$ for simplicity. For $u \in \mathcal{H}_1$ and $v \in \mathcal{H}_2$, we denote by $v \otimes u : \mathcal{H}_1 \to \mathcal{H}_2$ the operator as $(v \otimes u)w = \langle u, w \rangle_{\mathcal{H}_1} v$ for any $w \in \mathcal{H}_1$. We denote by $I$ the identity operator and $I_n$ the identity matrix of size $n$. For a vector $\mathbf{a}$, we denote by $\|\mathbf{a}\|_2$ the Euclidean norm. For two sequences $\{a_n\}_{n\geq 1}$ and $\{b_n\}_{n\geq 1}$, we write $a_n \lesssim b_n$ if there exists a constant $C > 0$ such that $a_n \leq Cb_n$. We write $a_n \asymp b_n$ if both $a_n \lesssim b_n$ and $b_n \lesssim a_n$ hold. We also use the big-O notation $O(\cdot)$, $\Omega(\cdot)$, and the small-o notation $o(\cdot)$ in the usual sense. For a positive integer $m$, we denote by $[m]$ the set $\{1, 2, \ldots, m\}$.

We also list some frequently used notations in Table 1 for easy reference.

## B  COMPARISON WITH THE RELATED WORK

In this section, we present a detailed comparison of our results with related work on spectral algorithms and their variants incorporating random projections. The discussion covers kernel ridge regression (KRR), kernel principal component regression (KPCR), gradient-based methods, general spectral algorithms, randomized sketches, Nyström sub-sampling, general random projections, and their combinations. Table 2 summarizes the conditions for the derived learning rates, time, and space complexities of different methods.

**Comparison with spectral algorithms.** Spectral learning offers a unified framework for regularizing ill-posed inverse problems (Engl et al., 1996) and has been extensively analyzed within learning theory. A substantial body of research has established optimal learning rates for spectral algorithms under various conditions, including Hölder source conditions (Gerfo et al., 2008; Bauer et al., 2007), general source conditions (Rastogi & Sampath, 2017), and polynomial eigenvalue decay (Caponnetto, 2006). Related studies have also investigated specific methods such as kernel ridge regression (KRR) (Caponnetto & De Vito, 2007; Smale & Zhou, 2007), kernel principal component regression (KPCR) (Blanchard et al., 2007; Dicker et al., 2017), and early-stopped gradient schemes (Yao et al., 2007). Despite these advances, most existing results primarily focus on the well-specified case with $s \geq 1$, and evaluate performance mainly in the traditional $L_{\rho_\mathcal{X}}^2$ norm or RKHS norm. More recently, there has been increasing interest in mis-specified spectral learning, where the regression function does not belong to the RKHS. Among these works, some have derived optimal learning rates for parts of the mis-specified setting with $s \geq 1 - 1/\beta$, including KRR (Wang & Jing, 2022), gradient methods (Lin & Rosasco, 2017; Lin et al., 2017), and general spectral algorithms (Lin & Cevher, 2020b).

Another line of research combines the embedding property of RKHSs with integral operator techniques to obtain optimal learning rates for spectral algorithms in mis-specified settings, covering KRR (Steinwart et al., 2009; Fischer & Steinwart, 2020; Zhang et al., 2023), gradient methods (Pillaud-Vivien et al., 2018), and spectral algorithms (Celisse & Wahl, 2021; Zhang et al., 2024). Our work differs from these studies in several key aspects: (1) we focus on spectral algorithms with random projections, whereas the above works address spectral algorithms without random projections; (2) we derive high-probability Sobolev norm error bounds under general source conditions, while

Table 1: Table of Notations

| Notation | Description |
|---|---|
| $\mathcal{X}, \mathcal{Y}$ | Input space, compact subset of $\mathbb{R}^d$, Output space, subset of $\mathbb{R}$ |
| $\rho, \rho_{\mathcal{X}}$ | Joint distribution on $\mathcal{X} \times \mathcal{Y}$, Marginal distribution on $\mathcal{X}$ |
| $\rho(y|\boldsymbol{x})$ | Conditional distribution of $y$ given $\boldsymbol{x}$ |
| $f_\rho(\boldsymbol{x})$ | True regression function, $f_\rho(\boldsymbol{x}) = \mathbb{E}[y|\boldsymbol{x}]$ |
| $\varepsilon$ | Noise term, $\varepsilon = y - f_\rho(\boldsymbol{x})$ |
| $K(\cdot, \cdot), K_{\boldsymbol{x}}$ | Positive definite kernel function, $K_{\boldsymbol{x}}(\cdot) = K(\boldsymbol{x}, \cdot)$ |
| $\mathcal{H}_K$ | Reproducing kernel Hilbert space induced by $K$ |
| $L^2_{\rho_{\mathcal{X}}}$ | $L^2$-space of square-integrable functions with respect to $\rho_{\mathcal{X}}$ |
| $\kappa$ | Upper bound on kernel function, $\sup_{\boldsymbol{x} \in \mathcal{X}} K(\boldsymbol{x}, \boldsymbol{x}) \leq \kappa^2$ |
| $\| \cdot \|_{\mathcal{H}}$ | RKHS norm |
| $\| \cdot \|_\rho$ | $L^2_{\rho_{\mathcal{X}}}$ norm, $\|f\|_\rho^2 = \int_{\mathcal{X}} f^2(\boldsymbol{x}) d\rho_{\mathcal{X}}$ |
| $S_K$ | Inclusion operator, $S_K : \mathcal{H}_K \to L^2_{\rho_{\mathcal{X}}}$ |
| $S_K^*$ | Adjoint of inclusion operator |
| $L_K$ | Integral operator, $L_K = S_K S_K^*$ |
| $C_K$ | Covariance operator, $C_K = S_K^* S_K$ |
| $C_{K,n}$ | Sample covariance operator |
| $S_{K,n}$ | Sampling operator |
| $S_{K,n}^*$ | Adjoint of sampling operator |
| $C_{K,\lambda}$ | Regularized covariance operator, $C_{K,\lambda} = C_K + \lambda I$ |
| $C_{K,n,\lambda}$ | Regularized sample covariance operator, $C_{K,n,\lambda} = C_{K,n} + \lambda I$ |
| $P$ | Projection operator satisfying $P^2 = P$ |
| $\mathcal{A}_{K,P}$ | Projection error term, $\mathcal{A}_{K,P} = \|(I - P)C_K^{1/2}\|$ |
| $\Delta, \Delta_1, \Delta_2$ | Random Projection error |
| $\mathbf{y}$ | Vector of normalized outputs, $\mathbf{y} = \frac{1}{\sqrt{n}}(y_1, \ldots, y_n)^T$ |
| $\mathbf{K}$ | Kernel matrix, $\mathbf{K} = (K(\boldsymbol{x}_i, \boldsymbol{x}_j))_{i,j=1}^n$ |
| $\mathbf{G}$ | Random projection matrix, $\mathbf{G} \in \mathbb{R}^{m \times n}$ |
| $m$ | Projection dimension |
| $\lambda$ | Regularization parameter |
| $g_\lambda(\cdot)$ | Filter function parameterized by $\lambda$ |
| $\tau$ | Qualification of spectral algorithm |
| $f_{n,\lambda}^{rp}$ | Spectral algorithm with random projections estimator |
| $\mathcal{E}(f)$ | Expected risk of function $f$ |
| $\mathcal{E}_n(f)$ | Empirical risk of function $f$ |
| $s$ | Source condition parameter, $s > 0$ |
| $\beta$ | Average capacity condition parameter, $\beta \in [1, \infty)$ |
| $\alpha$ | embedding condition parameter, $\alpha \in [1/\beta, 1]$ |
| $\gamma$ | Sobolev norm parameter, $\gamma \in [0, s \wedge 1]$ |
| $\phi(\cdot)$ | Source index function |
| $\psi(\cdot), \nu(\cdot)$ | Composition functions of source index function, $\phi = \psi\nu$ |
| $\Phi(\cdot)$ | Function $\Phi(t) = (\phi(t)/\phi(1))^2 t^\gamma$ |
| $\Omega_{\phi,s,R}$ | Source condition set |
| $R$ | Smoothness parameter |
| $\mathcal{N}_{\boldsymbol{x}}(\lambda)$ | Random variable $\langle K_{\boldsymbol{x}}, (C_K + \lambda I)^{-1} K_{\boldsymbol{x}} \rangle_{\mathcal{H}}$ |
| $\mathcal{N}(\lambda)$ | Average effective dimension |
| $\mathcal{N}_\infty(\lambda)$ | Maximal effective dimension |
| $Q$ | Constant in average capacity condition |
| $A$ | Constant in embedding condition |
| $\sigma$ | Noise variance parameter |
| $M$ | Light-tailed noise parameter |
| $\delta$ | Confidence parameter |
| $\mathcal{P}$ | Set of probability distributions satisfying assumptions |
| $\{\mu_k\}_{k\geq 1}, \{e_k\}_{k\geq 1}$ | Eigenvalues and eigenfunctions of the integral operator $L_K$ |

Hölder source condition is a special case; (3) Our theoretical error decomposition and analysis take a different form, as detailed in Section E.

**Comparison with spectral algorithms with random projections.** Random projections have been widely employed to scale up kernel methods, and several studies have analyzed the learning performance of both specific and general spectral algorithms with random projections. Yang et al. (2017) established optimal learning rates for KRR with randomized sketches under fixed design and the source condition with $s = 1$. Wang et al. (2018) further investigated the statistical and optimization effects of classical sketches and Hessian sketches used to approximately solve matrix ridge regression. For the Nyström sub-sampling, Bach (2013); Alaoui & Mahoney (2015) derived sharp learning rates for KRR in expectation under fixed design. The breakthrough work of Rudi et al. (2015) obtained capacity-dependent learning rates for KRR with plain and ALS Nyström under random design. However, these optimal statistical guarantees assumed that the target function belongs to the RKHS and only considered the traditional $L^2_{\rho_\mathcal{X}}$ norm. For the mis-specified case, Kriukova et al. (2017) and Lu et al. (2019) provided the generalization guarantees for Nyström approximations, but their results were either suboptimal or capacity-independent. Li et al. (2023a) introduced a compatibility condition to derive optimal learning rates for KRR with Nyström sub-sampling under the mis-specified case where $s \geq \alpha - 1/\beta$. While the derived learning rates are optimal in the $L^2_{\rho_\mathcal{X}}$ norm for $s \leq 2$, they do not extend to Sobolev norms and $s > 2$. Moreover, their analysis requires the uniform boundedness of $f_\rho$, and the compatibility condition is hard to verify in practice. The above works mainly focus on KRR with random projections. Recently, Lin & Cevher (2020a) derived optimal learning rates for general spectral algorithms with randomized sketches and with plain and ALS Nyström under random design, and they consider part of the mis-specified case with $s \geq 1 - 1/\beta$ in various norms. Nevertheless, their analysis only consider the Hölder source condition and require the uniform boundedness of $f_\rho$, which is a strong assumption and may not hold in many practical scenarios.

Compared to these works, our work extends the analysis of SARP to the mis-specified case with $s \geq \alpha - 1/\beta$ and Sobolev norms $\gamma \in [0, 1 \wedge s]$ under more general assumptions, which were not covered in prior studies. Moreover, our analytical techniques yield improved conditions on the sample size $n$ and projection dimension $m$ for achieving optimal learning rates, as discussed in Remarks 3.

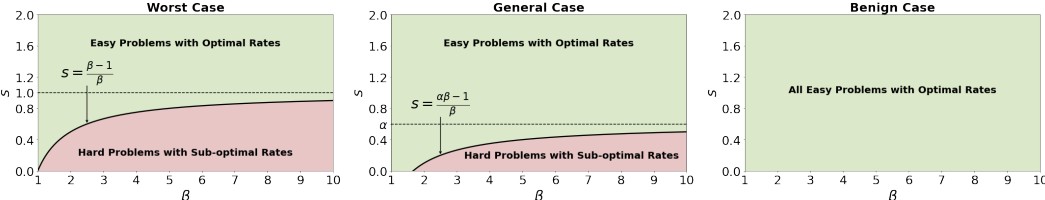

Figure 1: Comparison of the range of optimality for Lin & Cevher (2020a) and our work for SARP. The green region represents the easy problem with optimal rates, while the pink region represents the hard problem with sub-optimal rates. The dashed line represents the boundary of the easy and hard problems. From left to right represent the worst case ($\alpha = 1$), general case ($\alpha \in [\frac{1}{\beta}, 1]$), and benign case ($\alpha = \frac{1}{\beta}$).

**Comparison in the time and space complexity.** The time and space complexities of spectral algorithms varies depending on the specific method. For instance, KRR and spectral cutoff have time complexity of $O(n^3)$ and space complexity of $O(n^2)$, respectively, wheras the gradient-based method require $O(Tn^2)$ time and $O(Tn)$ space with $T$ denoting the iterations, respectively. For simplicity, we use $O(n^3)$ and $O(n^2)$ to denote the time and space complexity of spectral algorithms. For spectral algorithms with random projections, the time and space complexities reduce to $O(nm^2)$ and $O(nm)$, where $m \ll n$ is the projection dimension, thereby significantly improving the computational and storage efficiency. The exact complexities for specific random projection methods may vary depending on the optimal choice of $m$.

Compared to the state-of-the-art results in Lin & Cevher (2020a), our proposed SARP achieves optimal learning rates with lower (or equal) time and space complexity. For example, for plain Nyström, the time and space complexities reduce from $O(n^{\frac{(s+2)\beta+1}{s\beta+1}})$ and $O(n^{\frac{(s+1)\beta+1}{s\beta+1}})$ when $s \in$

Table 2: Summary of conditions for the derived learning rates and time-space complexities in different methods. ("KRR" stands for Kernel Ridge Regression, "GM" stands for Gradient Method, "KPCR" stands for Kernel Principal Component Regression, "KRR-RS" stands for Kernel Ridge Regression with randomized sketches, "KRR-Ny" stands for Kernel Ridge Regression with Nyström sub-sampling, "KRR-Ny-Plain" stands for Kernel Ridge Regression with Plain Nyström, "KRR-Ny-ALS" stands for Kernel Ridge Regression with ALS Nyström, "SA" stands for Spectral Algorithms, "SA-RP" stands for Spectral Algorithms with random projections, "SA-RS" stands for Spectral Algorithms with randomized sketches, "SA-Ny" stands for Spectral Algorithms with Nyström sub-sampling, "SA-Ny-Plain" stands for Spectral Algorithms with Plain Nyström, "SA-Ny-ALS" stands for Spectral Algorithms with ALS Nyström, and "UBC" stands for Uniform Boundedness Condition.

| Method | Source | Capacity | Embedding | UBC | Sobolev-norms | Lower bound | Convergence | Upper bound | Optimality | Projection dimension $m$ | Time | Space |
|---|---|---|---|---|---|---|---|---|---|---|---|---|
| KRR (Smale & Zhou, 2007) | Hölder, $s \in (0, 2]$ | ✗ | ✗ | ✓ | ✗ | ✗ | Probability | | ✓ | ✗ | $n^3$ | $n^2$ |
| KRR (Caponnetto & De Vito, 2007) | Hölder, $s \in [1, 2]$ | ✓ | ✗ | ✓ | ✗ | ✓ | Probability | | ✓ | ✗ | $n^3$ | $n^2$ |
| KRR (Steinwart et al., 2009) | Hölder, $s \in (0, 1]$ | ✓ | ✓ | ✓ | ✗ | ✓ | Probability | | ✓ | ✗ | $n^3$ | $n^2$ |
| KRR (Fischer & Steinwart, 2020) | Hölder, $s \in (0, 2]$ | ✓ | ✓ | ✓ | ✓ | ✓ | Probability | | ✓ | ✗ | $n^3$ | $n^2$ |
| KRR (Zhang et al., 2023) | Hölder, $s \in (0, 2]$ | ✓ | ✗ | ✓ | ✓ | ✓ | Probability | | ✓ | ✗ | $n^3$ | $n^2$ |
| GM (Yao et al., 2007) | Hölder, $s \in (0, 2]$ | ✗ | ✗ | ✓ | ✗ | ✓ | Probability | | ✗ | ✗ | $n^3$ | $n$ |
| GM (Pillaud-Vivien et al., 2018) | Hölder, $s \in (1, 2]$ | ✓ | ✗ | ✓ | ✗ | ✗ | Expectation | | ✓ | ✗ | $n^3$ | $n$ |
| KPCR (Dicker et al., 2017) | Hölder, $s \in (1, 2]$ | ✓ | ✗ | ✓ | ✗ | ✗ | Probability | | ✓ | ✗ | $n^3$ | $n^2$ |
| SA (Gerfo et al., 2008) | Hölder, $s \in [1, 2\tau]$ | ✗ | ✗ | ✓ | ✗ | ✗ | Probability | | ✓ | ✗ | $n^3$ | $n^2$ |
| SA (Bauer et al., 2007) | Hölder, $s \in [1, 2\tau]$ | ✗ | ✗ | ✓ | ✗ | ✗ | Probability | | ✓ | ✗ | $n^3$ | $n^2$ |
| SA (Bauer et al., 2007) | General, $s \in [1, 2\tau]$ | ✗ | ✗ | ✓ | ✗ | ✗ | Probability | | ✓ | ✗ | $n^3$ | $n^2$ |
| SA (Caponnetto, 2006) | Hölder, $s \in [1 - \frac{2}{\beta}, 2\tau]$ | ✓ | ✗ | ✓ | ✗ | ✓ | Probability | | ✓ | ✗ | $n^3$ | $n^2$ |
| SA (Rastogi & Sampath, 2017) | General, $s \in [1, 2\tau]$ | ✓ | ✗ | ✓ | ✗ | ✓ | Probability | | ✓ | ✗ | $n^3$ | $n^2$ |
| SA (Rastogi & Sampath, 2017) | Hölder, $s \in [1, 2\tau]$ | ✓ | ✗ | ✓ | ✗ | ✓ | Probability | | ✓ | ✗ | $n^3$ | $n^2$ |
| SA (Lin & Cevher, 2020b) | General, $s \in [1 - \frac{1}{\beta}, 2\tau]$ | ✓ | ✗ | ✓ | ✗ | ✓ | Probability | | ✓ | ✗ | $n^3$ | $n^2$ |
| SA (Lin & Cevher, 2020b) | Hölder, $s \in [1, 2\tau]$ | ✓ | ✗ | ✓ | ✓ | ✓ | Probability | | ✓ | ✗ | $n^3$ | $n^2$ |
| SA (Zhang et al., 2024) | Hölder, $s \in [1, 2\tau]$ | ✓ | ✓ | ✓ | ✓ | ✓ | Probability | | ✓ | ✗ | $n^3$ | $n^2$ |
| KRR-RS Yang et al. (2017) | Hölder, $s = 1$ | ✓ | ✗ | ✓ | ✗ | ✗ | Probability | | ✓ | | | |
| KRR-Ny (Kriukova et al., 2017) | Hölder, $s \in (0, 2]$ | ✗ | ✗ | ✓ | ✗ | ✗ | Probability | | ✗ | | | |
| KRR-Ny-Plain (Rudi et al., 2015) | Hölder, $s \in [1, 2]$ | ✓ | ✗ | ✓ | ✗ | ✗ | Probability | | ✓ | | | |
| KRR-Ny-ALS (Rudi et al., 2015) | Hölder, $s \in [1, 2]$ | ✓ | ✗ | ✓ | ✗ | ✗ | Probability | | ✓ | | | |
| KRR-Ny-Plain (Li et al., 2023a) | Hölder, $s \in [1 - \frac{1}{\beta}, 2]$ | ✓ | ✗ | ✓ | ✗ | ✗ | Probability | | ✓ | | | |
| KRR-Ny-ALS (Li et al., 2023a) | Hölder, $s \in (0, 2]$ | ✓ | ✗ | ✓ | ✗ | ✗ | Probability | | ✓ | | | |
| SA-RP (Lin & Cevher, 2020a) | Hölder, $s \in [1 - \frac{1}{\beta}, \infty)$ | ✓ | ✗ | ✓ | ✓ | ✗ | Probability | | ✓ | – | $nm^2$ | $nm$ |
| **SA-RP (Theorem 3)** | General, $s \in [\alpha - \frac{1}{\beta}, \infty)$ | ✓ | ✓ | ✓ | ✓ | ✓ | Probability | | ✓ | | $nm^2$ | $nm$ |
| **SA-RS (Corollary 2)** | Hölder, $s \in (0, 2\tau]$ | ✓ | ✓ | ✓ | ✓ | ✓ | Probability | | ✓ | | | |
| **SA-Ny-Plain (Corollary 2)** | Hölder, $s \in (0, 2\tau]$ | ✓ | ✓ | ✓ | ✓ | ✓ | Probability | | ✓ | | | |
| **SA-Ny-ALS (Corollary 2)** | Hölder, $s \in (0, 2\tau]$ | ✓ | ✓ | ✓ | ✓ | ✓ | Probability | | ✓ | | | |

$[1 - \frac{1}{\beta}, 2)$ to $O(n^{\frac{(s+2\alpha)\beta+1}{s\beta+1}})$ and $O(n^{\frac{(s+\alpha)\beta+1}{s\beta+1}})$ when $s \in [\alpha - \frac{1}{\beta}, 2)$, from $O(n^{1+\frac{2\beta(s-\gamma)}{(2-\gamma)(s\beta+1)}})$ and $O(n^{1+\frac{\beta(s-\gamma)}{(2-\gamma)(s\beta+1)}})$ to $O(n^{1+\frac{2\alpha\beta(s-\gamma)}{(2-\gamma)(s\beta+1)}})$ and $O(n^{1+\frac{\alpha\beta(s-\gamma)}{(2-\gamma)(s\beta+1)}})$ when $s \geq 2$. For the randomized sketches and ALS Nyström, the time and space complexity match those in Lin & Cevher (2020a), while we extend the source range from $s \geq 1 - \frac{1}{\beta}$ to $s \geq \alpha - \frac{1}{\beta}$. When $s < \alpha - \frac{1}{\beta}$ and the optimality does not hold, the time and space complexity of SARP are slightly larger than those in Lin & Cevher (2020a). This is reasonable since we consider a more difficult case with a smaller source index $s$. More detailed results regarding time and space complexity are provided in the last two columns of Table 2.

**Proof Novelty.** Compared with the existing SA literature, our upper-bound proof has to handle an additional error term induced by random projections, which is controlled by the operator similarity parameter $\mathcal{A}_{K,P}$. This term also enters the empirical error term, so the analysis cannot be reduced to a direct adaptation of classical SA arguments. Compared with previous SARP works, our error decomposition is also different, especially in the treatment of the empirical error term. We explicitly distinguish the regimes $s < 2$ and $s \geq 2$, and our analysis of the term $T_{n,\lambda}$ is carried out separately for three different types of source conditions, with tailored bounds in each case. These three cases and their corresponding bounds are new. At the same time, our overall decomposition is simpler and more transparent than that in Lin & Cevher (2020a), while still yielding optimal rates under more general assumptions. More detailed proof novelties can be found in Remarks 4 and 5 and around the proof of lemmas in Appendix E

We also handle a sharper error control of the operator similarity. By using the embedding condition, we refine the analysis of operator similarity (Lemmas 11–17) and obtain sharper error bounds in the final results. In particular, we derive tighter estimates of (i) the similarity between the covariance operator $C_K$ and the sample covariance operator $C_{K,n}$, leading to the condition on $n$ from $n \gtrsim \lambda^{-1}$ to $n \gtrsim \lambda^{-\alpha}$, and (ii) the sample-variance term, improving it roughly from $\mathcal{O}(n^{-1}\lambda^{\min\{-1/2, s/2-1\}} + n^{1/2}\lambda^{(s-1)/2})$ to $\mathcal{O}(n^{-1}\lambda^{\min\{-\alpha/2, s/2-\alpha\}} + n^{1/2}\lambda^{(s-\alpha)/2})$. Different from the analysis in Lin & Cevher (2020a), we further do not require the uniform boundedness of $f_\rho$.

## C    SOME DISCUSSIONS AND EXAMPLES

In this section, we present several examples of spectral algorithms, randomized sketches, and benign RKHSs with $\alpha = \frac{1}{\beta}$. We also provide some discussions on the general source condition and the interpolation spaces.

### C.1    EXAMPLES OF SPECTRAL ALGORITHMS

Table 3 lists some common spectral algorithms, along with their corresponding filter functions $g_\lambda(t)$, parameters, qualifications $\tau$, and $(E, F_\tau)$ pairs.

Table 3: Some common spectral algorithms, and their corresponding notations.

| Method | Filter Function $g_\lambda(t)$ | Parameters | Qualification $\tau$ | $(E, F_\tau)$ |
|---|---|---|---|---|
| **Tikhonov Regularization (KRR)** | $\frac{1}{t + \lambda}$ | $\lambda > 0$ | 1 | $(1, 1)$ |
| **Iterative Regularization** | $\sum_{k=1}^{l} \lambda^{k-1}(\lambda + t)^{-k}$ | $l \in \mathbb{N}$ | $l$ | $(l, 1)$ |
| **Gradient methods (GM)** | $\sum_{k=1}^{l} \eta(1 - \eta t)^{l-k}$ | $\eta \in [0, \kappa^2], \lambda = \frac{1}{\eta l}$ | any $\tau > 0$ | $(1, (\tau/e)^\tau)$ |
| **Spectral cutoff (SC)** | $\begin{cases} 1/t, & t \geq \lambda, \\ 0, & \text{otherwise} \end{cases}$ | $\lambda > 0$ | any $\tau > 0$ | $(1, 1)$ |

### C.2    EXAMPLES OF RANDOMIZED SKETCHES

**Sub-Gaussian random sketches.** A sketch matrix $\mathbf{G} \in \mathbb{R}^{m \times n}$ is called zero-mean 1-sub-Gaussian if its entries $\{g_{ij}\}_{i,j=1}^{m,n}$ are independent, mean-zero, sub-Gaussian random variables with variance proxy $1/m$; that is, there exists a universal constant $c > 0$ such that for all $t \in \mathbb{R}$, $\mathbb{E}\exp(tg_{ij}) \leq$

$\exp\left(c^2 t^2 / 2m\right)$. Typical examples include matrices with i.i.d. Gaussian entries or i.i.d. Rademacher (Bernoulli $\pm 1$) entries, scaled by $1/\sqrt{m}$. Closely related constructions draw independent rows uniformly from a rescaled sphere. Such rows are isotropic and sub-Gaussian and yield comparable sketching guarantees (Yang et al., 2017).

**Randomized orthogonal system (ROS) sketches.** A ROS sketch matrix $\mathbf{G} \in \mathbb{R}^{m \times n}$ is constructed as $\mathbf{G} = \sqrt{n/m}\mathbf{RHD}$, where $\mathbf{D} \in \mathbb{R}^{n \times n}$ is a diagonal matrix with i.i.d. Rademacher entries, $\mathbf{H} \in \mathbb{R}^{n \times n}$ is an orthonormal matrix (e.g., Hadamard or Fourier matrix), and $\mathbf{R} \in \mathbb{R}^{m \times n}$ is a random sampling matrix that selects $m$ rows uniformly at random from the $n \times n$ identity matrix. ROS sketches are computationally efficient due to the fast multiplication algorithms available for orthonormal matrices like Hadamard and Fourier transforms. Typically, a matrix-vector multiplication with a ROS sketch can be performed in $O(n \log m)$ time, which is significantly faster than the $O(mn)$ time required for dense random matrices (Ailon & Liberty, 2009).

**Sub-sampling sketches.** A sub-sampling sketch matrix $\mathbf{G} \in \mathbb{R}^{m \times n}$ is constructed by selecting $m$ rows uniformly at random without replacement from the $n \times n$ identity matrix. This means that each row of $\mathbf{G}$ has exactly one entry equal to $\sqrt{n/m}$, and all other entries are 0.

## C.3 EXAMPLES OF BEGNIN RKHSs WHEN $\alpha = 1/\beta$

This section presents examples of benign RKHSs with $\alpha = 1/\beta$ where the optimality can be extended to the full source range $s > 0$.

### C.3.1 RKHSs WITH UNIFORMLY BOUNDED EIGENFUNCTIONS

We first consider RKHSs whose eigenfunctions are uniformly bounded, that is,

$$\|e_k\|_\infty = \sup_{\boldsymbol{x} \in \mathcal{X}} |e_k(\boldsymbol{x})| \leq C, \quad \forall k \geq 1,$$

for some constant $C > 0$. According to Lemma 10 in Fischer & Steinwart (2020), such RKHSs satisfy Assumption 3 with $\alpha = 1/\beta$. The uniformly bounded eigenfunctions (UBE) assumption was also proposed in Williamson et al. (2002); Mendelson & Neeman (2010); Feng et al. (2023); Ma et al. (2023). We provide two examples in the following.

**Example 1. Translation-invariant kernels (Zhang et al., 2024)** Without loss of generality, consider translation-invariant kernels defined as $k(\boldsymbol{x}, \boldsymbol{x}') = g(x - x')$ on $[-1, 1] \times [-1, 1]$, where $g$ is a positive definite, even function on $\mathcal{X}$, satisfying that $g(t + 2k) = g(t)$ for $t \in [-1, 1]$ and $k \in \mathbb{Z}$. We further assume that $\rho_{\mathcal{X}}$ is the uniform distribution on $[-1, 1]$. The evenness of $g$ ensures the kernel is symmetric and the corresponding integral operator is a convolution operator, i.e.,

$$L_K f(x) = \int_{\mathcal{X}} k(x, x') f(x') dx' = \int_{\mathcal{X}} g(x - x') f(x') dx'.$$

The eigenfunctions of convolution operators are given by the Fourier basis, specifically

$$e_k(x) = \cos(\pi k x), \quad \forall k \geq 1,$$

which are clearly uniformly bounded. Consequently, the RKHSs induced by such translation-invariant kernels are benign with $\alpha = 1/\beta$. Multi-dimensional translation-invariant kernels can be constructed analogously; see Section 4.3 in Zhang et al. (2024).

**Example 2. First-order Sobolev RKHSs (Wainwright, 2019).** We consider the first-order Sobolev space defined as

$$\mathcal{H}^1([0,1]) = \left\{ f : [0,1] \to \mathbb{R} \mid f(0) = 0 \text{ and } f \text{ is absolutely continuous with } f' \in L^2_{\rho_{\mathcal{X}}}([0,1]) \right\}$$

equipped with the inner product $\langle f, g \rangle_{\mathcal{H}^1([0,1])} = \int_0^1 f'(x) g'(x) dx$, where we assume that $\rho_{\mathcal{X}}$ is the uniform distribution on $[-1, 1]$.. The associated kernel is $K(x, x') = \min\{x, x'\}$, with eigenfunction–eigenvalue pairs given by

$$e_k(x) = \sin\frac{(2k-1)\pi x}{2}, \quad \mu_k = \left(\frac{2}{(2k-1)\pi}\right)^2, \quad \forall k \geq 1.$$

These eigenfunctions are clearly uniformly bounded. Hence, first-order Sobolev RKHSs are benign with $\alpha = 1/\beta$.

Although there are some concrete examples satisfying the uniformly bounded eigenfunctions (UBE) condition, we want to emphasize that this condition is not mild and hard to check. Even for the Gaussian kernel on $[-1, 1]$, it is unknown whether UBE holds, though we expect that this is true. A counterexample of a $C^\infty$ Mercer kernel is provided in example 1 of Zhou (2002) and further mentioned in Minh et al. (2006); Steinwart & Scovel (2012).

### C.3.2 BESOV RKHSS

The second type of benign RKHSs is the Besov RKHSs. Here, we consider $\mathcal{X} \in \mathbb{R}^d$ is a non-empty open, connected, and bounded set with smooth boundary, and is equipped with the Lebesgue-Borel $\sigma$-algebra $\mathcal{B}$. For $l \geq 0$, we denote the Sobolev space $W^{l,2}(\mathcal{X})$ as $W^l(\mathcal{X})$, i.e.,

$$W^l(\mathcal{X}) = \left\{ f : \mathcal{X} \to \mathbb{R} \mid D^{\boldsymbol{a}} f \in L^2_{\rho_{\mathcal{X}}}, \forall |\boldsymbol{a}|_1 \leq l \right\},$$

where $\boldsymbol{a} = (a_1, \ldots, a_d)$ is a multi-index, $D^{\boldsymbol{a}} f = \frac{\partial^{|\boldsymbol{a}|_1} f}{\partial x_1^{a_1} \cdots \partial x_d^{a_d}}$ is the weak derivative of order $\boldsymbol{a}$, and $|\boldsymbol{a}|_1 = a_1 + \ldots + a_d$. For $t > 0$, the Besov space $B_{2,2}^t(\mathcal{X})$ is defined as the interpolation space between $L^2_{\rho_{\mathcal{X}}}$ and $W^l(\mathcal{X})$ with $l$ being the smallest integer greater than $t$, i.e.,

$$B_{2,2}^t(\mathcal{X}) = [L^2_{\rho_{\mathcal{X}}}, W^l(\mathcal{X})]_{\theta,2}$$

with $\theta = t/l$. The Besov $B_{2,2}^t(\mathcal{X})$ for some $t > d/2$ is a separate RKHS with a bounded kernel and the average capacity parameter $\beta = 2t/d$ (see Theorem 7.24 in (Adams & Fournier, 2003)). In the following, we assume that $\rho_{\mathcal{X}}$ is equivalent to the Lebesgue measure $\rho'$ on $\mathcal{X}$, $\rho_{\mathcal{X}} \ll \rho'$, $\rho' \ll \rho_{\mathcal{X}}$, and there exists constants $c, C > 0$ such that $\frac{d\rho_{\mathcal{X}}}{d\rho'} \in [c, C]$ is $\rho'$-almost surely satisfied.

For $t > j + d/2$ where $j \geq 0$, there holds that

$$B_{2,2}^t(\mathcal{X}) \hookrightarrow C_j(\mathcal{X})$$

where $C_j(\mathcal{X})$ is the Hölder space of $j$-times continuously differentiable functions on $\mathcal{X}$ (see, e.g, 7.57 in Adams & Fournier (2003)). And for $t > d/2$ and any $d/2t < \alpha_0 < 1$,

$$[B_{2,2}^t(\mathcal{X})]^{\alpha_0} = B_{2,2}^{t\alpha_0}(\mathcal{X}) \hookrightarrow C_0(\mathcal{X}) \hookrightarrow \mathcal{L}^\infty_{\rho_{\mathcal{X}}}.$$

Thus, Assumption 3 holds with $\alpha = \alpha_0$. Since $\alpha_0$ can be chosen arbitrarily close to $\beta = 2t/d$, we have the embedding condition in Assumption 3 holds with $\alpha = 1/\beta$. Therefore, Besov RKHSs are benign with $\alpha = 1/\beta$.

### C.4 DISCUSSIONS ON SOURCE CONDITION

In statistical learning theory, it is standard to restrict the function class of the underlying problem. A common approach is to consider a family of probability measures $\mathcal{P}(\Omega)$ whose associated regression function satisfies $f_\rho \in \Omega$, where $\Omega$ is a hypothesis set. Typically, $\Omega$ is compact and specified by regularity (smoothness) conditions. In the context of kernel methods, a widely used regularity condition is the Hölder source condition (Caponnetto & De Vito, 2007; Steinwart et al., 2009; Guo et al., 2017) in terms of the integral operator $L_K$:

$$f_\rho \in \Omega_{s,R} := \left\{ f \in L^2_{\rho_{\mathcal{X}}} : f = L_K^{s/2} g, \|g\|_\rho \leq R \right\}. \tag{16}$$

where $s > 0$ is the smoothness parameter. When $s = 1$, the condition is equivalent to assuming that $f \in \mathcal{H}_K$, which is independent of the probability measure $\rho_{\mathcal{X}}$. We denote the function class of $f_\rho$ as $\mathcal{F} = L_K^{s/2}(L^2_{\rho_{\mathcal{X}}})$, and note that $\mathcal{H}_K = L_K^{1/2}(L^2_{\rho_{\mathcal{X}}})$, and $L_K^{s_1/2}(L^2_{\rho_{\mathcal{X}}}) \subseteq L_K^{s_2/2}(L^2_{\rho_{\mathcal{X}}})$ (Steinwart & Christmann, 2008), if $s_1 \geq s_2$. When $s \geq 1$, the functional class $\mathcal{F}$ is a subset of the assumed RKHS $\mathcal{H}_K$, so we have $f_\rho \in \mathcal{H}_K$ (well-specified case). When $s \in (0, 1)$, the functional class $\mathcal{F}$ is larger than the assumed RKHS $\mathcal{H}_K$, and there exists some cases where $f_\rho \notin \mathcal{H}_K$ (mis-specified case).

The Hölder source condition characterizes the regularity of $f_\rho$ in terms of the decay of the coefficients in the eigenbasis of $L_K$. Specifically, if $\{(\mu_i, e_i)\}_{i \geq 1}$ are the eigenpairs of $L_K$, then $f_\rho$ can be

expressed as $f_\rho = \sum_{i=1}^\infty \mu_i \langle f_\rho, e_i \rangle_\rho e_i$. The Hölder source condition implies that the coefficients $\langle f_\rho, e_i \rangle_\rho$ decay at a rate determining by $r$, i.e.,

$$\sum_{i=1}^\infty \frac{\langle f_\rho, e_i \rangle_\rho^2}{\mu_i^s} < \infty,$$

which is known as Picard's criterion (Bauer et al., 2007). Therefore, it is natural to extend the measurement by some stronger conditions, i.e.,

$$\sum_{i=1}^\infty \frac{\langle f_\rho, e_i \rangle_\rho^2}{\phi^2(\mu_i)} < \infty,$$

where $\phi(\cdot)$ is an index function. Thus the general source condition can be defined as

$$f_\rho \in \Omega_{\phi,s,R} := \left\{ f \in L_{\rho_X}^2 : f = \phi(L_K)g, \|g\|_\rho \leq R, \phi \in C_s \right\}, \tag{17}$$

where $\mathcal{C}_s$ is the index function class. In this paper, we consider

$$\mathcal{C}_s = \Big\{ \phi : [0, \kappa^2] \to \mathbb{R}^+ \big| \phi \text{ is non-decreasing}, \phi(0) = 0, \phi(\kappa^2) < \infty,$$

$$\text{and } \frac{\phi^2}{t^s} \text{ is non-decreasing for some } s \in (0, 2\tau] \Big\}.$$

For any operator $A$ and projection operator $P$, we also assume that $P\phi(A)P = \phi(PAP)$.

The Hölder source condition is a special case of the general source condition with $\phi(t) = t^{s/2}$, and it satisfies $P\phi(A)P = \phi(PAP)$. To prove this, we only need to show that $P^a = P$ for any $a \in \mathbb{R}$. The proof when $a \in Z$ is trivial since $P^2 = P$. When $a \notin Z$, the identity $P^2 = P$ implies that the eigenvalues of $P$ are only 0 and 1. Then there exists an invertible matrix $B$ such that

$$P = SDS^{-1}, \quad D = \begin{pmatrix} I_k & 0 \\ 0 & 0 \end{pmatrix}$$

Hence $P^a = BD^aB^{-1} = BDB^{-1} = P$.

We further generalize the source condition by allowing the index function $\phi$ to be a composition of two index functions $\phi = \psi\nu$, where $\psi$ is non-decreasing and $\nu$ is Lipschitz continuous with constant 1 and operator monotone on $[0, \kappa^2]$ with $\nu(0) = 0$, and satisfies that that $P\vartheta(A)P = \vartheta(PAP)$ for any operator $A$ and projection operator $P$.

## C.5 Discussions on Capacity Condition in Assumption 2

Assumption 2 is the usual capacity condition, which is widely used in the literature on kernel methods (Lin et al., 2017; Guo et al., 2017). Let $\mu_k$ be the eigenvalues of $L_K$, then

$$\mathcal{N}(\lambda) = \sum_{k=1}^\infty \frac{\mu_k}{\mu_k + \lambda} \leq \frac{1}{\lambda} \sum_{k=1}^\infty \mu_k = \frac{\text{Tr}(L_K)}{\lambda},$$

so Assumption 2 always holds with $\beta = 1$ and $Q^2 = \text{Tr}(L_K)$.

Below we list several standard examples of RKHSs for which Assumption 2 holds, together with the corresponding values of $\beta$.

- **Finite rank RKHSs.** Suppose the kernel has finite rank $C_R$, i.e., the eigenvalues are non-zero only up to some index $C_R < \infty$. Examples include the linear kernel, polynomial kernels, approximate kernels with a fixed number of random features, and neural tangent kernels with finite network width. In this case

$$\mathcal{N}(\lambda) = \sum_{k=1}^\infty \frac{\mu_k}{\mu_k + \lambda} \leq C_R,$$

  so Assumption 2 holds, for instance, with $\beta = 1$ and $Q^2 = C_R$.

- **Finite smoothness RKHSs.** If the eigenvalues decay polynomially, $\mu_k \leq q_2 k^{-\beta}$ for some $\beta > 0$, then there exists a constant $C_\beta > 0$ such that

$$\mathcal{N}(\lambda) = \sum_{k=1}^\infty \frac{\mu_k}{\mu_k + \lambda} \leq \sum_{k=1}^\infty \frac{q_2}{q_2 + k^\beta \lambda} \leq \int_0^\infty \frac{q_2}{q_2 + t^\beta \lambda} dt$$

$$\leq \lambda^{-1/\beta} \int_0^\infty \frac{q_2}{q_2 + t^\beta} dt \leq \left(1 + \frac{q_2}{b-1}\right) \lambda^{-\frac{1}{\beta}},$$

so Assumption 2 holds with $Q^2 = 1 + \frac{q_2}{b-1}$. A classical example is the Sobolev RKHS $\mathcal{H}^r([0,1]^d)$ with smoothness parameter $r > d/2$, for which $\mu_k \asymp k^{-2r/d}$ and Assumption 2 holds with $\beta = 2r/d$.

- **Gaussian RKHSs.** If $K$ is a Gaussian kernel and the input domain is compact, then the eigenvalues decay exponentially fast, which is stronger than any polynomial decay. In this case $\mathcal{N}(\lambda)$ grows at most poly-logarithmically in $1/\lambda$, and therefore Assumption 2 holds for any $\beta \geq 1$.

In order to construct the lower bound, we need the upper and lower bounds on the eigenvalues, not just an upper bound on the effective dimension. Note that in Assumption 5, we assume that there exists some positive constant $0 < q_1 \leq q_2$ and $1 \leq \beta < \infty$ that

$$q_1 k^{-\beta} \leq \mu_k \leq q_2 k^{-\beta}, \quad \forall k \geq 1,$$

Similarly, we can also prove that $\mathcal{N}(\lambda) \gtrsim \lambda^{-\frac{1}{\beta}}$, so we have $\mathcal{N}(\lambda) \asymp \lambda^{-\frac{1}{\beta}}$. Thus Assumption 5 can be regarded as an enhanced version of Assumption 2.

## C.6 DISCUSSIONS ON BERNSTEIN CONDITION IN ASSUMPTION 1

The Bernstein condition in Assumption 1 imposes a light-tailed requirement on the noise term $\varepsilon = y - f_\rho(\boldsymbol{x})$. This condition is weaker than the commonly used boundedness assumption on the output $y$ (e.g., Smale & Zhou (2007); Yao et al. (2007); Steinwart et al. (2009); Pillaud-Vivien et al. (2018); Li et al. (2023a)), which requires $|y| \leq C$ almost surely for some constant $C > 0$. It is satisfied by many widely used noise distributions, such as Gaussian and sub-Gaussian distributions (Guo et al., 2017; Lin et al., 2020b). Moreover, it is weaker than imposing the Bernstein condition directly on the output $y$ (e.g., Lin & Cevher (2020a;b)), since the latter ensures that the regression function $f_\rho$ is uniformly bounded, which is not guaranteed under Assumption 1. Specifically, a simple calculation shows that

$$\|f_\rho\|_\infty = \sup_{\boldsymbol{x}} |f_\rho(\boldsymbol{x})| \leq \int_{\mathcal{Y}} |y| \, d\rho(y|\boldsymbol{x}) \leq \left(\int_{\mathcal{Y}} y^2 \, d\rho(y|\boldsymbol{x})\right)^{1/2} \leq C,$$

for some constant $C > 0$.

In this paper, we adopt the Bernstein condition on the noise term $\varepsilon$ to derive our results, which yields a more general and practically applicable framework. Theoretically, we establish minimax upper and lower bounds over a broader function class $\Omega_{\phi,s,R}$ rather than $[\mathcal{H}]_\rho^s \cap L_{\rho_{\mathcal{X}}}^\infty$ in Lin & Cevher (2020a). This extension is nontrivial for $s \leq \alpha$, since $[\mathcal{H}]_\rho^s \cap L_{\rho_{\mathcal{X}}}^\infty \subsetneq [\mathcal{H}]_\rho^s$ for $s \leq \alpha$ when $\phi(t) = t^{s/2}$ (Zhang et al., 2024). Following Zhang et al. (2023; 2024), we employ a truncation technique (Lemma 17) together with the $L^q$-embedding properties of the interpolation spaces (Lemma 4) to handle the potential unboundedness of $f_\rho$. Compared with Zhang et al. (2023; 2024), our analysis accommodates more general source conditions and extends to spectral algorithms with random projections.

We also note that the Bernstein condition on the noise term $\varepsilon$ has been adopted in various settings, including KRR (Caponnetto & De Vito, 2007), spectral algorithms (Rastogi & Sampath, 2017), and KRR with Nyström sub-sampling (Rudi et al., 2015). However, these works are restricted to the well-specified case with $s \geq 1$, and their error analysis do not need to address the potential unboundedness of $f_\rho$. In contrast, our study tackles the more challenging mis-specified case with $s \geq \alpha - 1/\beta$ and employs a more general source condition, which necessitates a careful treatment of the possible unboundedness of $f_\rho$ in our error analysis.

Recently, Mollenhauer et al. (2025) studied the performance of KRR in the presence of noise that exhibits a finite number of higher moments. By using the Fuk–Nagaev inequalities for Hilbert-space valued random variables, they derived the learning rates consisting of sub-gaussian and polynomial terms. Under certain conditions, these rates are minimax optimal against heavy-tailed noise. Our proofs currently rely on a Bernstein-type condition to obtain exponential concentration for the sample variance term. In principle, one could combine our SARP analysis with robust concentration results tailored to heavy-tailed noise, while keeping the rest of the argument unchanged. We therefore expect that our results can be extended to certain heavy-tailed noise settings, but working out the details would require a separate, careful study.

### C.7 DISCUSSIONS ON THE INTERPOLATION SPACE AND SOME LEMMAS

In this section, we introduce the definition and properties of the interpolation space and some useful lemmas. Our intention was not to present new technical contributions, but to collect auxiliary results that are used several times later, so that the proofs can be read in a self-contained way.

In mathematical analysis, interpolation spaces are a family of function spaces that lie between two given Banach spaces. They are used to study the properties of functions and operators that may not be easily analyzed in either of the original spaces. Interpolation spaces are particularly useful in the context of partial differential equations, harmonic analysis, and functional analysis (Bergh & Löfström, 2012; Bennett & Sharpley, 1988). We introduce the K-method of real interpolation, which is one of the most commonly used methods to construct interpolation spaces (Bennett & Sharpley, 1988; Lunardi, 2018).

**Definition 4** (K-functional and interpolation space). *Let $\mathcal{B}_1$ and $\mathcal{B}_2$ be two Banach spaces, for any $u \in \mathcal{B}_1 + \mathcal{B}_2$ and $t > 0$, we define the K-functional as*

$$\mathcal{K}(u, t, \mathcal{B}_1, \mathcal{B}_2) = \inf \left( \|u_1\|_{\mathcal{B}_1} + t\|u_2\|_{\mathcal{B}_2} : u = u_1 + u_2, u_1 \in \mathcal{B}_1, u_2 \in \mathcal{B}_2 \right),$$

*and for $\theta \in (0,1)$ and $q \in [1, \infty]$, we define the real interpolation space $(\mathcal{B}_1, \mathcal{B}_2)_{\theta,q}$ as*

$$(\mathcal{B}_1, \mathcal{B}_2)_{\theta,q} = \left\{ u \in \mathcal{B}_1 + \mathcal{B}_2 : \|u\|_{(\mathcal{B}_1,\mathcal{B}_2)_{\theta,q}} = \left( \int_0^\infty \left( t^{-\theta} \mathcal{K}(u, t, \mathcal{B}_1, \mathcal{B}_2) \right)^q \frac{dt}{t} \right)^{1/q} < \infty \right\},$$

*with norm $\| \cdot \|_{(\mathcal{B}_1,\mathcal{B}_2)_{\theta,q}}$, and the usual modification when $q = \infty$:*

$$\|u\|_{(\mathcal{B}_1,\mathcal{B}_2)_{\theta,\infty}} = \sup_{t>0} t^{-\theta} \mathcal{K}(u, t, \mathcal{B}_1, \mathcal{B}_2).$$

The following lemma shows some properties of the real interpolation space (Bennett & Sharpley, 1988).

**Lemma 1.** *Let $\mathcal{B}_1$ and $\mathcal{B}_2$ be two Banach spaces and $\mathcal{B}_2 \hookrightarrow \mathcal{B}_1$. Then for any $0 < \theta_1 \leq \theta_2 < 1$ and $1 \leq q_1 \leq q_2 \leq \infty$, there holds that*

- $\mathcal{B}_2 \hookrightarrow (\mathcal{B}_1, \mathcal{B}_2)_{\theta_1,q_1} \hookrightarrow \mathcal{B}_1$;

- $(\mathcal{B}_1, \mathcal{B}_2)_{\theta_2,q_1} \hookrightarrow (\mathcal{B}_1, \mathcal{B}_2)_{\theta_1,q_1}$;

- $(\mathcal{B}_1, \mathcal{B}_2)_{\theta_1,q_1} \hookrightarrow (\mathcal{B}_1, \mathcal{B}_2)_{\theta_1,q_2}$.

The proof of the above lemma can be found in Chapter 5 of Bennett & Sharpley (1988).

**Definition 5** (Lorentz space). *Let $(\mathcal{X}, \mathcal{A}, \rho_\mathcal{X})$ be a measure space and $f : \mathcal{X} \to \mathbb{R}$ be a measurable function. For $p \in (0, \infty)$ and $q \in [1, \infty]$, we define the Lorentz space $L_\rho^{p,q}(\mathcal{X})$ as*

$$L_\rho^{p,q}(\mathcal{X}) = \left\{ f : \|f\|_{L_\rho^{p,q}(\mathcal{X})} := \left( \int_0^\infty \left( t\rho_\mathcal{X} \{x : |f(x)|^p \geq t\} \right)^{q/p} \frac{dt}{t} \right)^{1/q} < \infty \right\},$$

*and when $q = \infty$, we define*

$$\|f\|_{L_\rho^{p,\infty}(\mathcal{X})} := \sup_{t>0} (t^p \rho_\mathcal{X} \{x : |f(x)| \geq t\}).$$

The Lorentz spaces, first introduced by Lorentz (1950), are generalizations of the more familiar Lebesgue spaces $L_\rho^p(\mathcal{X})$. The Lorentz norms provide tighter control over both qualities than the $L_\rho^p(\mathcal{X})$ norm, by exponentially rescaling the measure of the level sets of $f$ in both the range $p$ and the domain $q$. In particular, we have $L_\rho^{p,p}(\mathcal{X}) \cong L_\rho^p(\mathcal{X})$ and $L_\rho^{p,\infty}(\mathcal{X})$ is the weak $L^p$ space. The following lemma shows the relation between the Lorentz space and the real interpolation space (Bennett & Sharpley, 1988).

**Lemma 2.** *Let $(\mathcal{X}, \mathcal{A}, \rho_\mathcal{X})$ be a measure space. For $p \in (1, \infty)$ and $q \in [1, \infty]$, there holds that*

$$(L_\rho^1(\mathcal{X}), L_\rho^\infty(\mathcal{X}))_{1-1/p,q} \cong L_\rho^{p,q}(\mathcal{X}).$$

*Furthermore, for $1 < p_1 \neq p_2 < \infty$, $q \in [1, \infty]$ and $\theta \in (0, 1)$ satisfying that $\frac{1}{p_\theta} = \frac{1-\theta}{p_1} + \frac{\theta}{p_2}$, there holds that*

$$(L_\rho^{p_1}(\mathcal{X}), L_\rho^{p_2}(\mathcal{X}))_{\theta,q} \cong L_\rho^{p_\theta,q}(\mathcal{X}),$$

*where $\cong$ denotes the isomorphism between the two spaces, and the norm equivalence holds between the two spaces.*

The proof of the above lemma can be found in Theorem 1.3.5 of Lunardi (2018) and Lemma 4 and 29 of Zhang et al. (2024). The next lemma shows the relation between the interpolation space of RKHS and the real interpolation space, which is crucial for our analysis.

**Lemma 3.** *Suppose that $(\mathcal{X}, \mathcal{A}, \rho_\mathcal{X})$ is a measure space and $\mathcal{X}$ is a subset of $\mathbb{R}^d$. Let $K$ be a bounded kernel on $\mathcal{X}$ and $\mathcal{H}$ be the associated RKHS. For $\theta \in (0, 1)$, there holds that*

$$[\mathcal{H}]_\rho^\theta \cong (L_\rho^2(\mathcal{X}), \mathcal{H})_{\theta,2},$$

*where $\cong$ denotes the isomorphism between the two spaces, and the norm equivalence holds between the two spaces.*

The proof of the above lemma can be found in Theorem 4.6 of Steinwart & Scovel (2012). We also introduce the following Riesz-Thorin interpolation theorem, which is useful for bounding the operator norm of linear operators between two Banach spaces (Lunardi, 2018).

The following lemma introduces the $L_q$-embedding property of the interpolation space, which is established in Theorem 5 of Zhang et al. (2024). To be self-contained, we also give the proof.

**Lemma 4.** *Under Assumption 3, then for any $\theta \leq \alpha \leq 1$, there holds that*

$$[\mathcal{H}]_\rho^\theta \hookrightarrow L_\rho^{q_\theta}(\mathcal{X}), \quad with \quad q_\theta = \frac{2\alpha}{\alpha - \theta}.$$

*Proof.* From Lemma 3, for any $\alpha' > \alpha$, we have

$$[\mathcal{H}]_\rho^\theta \hookrightarrow [[\mathcal{H}]_\rho^{\alpha'}]^{\frac{\theta}{\alpha'}} \cong (L_\rho^2(\mathcal{X}), [\mathcal{H}]_\rho^{\alpha'})_{\frac{\theta}{\alpha'},2}.$$

Together with Assumption 3 that $[\mathcal{H}]_\rho^{\alpha'} \hookrightarrow [\mathcal{H}]_\rho^\alpha \hookrightarrow L_\rho^\infty(\mathcal{X})$ and Lemma 2, we have

$$[\mathcal{H}]_\rho^\theta \hookrightarrow (L_\rho^2(\mathcal{X}), L_\rho^\infty(\mathcal{X}))_{\frac{\theta}{\alpha'},2} \hookrightarrow (L_\rho^2(\mathcal{X}), L_\rho^C(\mathcal{X}))_{\frac{\theta}{\alpha'},2} \cong L_\rho^{q_\theta',2}(\mathcal{X})$$

where $0 < C < \infty$ and $\frac{1}{q_\theta'} = \frac{\alpha'-\theta}{2\alpha'} + \frac{\theta}{\alpha'C}$. Let $C$ is large enough such that $\frac{\alpha'-\theta}{2\alpha'} + \frac{\theta}{\alpha'C} \leq \frac{\alpha-\theta}{2\alpha}$, then $2 < q_\theta < q_\theta'$. By Lemma 1, we have

$$L_\rho^{q_\theta',2}(\mathcal{X}) \hookrightarrow L_\rho^{q_\theta',q_\theta'}(\mathcal{X}) \cong L_\rho^{q_\theta'}(\mathcal{X}) \hookrightarrow L_\rho^{q_\theta}(\mathcal{X}).$$

Thus we complete the proof.

$\square$

## D    LEARNING RATES FOR SARP IN EXPECTATION

In the main text, we present the high-probability bounds for the convergence issues of SARP in different settings by exponential tail inequalities such that

$$\mathbb{P}_{\mathbf{z}} \left[ \|f_{n,\lambda}^{rp} - f_\rho\|_{[\mathcal{H}]_\rho^\gamma} \leq \epsilon(n) \log \frac{C}{\delta} \right] \geq 1 - \delta$$

for $\delta \in (0,1)$ and some constant $C > 0$, and $\epsilon(n)$ is a positive and decreasing function depending on the sample size $n$. Based on the high-probability bounds, we can derive the corresponding bounds in expectation by integration of the tail inequalities:

$$\mathbb{E}_{\mathbf{z}}[\|f_{n,\lambda}^{rp} - f_\rho\|_{[\mathcal{H}]_\rho^\gamma}] = \int_0^\infty \mathbb{P}[\|f_{n,\lambda}^{rp} - f_\rho\|_{[\mathcal{H}]_\rho^\gamma} > t]dt$$

$$\leq \int_0^{\epsilon(n) \log C} 1 dt + \int_{\epsilon(n) \log C}^\infty \mathbb{P}[\|f_{n,\lambda}^{rp} - f_\rho\|_{[\mathcal{H}]_\rho^\gamma} > t]dt$$

$$\leq \epsilon(n) \log C + C \int_0^\infty \exp\left(-\frac{t}{\epsilon(n)}\right) dt$$

$$\leq \epsilon(n) + A\epsilon(n) \int_1^\infty e^{-u} du$$

$$\lesssim \epsilon(n).$$

Here $\mathbb{P}_{\mathbf{z}}$ and $\mathbb{E}_{\mathbf{z}}$ denote the probability and expectation with respect to the random sample $\mathbf{z} = \{(\boldsymbol{x}_i, y_i)\}_{i=1}^n$, respectively. The second inequality follows from the high-probability bounds, and the third inequality is due to the change of variable $u = t/\epsilon(n)$. Therefore, all the high-probability bounds presented in the main text can be easily extended to the corresponding bounds in expectation.

Theorem 4 presents the learning rates of SARP in expectation under general source conditions and random projection approaches. Corollary 3 presents the learning rates in expectation under Hölder source conditions and specific random projection approaches: randomized sketches, Plain and ALS Nyström sub-sampling.

**Theorem 4.** *Under the same assumptions and conditions as Theorem 3, there holds*

$$\mathbb{E}_{\mathbf{z}}[\|f_{n,\lambda}^{rp} - f_\rho\|_{[\mathcal{H}]_\rho^\gamma}^2] \lesssim (\phi(\Phi^{-1}(n^{-1})) + \Delta)^2 (\Phi^{-1}(n^{-1}))^{-\gamma},$$

*where the projection error $\Delta$ is defined as in Theorem 2.*

**Corollary 3.** *Under the same assumptions and conditions as Corollary 2, we consider the following three cases of $s$:*

*(1) When $s < \alpha - \frac{1}{\beta}$, let $\lambda = n^{-\frac{1}{\alpha}}$, $n \geq n_0$ for some $n_0 \geq 0$, and the projection dimension $m$ satisfies*

$$m \gtrsim \begin{cases} n^{\frac{1}{\alpha\beta}}, & \text{for Randomized Sketches and ALS Nyström}; \\ n, & \text{for Plain Nyström}. \end{cases}$$

*Then there holds*

$$\mathbb{E}_{\mathbf{z}}[\|f_{n,\lambda}^{rp} - f_\rho\|_{[\mathcal{H}]_\rho^\gamma}^2] \lesssim n^{-\frac{s-\gamma}{\alpha}}.$$

*(2) When $s \geq \alpha - \frac{1}{\beta}$ and $s < 2$, $n \geq n_0$ for some $n_0 \geq 0$, let $\lambda = n^{-\frac{\beta}{s\beta+1}}$, and the projection dimension $m$ satisfies*

$$m \gtrsim \begin{cases} n^{\frac{1}{s\beta+1}}, & \text{for Randomized Sketches and ALS Nyström}; \\ n^{\frac{\alpha\beta}{s\beta+1}}, & \text{for Plain Nyström}. \end{cases}$$

*Then there holds*

$$\mathbb{E}_{\mathbf{z}}[\|f_{n,\lambda}^{rp} - f_\rho\|_{[\mathcal{H}]_\rho^\gamma}^2] \lesssim n^{-\frac{(s-\gamma)\beta}{s\beta+1}}.$$

*(3) When $s \geq 2$, let $\lambda = n^{-\frac{\beta}{s\beta+1}}$, $n \geq n_0$ for some $n_0 \geq 0$, and the projection dimension $m$ satisfies*

$$m \gtrsim \begin{cases} n^{\frac{s-\gamma}{(2-\gamma)(s\beta+1)}}, & \text{for Randomized Sketches and ALS Nyström;} \\ n^{\frac{\alpha\beta(s-\gamma)}{(2-\gamma)(s\beta+1)}}, & \text{for Plain Nyström.} \end{cases}$$

*Then there holds*

$$\mathbb{E}_{\mathbf{z}}[\|f_{n,\lambda}^{rp} - f_\rho\|_{[\mathcal{H}]_\rho^\gamma}^2] \lesssim n^{-\frac{(s-\gamma)\beta}{s\beta+1}}.$$

# E  MAIN PROOFS

In this section, we present detailed proofs of the main results stated in the paper. We begin by establishing the minimax lower bound for spectral algorithms under general conditions in Section E.1. We then prove the learning rates stated in Theorems 2–3 and Corollary 2. Specifically, Section E.2 introduces the error decomposition, and Section E.3 bounds each component of the decomposition, namely the approximation error, projection error, and empirical error by using some operator representations. Section E.4 gives a refined analysis of these operator representations to obtain sharper error bounds. By combining the results from Sections E.2–E.4, we prove Theorems 2–3 and Corollary 2 in Sections E.5–E.7. Finally, Section E.8 provides auxiliary lemmas from the literature used throughout our analysis.

## E.1  PROOF OF THEOREM 1 AND COROLLARY 1

We employ standard tools from information theory (MacKay, 2003), including the Kullback–Leibler (KL) divergence and Fano's inequality, to establish the minimax lower bound under general conditions, including Assumptions 1-4.

**Remark 4.** *In the existing literature, minimax lower bounds are typically derived for the well-specified case where $f_\rho \in \mathcal{H}_K$ (Caponnetto & De Vito, 2007; Rastogi & Sampath, 2017). In this setting, the source condition is usually taken to satisfy $s \geq 1$. For the mis-specified case where $f_\rho \notin \mathcal{H}_K$, Fischer & Steinwart (2020); Zhang et al. (2023; 2024) provide the minimax lower bounds with $s > 0$. However, these studies all assume the Hölder source condition, which is a special case of the more general source condition considered in this paper. To the best of our knowledge, this is the first work to establish minimax lower bounds under general source conditions for both well-specified and mis-specified cases. Furthermore, our analysis does not rely on the uniform boundedness assumption on $f_\rho$.*

We begin by introducing several preliminary definitions and lemmas that will be used in the proof of Theorem 1.

**Definition 6** (Kullback-Leibler divergence)**.** *For two probability measures $P$ and $Q$ defined on the same measurable space $(\mathcal{X}, \mathcal{A})$, the Kullback-Leibler divergence is defined as*

$$K(P,Q) = \begin{cases} \int \log \frac{dP}{dQ} dP, & \text{if } P \ll Q, \\ +\infty, & \text{otherwise.} \end{cases}$$

*where $dP/dQ$ is the Radon-Nikodym derivative of $P$ with respect to $Q$.*

The following lemma describes the KL divergence between two specific probability measures considered in the main text.

**Lemma 5.** *Suppose $\rho_i, i = 1, 2$ are two joint distributions on $\mathcal{X} \times \mathcal{Y}$ with the same marginal distribution $\rho_{\mathcal{X}}$ on $\mathcal{X}$, and $\rho_1(y|\boldsymbol{x})$ and $\rho_2(y|\boldsymbol{x})$ are the conditional distributions of $y$ given $\boldsymbol{x}$ from the following models that*

$$y = f_i(\boldsymbol{x}) + \varepsilon, \quad i = 1, 2$$

*with $\varepsilon$ is a Gaussian noise with zero mean and variance $\sigma^2$, and $f_i \in L^2_{\rho_{\mathcal{X}}}$. Then the KL divergence between $\rho_1$ and $\rho_2$ can be expressed as*

$$KL(\rho_1, \rho_2) = \frac{\|f_1 - f_2\|_\rho^2}{2\sigma^2}.$$

*Moreover, if we consider the product measures $\rho_1^n$ and $\rho_2^n$ of $n$ independent distributions $\rho_1$ and $\rho_2$, then we have*

$$KL(\rho_1^n, \rho_2^n) = \frac{n\|f_1 - f_2\|_\rho^2}{2\sigma^2}.$$

The proof of Lemma 5 follows directly from the definition and chain rule of the KL divergence and is therefore omitted. The next lemma provides a bound on the packing numbers of sets of binary strings, which plays a key role in establishing the minimax lower bound.

**Lemma 6** (Proposition 6 in Caponnetto & De Vito (2007)). *For every $l \geq 16$, there exist $L \in \mathbb{N}$ and $\boldsymbol{\omega}_0, \boldsymbol{\omega}_1, \ldots, \boldsymbol{\omega}_L \in \{0,1\}^l$ such that*

$$\sum_{k=1}^{l} (\omega_i^k - \omega_j^k)^2 \geq l, \quad \forall 0 \leq i < j \leq L,$$

$$L \geq e^{l/24},$$

*where $\boldsymbol{\omega}_i = (\omega_i^1, \ldots, \omega_i^l)$ and $\boldsymbol{\omega}_j = (\omega_j^1, \ldots, \omega_j^l)$.*

**Lemma 7** (Theorem 2.5 in Tsybakov (2008)). *Consider a nonparametric class of functions $\Theta$ containing the function $\theta$ that we want to estimate, a family $\{P_\theta, \theta \in \Theta\}$ of probability measures indexed by $\Theta$ on a measurable space $(\mathcal{X}, \mathcal{A})$ associated with data, and a semi-distance $d$ on $\mathcal{F}$ used to define the risk. Assume that $L \geq 2$ and suppose that there exists a set distinct elements $\{\theta_0, \theta_1, \ldots, \theta_L\}$ such that*

*(i) $d(\theta_j, \theta_k) \geq 2\epsilon > 0, \ \forall 0 \leq j < k \leq L$;*

*(ii) $P_j \ll P_0, \quad \forall j = 1, \ldots, L$, and*

$$\frac{1}{L} \sum_{j=1}^{L} KL(P_j, P_0) \leq \zeta \log L$$

*with $0 < \zeta < 1/8$ and $P_j = P_{\theta_j}, j = 0, 1, \ldots, L$. Then*

$$\inf_{\hat{\theta}} \sup_{\theta \in \Theta} P_\theta(d(\hat{\theta}, \theta) \geq \epsilon) \geq \frac{\sqrt{L}}{1 + \sqrt{L}} \left(1 - 2\zeta - \sqrt{\frac{2\zeta}{\log L}}\right) > 0.$$

Now we are ready to prove Theorem 1.

**Proof** For given $\epsilon > 0$, we define

$$g = \sum_{k=l+1}^{2l} \sqrt{\frac{\epsilon}{l}} \frac{\omega^{k-l} e_k}{\Upsilon(\mu_k)},$$

where $\Upsilon(t) = \phi(t) t^{-\frac{\gamma}{2}}$, $\omega^k$ is the $k$-th coordinate of the binary string $\boldsymbol{\omega}$, and $e_k$ is the $k$-th eigenfunction of the integral operator $L_K$. Under the Assumption 5 that $\mu_k \geq q_1 k^{-\beta}$ and $\Upsilon(t) = \phi(t) t^{-\frac{s}{2}} \cdot t^{\frac{s-\gamma}{2}}$ is non-decreasing from Assumption 4 and $\gamma \leq s$, we have

$$\|g\|_\rho^2 = \sum_{k=l+1}^{2l} \frac{\epsilon}{l} \frac{(\omega^{k-l})^2}{\Upsilon^2(\mu_k)} \leq \sum_{k=1}^{l} \frac{\epsilon}{l} \frac{1}{\Upsilon^2(q_1 k^{-\beta})} \leq \frac{\epsilon}{\Upsilon^2(q_1 l^{-\beta})}.$$

By taking

$$l \leq \left(\frac{q_1}{\Upsilon^{-1}(\sqrt{\epsilon/R})}\right)^{1/\beta},$$

we can ensure that $\|g\|_\rho^2 \leq R$. For $l = l_\epsilon = \left\lfloor \left(\frac{q_1}{\Upsilon^{-1}(\sqrt{\epsilon/R})}\right)^{1/\beta} \right\rfloor$, choose $\epsilon_0$ such that $l_{\epsilon_0} \geq 16$. Then we can apply Lemma 6, for every positive $\epsilon < \epsilon_0$, so $l_\epsilon > l_{\epsilon_0}$, there exist $L_\epsilon \in \mathbb{N}$, and $\boldsymbol{\omega}_1, \ldots, \boldsymbol{\omega}_{L_\epsilon} \in \{0,1\}^{l_\epsilon}$ such that

$$\sum_{k=1}^{l_\epsilon} (\omega_i^k - \omega_j^k)^2 \geq l_\epsilon, \quad \forall 0 \leq i < j \leq L_\epsilon, \tag{18}$$

and

$$L_\epsilon \geq e^{l_\epsilon/24}. \tag{19}$$

Hence, we define the probability distribution $\rho_{f_i}$ on $\mathcal{X} \times \mathcal{Y}$ as follows:

$$y = f_i(\boldsymbol{x}) + \varepsilon, \quad \boldsymbol{x} \sim \rho_{\mathcal{X}}, \quad \varepsilon \sim N(0, \sigma^2), \quad i = 0, 1, \ldots, L_\epsilon,$$

where $f_i = \phi(L_K)g_i$ and for $\epsilon > 0$,

$$g_i = \sum_{k=l_\epsilon+1}^{2l_\epsilon} \sqrt{\frac{\epsilon}{l_\epsilon}} \frac{\omega_i^{k-l_\epsilon} e_k}{\Upsilon(\mu_k)}, \text{ for } i = 0, 1, \ldots, L_\epsilon, \tag{20}$$

where $\omega^k$ is the $k$-th coordinate of the binary string $\boldsymbol{\omega}$. Consequently, Assumption 4 holds. For Assumption 1, we can easily verify that from the property of moments of the Gaussian distribution when $\sigma \leq M$.

Now we check the conditions in Lemma 7. For (i), from (18) and (20), we have

$$\|f_i - f_j\|_{[\mathcal{H}]_\rho^\gamma}^2 \leq \|L_K^{-\frac{\gamma}{2}}(f_i - f_j)\|_\rho^2 = \frac{\epsilon}{l_\epsilon} \sum_{k=l_\epsilon+1}^{2l_\epsilon} (\omega_i^{k-l_\epsilon} - \omega_j^{k-l_\epsilon})^2 \geq \epsilon. \quad \forall 0 \leq i < j \leq L_\epsilon. \tag{21}$$

For (ii), according to Lemma 5, for any $i \in \{1, \ldots, L_\epsilon\}$, we have

$$KL(\rho_{f_i}^n, \rho_{f_0}^n) = \frac{n\|f_i - f_0\|_\rho^2}{2\sigma^2} = \frac{n\|\phi(L_K)(g_i - g_0)\|_\rho^2}{2\sigma^2} \leq \frac{n\epsilon}{2\sigma^2 l_\epsilon} \sum_{k=l_\epsilon+1}^{2l_\epsilon} \frac{q_2^{\frac{\gamma}{2}}(\omega_i^{k-l_\epsilon} - \omega_0^{k-l_\epsilon})^2}{k^{\frac{\beta\gamma}{2}}}$$

$$\leq \frac{2nq_2^{\frac{\gamma}{2}}\epsilon}{\sigma^2 l_\epsilon} \sum_{k=l_\epsilon+1}^{2l_\epsilon} \frac{1}{k^{\frac{\beta\gamma}{2}}} \leq \frac{2nq_2^{\frac{\gamma}{2}}\epsilon}{\sigma^2 l_\epsilon} \int_{l_\epsilon+1}^{2l_\epsilon} \frac{1}{x^{\frac{\beta\gamma}{2}}} dx = c_k \frac{n\epsilon}{l_\epsilon^{\frac{\beta\gamma}{2}}},$$

where $c_k = \frac{2q_2^{\frac{\gamma}{2}}}{\sigma^2(\frac{\beta\gamma}{2}-1)}(1 - 2^{1-\frac{\beta\gamma}{2}})$. For given $\epsilon > 0$ and $\zeta \in (0, 1/8)$, we can take $q_2$ small enough such that

$$c_k \frac{n\epsilon}{l_\epsilon^{\frac{\beta\gamma}{2}}} \leq \frac{\zeta l_\epsilon}{24} \leq \zeta \log L_\epsilon,$$

where the last inequality is from (19), then we can ensure that (ii) in Lemma 7 holds.

Finally, we apply Lemma 7 with $d(f,g) = \|f - g\|_{[\mathcal{H}]_\rho^\gamma}$, and the minimax lower bound follows that

$$\inf_{\widehat{f}} \sup_{f_\rho \in \Omega_{\phi,s,R}} \mathbb{P}_{\rho_f} \left( \|\widehat{f} - f_\rho\|_{[\mathcal{H}]_\rho^\gamma}^2 \geq \frac{\epsilon}{2} \right) \geq \frac{\sqrt{L_\epsilon}}{1 + \sqrt{L_\epsilon}} \left( 1 - 2\zeta - \sqrt{\frac{2\zeta}{\log L_\epsilon}} \right), \tag{22}$$

where

$$L_\epsilon = \exp\left\{ \frac{1}{24} \left\lfloor \left( \frac{q_1}{\Upsilon^{-1}(\sqrt{\epsilon/R})} \right)^{1/\beta} \right\rfloor \right\}.$$

For $\zeta \in (0, 1/8)$, let $\epsilon_n = 2R\zeta\phi^2(\Phi^{-1}(n^{-1}))(\Phi^{-1}(n^{-1}))^{-\gamma} = 2R\zeta\Upsilon^2(\Phi^{-1}(n^{-1}))$, then

$$L_{\epsilon_n} = \exp\left\{ \frac{1}{24} \left\lfloor \left( \frac{q_1}{\Upsilon^{-1}(\sqrt{2\zeta}\Upsilon(\Phi^{-1}(n^{-1})))} \right)^{1/\beta} \right\rfloor \right\}$$

$$> \exp\left\{ \frac{1}{24} \left\lfloor \left( \frac{q_1}{\Phi^{-1}(n^{-1})} \right)^{1/\beta} \right\rfloor \right\},$$

where the last inequality is from the fact that $\Upsilon^{-1}(\cdot)$ is non-decreasing and $\sqrt{2\zeta} \in (0, 1/2)$.

Note that $\Phi(t) = (\phi(t)/\phi(1))^2 t^{1/\beta}$ is non-decreasing and $\Phi(0) = 0$, thus when $n$ is sufficiently large, $L_{\epsilon_n}$ can be sufficiently large. Consequently, we can obtain that the right side of (22) is larger than $1 - 3\zeta$. Let $\delta = 3\zeta \in (0, 3/8)$, there holds that

$$\inf_{\widehat{f}} \sup_{f_\rho \in \Omega_{\phi,s,R}} \mathbb{P}_{\rho_f} \left( \|\widehat{f} - f_\rho\|_{[\mathcal{H}]_\rho^\gamma}^2 \geq \frac{R\delta}{3}\phi^2(\Phi^{-1}(n^{-1}))(\Phi^{-1}(n^{-1}))^{-\gamma} \right) \geq 1 - \delta.$$

Thus we complete the proof of Theorem 1.

The proof of Corollary 1 follows directly from Theorem 1 by taking $\phi(t) = t^{\frac{s}{2}}$.

## E.2 ERROR DECOMPOSITION

In this section, we present the error decomposition for the estimator defined in (14). Define the following data-free and noise-free intermediate estimator $f_\lambda$

$$f_\lambda = g_\lambda(C_K)S_K^* f_\rho \tag{23}$$

which is the optimal approximation of $f_\rho$ in the RKHS $\mathcal{H}_K$. Using the triangle inequality, we can decompose the error $\|f_{n,\lambda}^{rp} - f_\rho\|_{[\mathcal{H}]_\rho^\gamma}$ into three parts: the empirical error $\|f_{n,\lambda}^{rp} - Pf_\lambda\|_{[\mathcal{H}]_\rho^\gamma}$, the random projections error $\|(I-P)f_\lambda\|_{[\mathcal{H}]_\rho^\gamma}$, and the approximation error $\|f_\lambda - f_\rho\|_{[\mathcal{H}]_\rho^\gamma}$. The following proposition presents the detailed error decomposition.

**Proposition 1.** *Let $f_{n,\lambda}^{rp}$ and $f_\lambda$ defined in (14) and (23), then for $\gamma \in [0, s]$, we have the following error decomposition:*

$$\|f_{n,\lambda}^{rp} - f_\rho\|_{[\mathcal{H}]_\rho^\gamma} \leq \underbrace{\|f_{n,\lambda}^{rp} - Pf_\lambda\|_{[\mathcal{H}]_\rho^\gamma}}_{\text{Empirical Error}} + \underbrace{\|(I-P)f_\lambda\|_{[\mathcal{H}]_\rho^\gamma}}_{\text{Random Projections Error}} + \underbrace{\|f_\lambda - f_\rho\|_{[\mathcal{H}]_\rho^\gamma}}_{\text{Approximation Error}}. \tag{24}$$

## E.3 BOUNDING THE ERROR TERMS

In this section, we bound each term in the error decomposition (24) by using some operator representations. We begin by bounding the approximation error in Section E.3.1, followed by the projection error in Section E.3.2, and finally the empirical error in Section E.3.3.

### E.3.1 BOUNDING THE APPROXIMATION ERROR

Define $r_\lambda(t) = 1 - tg_\lambda(t)$, and we have the following lemma using the properties of $g_\lambda(t)$. We assume that $s \in (0, 2\tau]$ throughout.

**Lemma 8.** *Under Assumption 4, then for any $\gamma \in [0, s]$, there holds*

$$\sup_{t \in [0, \kappa^2]} |r_\lambda(t)|\phi(t)t^{-\frac{\gamma}{2}} \leq c_\tau \phi(\lambda)\lambda^{-\frac{\gamma}{2}}, \quad \forall \lambda \in [0, \kappa^2],$$

*where $c_\tau = \frac{F_\tau}{c \wedge 1}$.*

*Proof.* For any $t \in [0, \kappa^2]$ and $\lambda \in (0, \kappa^2]$, we consider two cases. If $t \leq \lambda$, then using the fact that $\phi(t)t^{-s/2}$ is non-decreasing and the property of filter function in (13) such that $|r_\lambda(t)|t^{(s-\gamma)/2} \leq \lambda^{(s-\gamma)/2}$, we have

$$|r_\lambda(t)|\phi(t)t^{-\gamma/2} = |r_\lambda(t)|t^{(s-\gamma)/2}\phi(t)t^{-s/2} \leq F_\tau \lambda^{(s-\gamma)/2}\phi(\lambda)\lambda^{-s/2} = F_\tau \phi(\lambda)\lambda^{-\gamma/2}.$$

If $t > \lambda$, then using Assumption 4 and the property of filter function in (13) such that $|r_\lambda(t)|t^{\tau-\gamma/2} \leq \lambda^{\tau-\gamma/2}$, we have

$$|r_\lambda(t)|\phi(t)t^{-\gamma/2} = |r_\lambda(t)|t^{\tau-\gamma/2}\phi(t)t^{-\tau} \leq |r_\lambda(t)|t^{\tau-\gamma/2}c^{-1}\phi(\lambda)\lambda^{-\tau} \leq F_\tau c^{-1}\phi(\lambda)\lambda^{-\gamma/2}.$$

Combining the above two cases, we obtain the final results, thus we complete the proof. $\square$

**Proposition 2.** *Under Assumption 4, then for any $\gamma \in [0, s]$ and $\lambda \in [0, \kappa^2]$, there holds*

$$\|f_\lambda - f_\rho\|_{[\mathcal{H}]_\rho^\gamma} \leq c_\tau R\phi(\lambda)\lambda^{-\frac{\gamma}{2}}.$$

*Proof.* Using the definition of $f_\lambda$ in (23), we have

$$S_K f_\lambda - f_\rho = S_K g_\lambda(C_K)S_K^* f_\rho - f_\rho = S_K g_\lambda(S_K^* S_K)S_K^* f_\rho - f_\rho$$
$$= S_K S_K^* g_\lambda(S_K S_K^*)f_\rho - f_\rho = (L_K g_\lambda(L_K) - I)f_\rho,$$

where the third equality holds by Lemma 27. By the definition of $r_\lambda(t)$ and Assumption 4 that $f_\rho = \phi(L_K)g$, we have

$$\|f_\lambda - f_\rho\|_{[\mathcal{H}]_\rho^\gamma} = \|L_K^{-\frac{\gamma}{2}}(f_\lambda - f_\rho)\|_\rho = \|L_K^{-\frac{\gamma}{2}}(S_K f_\lambda - f_\rho)\|_\rho = \|L_K^{-\frac{\gamma}{2}}r_\lambda(L_K)\phi(L_K)g\|_\rho$$
$$\leq \|L_K^{-\frac{\gamma}{2}}r_\lambda(L_K)\phi(L_K)\|R \leq \sup_{t \in [0, \kappa^2]}|r_\lambda(t)|\phi(t)t^{-\frac{\gamma}{2}}R \leq c_\tau R\phi(\lambda)\lambda^{-\frac{\gamma}{2}},$$

where the second inequality holds by the fact that $\|L_K\| \leq \kappa^2$ and Lemma 8. Thus we complete the proof.

$\square$

**Lemma 9.** *Under Assumption 4, then for any $\gamma \geq -s$ and $\lambda \in (0, \kappa^2]$, there holds*

$$\|C_K^{\frac{\gamma-1}{2}} f_\lambda\|_{\mathcal{H}} \leq \begin{cases} E\lambda^{\frac{\gamma+s-2}{2}}\phi(\kappa^2)\kappa^{-s}, & \text{if } -s \leq \gamma < 2 - s; \\ ER\phi(\kappa^2)\kappa^{\gamma-2}, & \text{if } \gamma \geq 2 - s. \end{cases}$$

*Proof.* From the definition of $f_\lambda$ and Assumption 4, we have

$$\|C_K^{\frac{\gamma-1}{2}} f_\lambda\|_{\mathcal{H}} = \|C_K^{\frac{\gamma-1}{2}} g_\lambda(C_K)S_K^*\phi(L_K)g\|_{\mathcal{H}} \leq \|C_K^{\frac{\gamma-1}{2}} g_\lambda(C_K)S_K^*\phi(L_K)\|R$$

According to the spectral theorem and the fact that $C_K = S_K^* S_K$ and $L_K = S_K S_K^*$, we have

$$\|C_K^{\frac{\gamma-1}{2}} g_\lambda(C_K)S_K^*\phi(L_K)\| = \sqrt{\|\phi(L_K)S_K g_\lambda(C_K)C_K^{\gamma-1} g_\lambda(C_K)S_K^*\phi(L_K)\|}$$

$$= \|g_\lambda(L_K)L_K^{\frac{\gamma}{2}}\phi(L_K)\| \leq \sup_{t\in[0,\kappa^2]} |g_\lambda(t)|\phi(t)t^{\frac{\gamma}{2}}.$$

From Assumption 4, we know $\phi(t)$ and $\phi(t)t^{-s/2}$ are non-decreasing functions and non-negative on $(0, \kappa^2]$. Thus, for any $s' \in [0, s]$, we have $\phi(t)t^{-s'/2}$ is also a non-decreasing function on $(0, \kappa^2]$. Thus if $\gamma \geq 2 - s$, we have

$$\sup_{t\in[0,\kappa^2]} |g_\lambda(t)|\phi(t)t^{\frac{\gamma}{2}} \leq \sup_{t\in[0,\kappa^2]} |g_\lambda(t)|t\phi(t)t^{\frac{\gamma}{2}-1} \leq E\phi(\kappa^2)\kappa^{\gamma-2}.$$

If $-s \leq \gamma < 2 - s$, we have

$$\sup_{t\in[0,\kappa^2]} |g_\lambda(t)|\phi(t)t^{\frac{\gamma}{2}} = \sup_{t\in[0,\kappa^2]} |g_\lambda(t)|t^{\frac{\gamma+s}{2}}\phi(t)t^{-\frac{s}{2}} \leq E\lambda^{\frac{\gamma+s-2}{2}}\phi(\kappa^2)\kappa^{-s}.$$

Combining the above two cases, we complete the proof. $\square$

**Lemma 10.** *Under Assumptions 3 and 4, there holds that*

$$\|f_\lambda\|_\infty \leq \begin{cases} EA\lambda^{\frac{s-\alpha}{2}}\phi(\kappa^2)\kappa^{-s}, & \text{if } s < \alpha; \\ EA\phi(\kappa^2)\kappa^{-\alpha}, & \text{if } s \geq \alpha. \end{cases}$$

*Proof.* According to Assumption 3 that $\|[\mathcal{H}]_\rho^\alpha \hookrightarrow L_\rho^\infty(\mathcal{X})\| \leq A$, then we have

$$\|f_\lambda\|_\infty \leq A\|f_\lambda\|_{[\mathcal{H}]_\rho^\alpha} = A\|L_K^{-\frac{\alpha}{2}}S_K f_\lambda\|_\rho$$

Now we only need to bound $\|L_K^{-\alpha/2}S_K f_\lambda\|_{\mathcal{H}}$. From Assumption 4 that $f_\rho = \phi(L_K)g$, we have

$$\|L_K^{-\alpha/2}S_K f_\lambda\|_{\mathcal{H}} = \|L_K^{-\alpha/2}S_K g_\lambda(C_K)S_K^* f_\rho\|_{\mathcal{H}} = \|L_K^{-\alpha/2}S_K g_\lambda(S_K^* S_K)S_K^*\phi(L_K)g\|_{\mathcal{H}}$$

$$= \|L_K^{-\alpha/2} g_\lambda(S_K S_K^*)S_K S_K^*\phi(L_K)g\|_{\mathcal{H}} = \|L_K^{-\alpha/2} g_\lambda(L_K)L_K\phi(L_K)g\|_{\mathcal{H}}$$

$$\leq \|L_K^{-\alpha/2} g_\lambda(L_K)L_K\phi(L_K)\|R = \|L_K^{1-\alpha/2} g_\lambda(L_K)L_K\phi(L_K)\|.$$

For $s \geq \alpha$, note that Assumption 4 indicates $\phi(t)$ and $\phi(t)t^{-s/2}$ are positive and non-decreasing in $[0, \kappa^2]$, thus $\phi(t)t^{-\alpha/2} = \phi(t)t^{-s/2}t^{(s-\alpha)/2}$ is also non-decreasing, so we can obtain that

$$\sup_{t\in[0,\kappa^2]} |g_\lambda(t)t^{1-\alpha/2}\phi(t)| = \sup_{t\in[0,\kappa^2]} |g_\lambda(t)t\phi(t)t^{-\alpha/2}| \leq E\phi(\kappa^2)\kappa^{-\alpha},$$

where we use the property of filter function $g_\lambda(t)$ in (13) with $\nu = 1$.

For $s < \alpha$, similarly we have

$$\sup_{t\in[0,\kappa^2]} |g_\lambda(t)t^{1-\alpha/2}\phi(t)| = \sup_{t\in[0,\kappa^2]} |g_\lambda(t)t^{s/2+1-\alpha/2}\phi(t)t^{-s/2}| \leq E\lambda^{\frac{s-\alpha}{2}}\phi(\kappa^2)\kappa^{-s}.$$

Thus we complete the proof.

$\square$

### E.3.2 Bounding the Random Projections Error

**Proposition 3.** *Under Assumption 4, then for any $\gamma \in [0, s \wedge 1]$ and $\lambda \in [0, \kappa^2]$, there holds*

$$\|(I - P)f_\lambda\|_{[\mathcal{H}]_\rho^\gamma} \leq \begin{cases} \mathcal{A}_{K,P}^{2-\gamma} ER\phi(\kappa^2)\kappa^{-s}\lambda^{\frac{s}{2}-1}, & \text{if } s < 2; \\ \mathcal{A}_{K,P}^{2-\gamma} ER\phi(\kappa^2)\kappa^{-2}, & \text{if } s \geq 2. \end{cases}$$

*Proof.* Since $P$ is an orthogonal projection operator, we have $(I - P)^2 = I - P$. Thus, we can write

$$\|(I - P)f_\lambda\|_{[\mathcal{H}]_\rho^\gamma} = \|L_K^{-\frac{\gamma}{2}}(I - P)f_\lambda\|_\rho = \|L_K^{-\frac{\gamma}{2}} S_K (I - P)f_\lambda\|_\rho$$

$$\leq \|L_K^{-\frac{\gamma}{2}} S_K C_K^{\frac{\gamma-1}{2}}\| \|C_K^{\frac{1-\gamma}{2}}(I - P)f_\lambda\|_\mathcal{H}$$

$$\leq \|C_K^{\frac{1-\gamma}{2}}(I - P)^{1-\gamma}(I - P)f_\lambda\|_\mathcal{H}$$

$$\leq \|C_K^{\frac{1-\gamma}{2}}(I - P)^{1-\gamma}\| \|(I - P)C_K^{\frac{1}{2}}\| \|C_K^{-\frac{1}{2}}f_\lambda\|_\mathcal{H}$$

$$\leq \|C_K^{1/2}(I - P)\|^{1-\gamma} \|(I - P)C_K^{\frac{1}{2}}\| ER\phi(\kappa^2)\kappa^{-(s\wedge 2)}\lambda^{-\left(1-\frac{s}{2}\right)_+}$$

$$= \mathcal{A}_{K,P}^{2-\gamma} ER\phi(\kappa^2)\kappa^{-(s\wedge 2)}\lambda^{-\left(1-\frac{s}{2}\right)_+},$$

where the third inequality holds since $\|L_K^{-\frac{\gamma}{2}} S_K C_K^{\frac{\gamma-1}{2}}\| \leq 1$ by the spectral theorem, and the last inequality holds by Lemma 9 and the Cordes inequality in Lemma 24 for $\gamma \in [0, 1]$. Thus we complete the proof.

$\square$

### E.3.3 Bounding the Empirical Error

Denote $C_{K,\lambda} = C_K + \lambda I$ and $C_{K,n,\lambda} = C_{K,n} + \lambda I$, $O_{K,n} = PC_{K,n}P$ and $O_{K,n,\lambda} = O_{K,n} + \lambda I$ for simplicity throughout the proof.

$$\mathcal{P}_{n,\lambda} = \left\| C_{K,\lambda}^{-1/2} \left[ (S_{K,n}^* \mathbf{y} - C_{k,n}f_\lambda) - (S_K^* f_\rho - C_K f_\lambda) \right] \right\|_\mathcal{H},$$

$$\mathcal{Q}_{n,\lambda} = \|C_{K,\lambda}^{1/2} C_{K,n,\lambda}^{-1/2}\|,$$

$$\mathcal{R}_{n,\lambda} = \|C_{K,\lambda}^{-1/2}(C_K - C_{K,n})C_{K,\lambda}^{-1/2}\|,$$

$$\mathcal{S}_{n,\lambda} = \|C_{K,\lambda}^{-1/2}(C_K - C_{K,n})\|,$$

$$\mathcal{U}_{n,\lambda} = \|C_K - C_{K,n}\|,$$

$$\mathcal{T}_{n,\lambda} = \|O_{K,n,\lambda}^{1/2} r_\lambda(O_{K,n}) P f_\lambda\|_\mathcal{H}.$$

**Proposition 4.** *For any $\gamma \in [0, s \wedge 1]$ and $\lambda \in [0, \kappa^2]$, there holds*

$$\|C_{K,n,\lambda}^{1/2}(f_{n,\lambda}^{rp} - Pf_\lambda)\|_\mathcal{H}$$

$$\leq \begin{cases} \lambda^{-\frac{\gamma}{2}} \mathcal{Q}_{n,\lambda}^{1-\gamma} \left[ \mathcal{Q}_{n,\lambda}(\mathcal{P}_{n,\lambda} + c_\tau R\phi(\lambda)) + C\lambda^{\frac{s}{2}-1} \mathcal{Q}_{n,\lambda}(\mathcal{A}_{K,P}^2 + \lambda^{1/2}\mathcal{A}_{K,P}) + \mathcal{T}_{n,\lambda} \right], & \text{if } s < 2; \\ \lambda^{-\frac{\gamma}{2}} \mathcal{Q}_{n,\lambda}^{1-\gamma} \left[ \mathcal{Q}_{n,\lambda}(\mathcal{P}_{n,\lambda} + c_\tau R\phi(\lambda)) + C(\mathcal{U}_{n,\lambda}^{1/2}\mathcal{A}_{K,P} + \mathcal{A}_{K,P}^2) + \mathcal{T}_{n,\lambda} \right], & \text{if } s \geq 2. \end{cases}$$

*where* $C = \begin{cases} ER\phi(\kappa^2)\kappa^{-2}, & \text{if } s < 2; \\ ER\phi(\kappa^2)\kappa^{-s}, & \text{if } s \geq 2. \end{cases}$

*Proof.*

$$\|f_{n,\lambda}^{rp} - Pf_\lambda\|_{[\mathcal{H}]_\rho^\gamma} = \|L_K^{-\frac{\gamma}{2}}(f_{n,\lambda}^{rp} - Pf_\lambda)\|_\rho$$

$$\leq \|L_K^{-\frac{\gamma}{2}} S_K C_K^{\frac{\gamma-1}{2}}\| \|C_K^{\frac{1-\gamma}{2}} C_{K,\lambda}^{\frac{\gamma-1}{2}}\| \|C_{K,\lambda}^{\frac{1-\gamma}{2}} C_{K,n,\lambda}^{\frac{\gamma-1}{2}}\| \|C_{K,n,\lambda}^{\frac{1-\gamma}{2}}(f_{n,\lambda}^{rp} - Pf_\lambda)\|_\mathcal{H}$$

$$\leq \|C_K C_{K,\lambda}^{-1}\|^{\frac{1-\gamma}{2}} \|C_{K,\lambda}^{1/2} C_{K,n,\lambda}^{-1/2}\|^{1-\gamma} \|C_{K,n,\lambda}^{\frac{1-\gamma}{2}}(f_{n,\lambda}^{rp} - Pf_\lambda)\|_\mathcal{H} \qquad (25)$$

$$\leq \mathcal{Q}_{n,\lambda}^{1-\gamma} \|C_{K,n,\lambda}^{\frac{1-\gamma}{2}}(f_{n,\lambda}^{rp} - Pf_\lambda)\|_\mathcal{H}$$

$$\leq \lambda^{-\frac{\gamma}{2}} \mathcal{Q}_{n,\lambda}^{1-\gamma} \|C_{K,n,\lambda}^{1/2}(f_{n,\lambda}^{rp} - Pf_\lambda)\|_\mathcal{H},$$

where the second inequality holds by the inequality that $\|L_K^{-\frac{\gamma}{2}} S_K C_K^{\frac{\gamma-1}{2}}\| \le 1$, and Cordes inequality in Lemma 24 for $\gamma \in [0, 1]$, the third and last inequalities hold from $\|C_K C_{K,\lambda}^{-1}\| \le 1$ and $\|C_{K,n,\lambda}^{-\frac{\gamma}{2}}\| \le \lambda^{-\frac{\gamma}{2}}$.

Now we bound $\|C_{K,n,\lambda}^{1/2}(f_{n,\lambda}^{rp} - Pf_\lambda)\|_{\mathcal{H}}$. From the definition of $f_{n,\lambda}^{rp}$ in (14) and $P^2 = P$, we have

$$\|C_{K,n,\lambda}^{1/2}(f_{n,\lambda}^{rp} - Pf_\lambda)\|_{\mathcal{H}} = \|C_{K,n,\lambda}^{1/2}(Pg_\lambda(PC_{K,n}P)PS_{K,n}^*\mathbf{y} - Pf_\lambda)\|_{\mathcal{H}}$$
$$= \|C_{K,n,\lambda}^{1/2}P(g_\lambda(PC_{K,n}P)PS_{K,n}^*\mathbf{y} - Pf_\lambda)\|_{\mathcal{H}}. \tag{26}$$

Note that for any $f \in \mathcal{H}_K$, there holds that

$$\|C_{K,n,\lambda}^{1/2}Pf\|_{\mathcal{H}}^2 = \langle PC_{K,n,\lambda}Pf, f\rangle_{\mathcal{H}} = \langle PC_{K,n}Pf, f\rangle_{\mathcal{H}} + \lambda\langle Pf, f\rangle_{\mathcal{H}}$$
$$\le \|PC_{K,n}Pf\|_{\mathcal{H}}\|f\|_{\mathcal{H}} + \lambda\langle If, f\rangle_{\mathcal{H}} \tag{27}$$
$$= \langle (PC_{K,n}P + \lambda I)f, f\rangle_{\mathcal{H}} = \|(PC_{K,n}P + \lambda I)^{1/2}f\|_{\mathcal{H}}^2,$$

where the inequality holds by the fact that $I - P$ is an orthogonal projection operator, thus $\langle (I - P)f, f\rangle_{\mathcal{H}} = \|(I - P)f\|_{\mathcal{H}}^2 \ge 0$.

Note that $O_{K,n} = PC_{K,n}P$ and $O_{K,n,\lambda} = O_{K,n} + \lambda I$, and plugging (27) into (26), we have

$$\|C_{K,n,\lambda}^{1/2}(f_{n,\lambda}^{rp} - Pf_\lambda)\|_{\mathcal{H}} \le \|O_{K,n,\lambda}^{1/2}g_\lambda(O_{K,n})PS_{K,n}^*\mathbf{y} - Pf_\lambda)\|_{\mathcal{H}}$$
$$= \|O_{K,n,\lambda}^{1/2}[g_\lambda(O_{K,n})PS_{K,n}^*\mathbf{y} - (r_\lambda(O_{K,n}) + g_\lambda(O_{K,n})O_{K,n})Pf_\lambda]\|_{\mathcal{H}}$$
$$= \|O_{K,n,\lambda}^{1/2}[g_\lambda(O_{K,n})P(S_{K,n}^*\mathbf{y} - C_{K,n}Pf_\lambda) - r_\lambda(O_{K,n})Pf_\lambda]\|_{\mathcal{H}}$$
$$\le \|O_{K,n,\lambda}^{1/2}g_\lambda(O_{K,n})P(S_{K,n}^*\mathbf{y} - C_{K,n}Pf_\lambda)\|_{\mathcal{H}} + \|O_{K,n,\lambda}^{1/2}r_\lambda(O_{K,n})Pf_\lambda\|_{\mathcal{H}}$$
$$\le \|O_{K,n,\lambda}^{1/2}g_\lambda(O_{K,n})PC_{K,n,\lambda}^{1/2}\|\|C_{K,n,\lambda}^{-1/2}(S_{K,n}^*\mathbf{y} - C_{K,n}Pf_\lambda)\|_{\mathcal{H}} + \|O_{K,n,\lambda}^{1/2}r_\lambda(O_{K,n})Pf_\lambda\|_{\mathcal{H}}$$
$$\le \|O_{K,n,\lambda}g_\lambda(O_{K,n})\|\|C_{K,n,\lambda}^{-1/2}(S_{K,n}^*\mathbf{y} - C_{K,n}Pf_\lambda)\|_{\mathcal{H}} + \|O_{K,n,\lambda}^{1/2}r_\lambda(O_{K,n})Pf_\lambda\|_{\mathcal{H}}$$
$$\le \sup_{t\in[0,\kappa^2]}|(t+\lambda)g_\lambda(t)|\|C_{K,n,\lambda}^{-1/2}(S_{K,n}^*\mathbf{y} - C_{K,n}Pf_\lambda)\|_{\mathcal{H}} + \|O_{K,n,\lambda}^{1/2}r_\lambda(O_{K,n})Pf_\lambda\|_{\mathcal{H}}$$
$$\le 2E\|C_{K,n,\lambda}^{-1/2}(S_{K,n}^*\mathbf{y} - C_{K,n}Pf_\lambda)\|_{\mathcal{H}} + \|O_{K,n,\lambda}^{1/2}r_\lambda(O_{K,n})Pf_\lambda\|_{\mathcal{H}} \tag{28}$$

where the first equality holds by the definition of $r_\lambda(t)$, the second equality holds by the fact that $O_{K,n}Pf_\lambda = PC_{K,n}P^2f_\lambda = PC_{K,n}Pf_\lambda = O_{K,n}f_\lambda$, and the last inequality holds by the property of filter function in (13).

So we need prove that $\|O_{K,n,\lambda}^{1/2}g_\lambda(O_{K,n})PC_{K,n,\lambda}^{1/2}\| = \|O_{K,n,\lambda}g_\lambda(O_{K,n})\|$ in the fourth inequality. Note that $C_{K,n}$ and $O_{K,n} = PC_{K,n}P$ are self-adjoint, then any continuous function $f$ and $g$, we have $f(O_{K,n})g(O_{K,n}) = g(O_{K,n})f(O_{K,n})$. So we have

$$\|O_{K,n,\lambda}^{1/2}g_\lambda(O_{K,n})PC_{K,n,\lambda}^{1/2}\| = \|g_\lambda(O_{K,n})O_{K,n,\lambda}^{1/2}PC_{K,n,\lambda}^{1/2}\|.$$

Note that for any $a \in \mathcal{H}_K$, $\langle P^2a, a\rangle_{\mathcal{H}} = \|Pa\|_{\mathcal{H}}^2 \le \|a\|_{\mathcal{H}}^2 = \langle a, a\rangle_{\mathcal{H}}$ for $\|P\| \le 1$. So we have

$$P(C_{K,n} + \lambda I)P \preceq PC_{K,n}P + \lambda I = O_{K,n,\lambda}.$$

Let $B = g_\lambda(O_{K,n})O_{K,n,\lambda}^{1/2}$, we have

$$\|BPC_{K,n,\lambda}^{1/2}\|^2 = \|BPC_{K,n,\lambda}PB^*\| \le \|BO_{K,n,\lambda}B^*\| = \|BO_{K,n,\lambda}^{1/2}\|^2,$$

where $B^* = O_{K,n,\lambda}^{1/2}g_\lambda(O_{K,n})$. Combining the above, we have

$$\|O_{K,n,\lambda}^{1/2}g_\lambda(O_{K,n})PC_{K,n,\lambda}^{1/2}\| = \|g_\lambda(O_{K,n})O_{K,n,\lambda}\| = \|O_{K,n,\lambda}g_\lambda(O_{K,n})\|.$$

Next, we bound the first term on the right-hand side of (28),

$$\|C_{K,n,\lambda}^{-1/2}(S_{K,n}^*\mathbf{y} - C_{K,n}Pf_\lambda)\|_{\mathcal{H}}$$

$$\leq \|C_{K,n,\lambda}^{-1/2}(S_{K,n}^*\mathbf{y} - C_{K,n}f_\lambda)\|_{\mathcal{H}} + \|C_{K,n,\lambda}^{-1/2}C_{K,n}(I-P)f_\lambda\|_{\mathcal{H}}$$

$$\leq \|C_{K,n,\lambda}^{-1/2}C_{k,\lambda}^{1/2}\|\|C_{K,\lambda}^{-1/2}(S_{K,n}^*\mathbf{y} - C_{K,n}f_\lambda)\|_{\mathcal{H}} + \|C_{K,n,\lambda}^{-1/2}C_{K,n}^{1/2}\|\|C_{K,n}^{1/2}(I-P)\|$$

$$\|(I-P)C_K^{1/2}\|\|C_K^{-1/2}f_\lambda\|_{\mathcal{H}} \tag{29}$$

$$\leq \mathcal{Q}_{n,\lambda}(\mathcal{P}_{n,\lambda} + \|C_{K,\lambda}^{-1/2}(S_K^*f_\rho - C_Kf_\lambda)\|_{\mathcal{H}}) + ER\phi(\kappa^2)\kappa^{-(2\wedge s)}\lambda^{-(1-\frac{s}{2})_+}\mathcal{A}_{K,P}\mathcal{A}_{K,n,P}$$

$$\leq \mathcal{Q}_{n,\lambda}(\mathcal{P}_{n,\lambda} + \|C_{K,\lambda}^{-1/2}S_K^*\|\|f_\rho - S_Kf_\lambda\|_\rho) + ER\phi(\kappa^2)\kappa^{-(2\wedge s)}\lambda^{-(1-\frac{s}{2})_+}\mathcal{A}_{K,P}\mathcal{A}_{K,n,P}$$

$$\leq \mathcal{Q}_{n,\lambda}(\mathcal{P}_{n,\lambda} + c_\tau R\phi(\lambda)) + ER\phi(\kappa^2)\kappa^{-(2\wedge s)}\lambda^{-(1-\frac{s}{2})_+}\mathcal{A}_{K,P}\mathcal{A}_{K,n,P},$$

where $\mathcal{A}_{K,n,P} = \|C_{K,n}^{1/2}(I-P)\|$, the third inequality holds by the Cordes inequality in Lemma 24, the definition of $\mathcal{P}_{n,\lambda}$, $\mathcal{Q}_{n,\lambda}$, $\mathcal{A}_{K,P}$ and $\mathcal{A}_{K,n,P}$, and Lemma 9, the last inequality holds by the fact that $\|C_{K,\lambda}^{-1/2}S_K^*\|^2 = \|S_KC_{K,\lambda}^{-1}S_K^*\| \leq 1$ and Proposition 2 with $\gamma = 0$.

Now we bound $\mathcal{A}_{K,n,P}$ by $\mathcal{A}_{K,P}$ and $\mathcal{Q}_{n,\lambda}$. When $s < 2$, note that

$$\|C_{K,n}^{1/2}(I-P)\| \leq \|C_{K,n,\lambda}^{1/2}(I-P)\| \leq \|C_{K,n,\lambda}^{1/2}C_{K,\lambda}^{-1/2}\|\|C_{K,\lambda}^{1/2}(I-P)\| \leq \mathcal{Q}_{n,\lambda}\|C_{K,\lambda}^{1/2}(I-P)\|.$$

And for any $f \in \mathcal{H}_K$ and $\|f\|_{\mathcal{H}} = 1$,

$$\|C_{K,\lambda}^{1/2}(I-P)f\|_{\mathcal{H}}^2 = \langle C_{K,\lambda}(I-P)f, (I-P)f\rangle_{\mathcal{H}} = \|C_K^{1/2}(I-P)f\|_{\mathcal{H}}^2 + \lambda\|(I-P)f\|_{\mathcal{H}}^2$$

$$\leq \mathcal{A}_{K,P}^2 + \lambda.$$

Combining the above two inequalities, we have

$$\mathcal{A}_{K,n,P} \leq \mathcal{Q}_{n,\lambda}(\mathcal{A}_{K,P}^2 + \lambda)^{1/2} \leq \mathcal{Q}_{n,\lambda}(\mathcal{A}_{K,P} + \lambda^{1/2}). \tag{30}$$

When $s \geq 2$, according to Lemma 26, we have

$$\mathcal{A}_{K,n,P}^2 = \|C_{K,n}^{1/2}(I-P)C_{K,n}^{1/2}\| \leq \|C_{K,n} - C_K\| + \|C_K^{1/2}(I-P)C_K^{1/2}\|$$

$$\leq \mathcal{U}_{n,\lambda} + \mathcal{A}_{K,P}^2. \tag{31}$$

Recall the definition of $\mathcal{T}_{n,\lambda}$, Combining (26), (28), (29) (30) and (31), we can obtain that

$$\|C_{K,n,\lambda}^{1/2}(f_{n,\lambda}^{rp} - Pf_\lambda)\|_{\mathcal{H}}$$

$$\leq \begin{cases} \lambda^{-\frac{\gamma}{2}}\mathcal{Q}_{n,\lambda}^{1-\gamma}\left[\mathcal{Q}_{n,\lambda}(\mathcal{P}_{n,\lambda} + c_\tau R\phi(\lambda)) + C\lambda^{\frac{s}{2}-1}\mathcal{Q}_{n,\lambda}(\mathcal{A}_{K,P}^2 + \lambda^{1/2}\mathcal{A}_{K,P}) + \mathcal{T}_{n,\lambda}\right], & \text{if } s < 2; \\ \lambda^{-\frac{\gamma}{2}}\mathcal{Q}_{n,\lambda}^{1-\gamma}\left[\mathcal{Q}_{n,\lambda}(\mathcal{P}_{n,\lambda} + c_\tau R\phi(\lambda)) + C(\mathcal{U}_{n,\lambda}^{1/2}\mathcal{A}_{K,P} + \mathcal{A}_{K,P}^2) + \mathcal{T}_{n,\lambda}\right], & \text{if } s \geq 2. \end{cases}$$

where $C = \begin{cases} ER\phi(\kappa^2)\kappa^{-2}, & \text{if } s < 2; \\ ER\phi(\kappa^2)\kappa^{-s}, & \text{if } s \geq 2. \end{cases}$

Thus we complete the proof. $\qquad\square$

**Remark 5.** *Compared with the empirical error bound in Lin & Cevher (2020a), our analysis differs in two key aspects. First, We exploit the embedding property in Assumption 3 to obtain a refined analysis and sharper bounds for $\mathcal{P}_{n,\lambda}$ and $\mathcal{Q}_{n,\lambda}$ (see Lemma 17 and Lemma 13). Second, we derive a more general bound for $\mathcal{T}_{n,\lambda}$ (see Lemma 18), which accommodates a wider class of source conditions. Moreover, the analysis for $\mathcal{T}_{n,\lambda}$ is more straightforward and easier to follow. These two improvements yield a sharper empirical error bound and thereby enlarge the range of the smoothness index s, for which optimal rates are achieved.*

## E.4 TECHNICAL LEMMAS AND OPERATOR SIMILARITY BOUNDS

In this section, we present several technical lemmas and operator similarity bounds that will later be used to control the error terms derived in the previous section. We begin by providing an upper bound on $\mathcal{N}_\infty(\lambda)$, as defined in Definition 1, by leveraging the embedding property stated in Assumption 3.

**Lemma 11.** *Under Assumption 3, for some $\alpha \in [\frac{1}{\beta}, 1]$, there holds*

$$\mathcal{N}_\infty(\lambda) = \sup_{x \in \mathcal{X}} \|C_{K,\lambda}^{-1/2} K_x\|_{\mathcal{H}}^2 \leq A^2 \lambda^{-\alpha}.$$

*Proof.* According to the definition of $\mathcal{N}_\infty(\lambda)$ in Definition 1, we have

$$\mathcal{N}_\infty(\lambda) = \sup_{\boldsymbol{x} \in \mathcal{X}} \langle K_{\boldsymbol{x}}, (C_K + \lambda I)^{-1} K_{\boldsymbol{x}} \rangle_{\mathcal{H}} = \sup_{\boldsymbol{x} \in \mathcal{X}} \|(C_K + \lambda I)^{-1/2} K_{\boldsymbol{x}}\|_{\mathcal{H}}^2$$

$$= \sup_{\boldsymbol{x} \in \mathcal{X}} \left\| \sum_{k=1}^\infty \frac{\mu_k^{1/2} e_k(\boldsymbol{x})}{(\mu_k + \lambda)^{1/2}} \mu_k^{1/2} e_k \right\|_{\mathcal{H}}^2 = \sup_{\boldsymbol{x} \in \mathcal{X}} \sum_{k=1}^\infty \frac{\mu_k}{\mu_k + \lambda} e_k^2(\boldsymbol{x})$$

$$= \left( \sup_{k \geq 1} \frac{\mu_k^{1-\alpha}}{\mu_k + \lambda} \right) \left( \sup_{\boldsymbol{x} \in \mathcal{X}} \sum_{k=1}^\infty \mu_k^\alpha e_k^2(\boldsymbol{x}) \right)$$

$$\leq A^2 \lambda^{-\alpha},$$

where the last inequality is from two facts: (i) according to the Assimption 3 and inequalities (16) and (17) in Fischer & Steinwart (2020), we have $\sup_{\boldsymbol{x} \in \mathcal{X}} \sum_{k=1}^\infty \mu_k^\alpha e_k^2(\boldsymbol{x}) \leq A^2$ and (ii) according to Lemma 23 that $\sup_{t \geq 0} t^a (t + \lambda)^{-1} \leq \lambda^{a-1}$ for $a \in [0, 1]$ and $\lambda > 0$. Thus we complete the proof. $\square$

This lemma is essential for the refined analysis of operator similarity, as it provides the foundation for establishing sharper bounds in the subsequent results. In the following, we separately bound the terms $\mathcal{R}_{n,\lambda}, \mathcal{Q}_{n,\lambda}, \mathcal{S}_{n,\lambda}, \mathcal{U}_{n,\lambda}, \mathcal{P}_{n,\lambda}$ and $\mathcal{T}_{n,\lambda}$ defined in Section E.3.3.

**Lemma 12.** *Under Assumption 3, for any $\delta \in (0, 1)$, with probability at least $1 - \delta$, there holds*

$$\mathcal{R}_{n,\lambda} \leq \frac{2A^2 \log(2/\delta)}{n\lambda^\alpha} + \sqrt{\frac{2A^2 \log(2/\delta)}{n\lambda^\alpha}}.$$

*Proof.* Recall the definition of $C_K$ and $C_{K,n}$, we have

$$C_{K,n} = \frac{1}{n} \sum_{i=1}^n K_{\boldsymbol{x}_i} \otimes K_{\boldsymbol{x}_i} \quad \text{and} \quad C_K = \mathbb{E}_{\boldsymbol{x}}[K_{\boldsymbol{x}} \otimes K_{\boldsymbol{x}}],$$

Note that $\langle K_{\boldsymbol{x}}, (C_K + \lambda I)^{-1} K_{\boldsymbol{x}} \rangle_{\mathcal{H}} \leq \mathcal{N}_\infty(\lambda)$ from Definition 1. Then by Lemma 32 with $Q = C_K$ and $v_i = K_{\boldsymbol{x}_i}$, we can obtain that

$$\mathcal{R}_{n,\lambda} = \|C_{K,\lambda}^{-1/2}(C_K - C_{K,n})C_{K,\lambda}^{-1/2}\| \leq \frac{2\mathcal{N}_\infty(\lambda) \log(2/\delta)}{n} + \sqrt{\frac{2\mathcal{N}_\infty(\lambda) \log(2/\delta)}{n}}$$

$$\leq \frac{2A^2 \log(2/\delta)}{n\lambda^\alpha} + \sqrt{\frac{2A^2 \log(2/\delta)}{n\lambda^\alpha}},$$

where the last inequality holds by Lemma 11. Thus we complete the proof. $\square$

**Lemma 13.** *Under Assumption 3, if $n \geq 16A^2\lambda^{-\alpha} \log(2/\delta)$, then for any $\delta \in (0, 1)$, with probability at least $1 - \delta$, there holds*

$$\mathcal{R}_{n,\lambda} \leq \frac{1}{2},$$

*and*

$$\mathcal{Q}_{n,\lambda} \leq \sqrt{2}.$$

*Proof.* If $n \geq 16A^2\lambda^{-\alpha}\log(2/\delta)$, from Lemma 12, we have

$$\mathcal{R}_{n,\lambda} \leq \frac{2A^2\log(2/\delta)}{n\lambda^\alpha} + \sqrt{\frac{2A^2\log(2/\delta)}{n\lambda^\alpha}} < \frac{1}{2}.$$

By Lemma 25 with $A = C_K$, $B = C_{K,n}$, and $\eta = 1/2$, we have

$$\|C_{K,n,\lambda}^{-1/2}C_{K,\lambda}^{1/2}\| \leq \sqrt{2}.$$

On the other hand, there holds that

$$\|C_{K,n,\lambda}^{1/2}C_{K,\lambda}^{-1/2}\|^2 = \|C_{K,\lambda}^{-1/2}C_{K,n,\lambda}C_{K,\lambda}^{-1/2}\| = \|C_{K,\lambda}^{-1/2}(C_{K,n} + \lambda I)C_{K,\lambda}^{-1/2}\|$$

$$= \|C_{K,\lambda}^{-1/2}(C_{K,n} - C_K + C_K + \lambda I)C_{K,\lambda}^{-1/2}\|$$

$$= \|C_{K,\lambda}^{-1/2}(C_{K,n} - C_K)C_{K,\lambda}^{-1/2} + I\| \leq \mathcal{R}_{n,\lambda} + 1 \leq \frac{3}{2}.$$

By combining the two inequalities, we have

$$\mathcal{Q}_{n,\lambda} = \|C_{K,\lambda}^{1/2}C_{K,n,\lambda}^{-1/2}\| \vee \|C_{K,\lambda}^{-1/2}C_{K,n,\lambda}^{1/2}\| \leq \sqrt{2}.$$

Thus we complete the proof. $\qquad\square$

**Lemma 14.** *Under Assumptions 2 and 3, for any $\delta \in (0,1)$, with probability at least $1 - \delta$, there holds*

$$\mathcal{S}_{n,\lambda} \leq C_{\mathcal{S}}\left(\frac{1}{n\lambda^{\alpha/2}} + \sqrt{\frac{1}{n\lambda^{1/\beta}}}\right)\log\frac{2}{\delta},$$

*where $C_{\mathcal{S}} = \max\{4\kappa A, 2\kappa^2 Q\}$.*

*Proof.* Recall the definition of $C_K$ and $C_{K,n}$, we have

$$C_{K,n} = \frac{1}{n}\sum_{i=1}^n K_{\boldsymbol{x}_i} \otimes K_{\boldsymbol{x}_i} \quad\text{and}\quad C_K = \mathbb{E}_{\boldsymbol{x}}[K_{\boldsymbol{x}} \otimes K_{\boldsymbol{x}}],$$

Note that $\mathcal{N}_\infty(\lambda) = \sup_{\boldsymbol{x}}\langle K_{\boldsymbol{x}}, (C_K + \lambda I)^{-1}K_{\boldsymbol{x}}\rangle_{\mathcal{H}} = \sup_{\boldsymbol{x}}\|(C_K + \lambda I)^{-1}K_{\boldsymbol{x}}\|_{\mathcal{H}}^2$ and from Definition 1 that

$$\mathcal{N}(\lambda) = \mathbb{E}\langle K_{\boldsymbol{x}}, (C_K + \lambda I)^{-1}K_{\boldsymbol{x}}\rangle_{\mathcal{H}}$$

$$= \int_{\mathcal{X}} \|(C_K + \lambda I)^{-1/2}K_{\boldsymbol{x}}\|_{\mathcal{H}}^2 d\rho_{\mathcal{X}}(\boldsymbol{x})$$

$$= \int_{\mathcal{X}} \text{Tr}\big((C_K + \lambda I)^{-1/2}K_{\boldsymbol{x}} \otimes K_{\boldsymbol{x}}(C_K + \lambda I)^{-1/2}\big)d\rho_{\mathcal{X}}(\boldsymbol{x}) \qquad(32)$$

$$= \text{Tr}\left((C_K + \lambda I)^{-1/2}\int_{\mathcal{X}} K_{\boldsymbol{x}} \otimes K_{\boldsymbol{x}}d\rho_{\mathcal{X}}(\boldsymbol{x})(C_K + \lambda I)^{-1/2}\right)$$

$$= \text{Tr}((C_K + \lambda I)^{-1}C_K) = \text{Tr}((L_K + \lambda I)^{-1}L_K).$$

Then by Lemma 31 with $Q = C_K$ and $v_i = z_i = K_{\boldsymbol{x}_i}$, we can obtain that

$$\|(C_K + \lambda I)^{-1/2}(C_K - C_{K,n})\| \leq \frac{4\kappa\sqrt{\mathcal{N}_\infty(\lambda)}\log(2/\delta)}{n} + \sqrt{\frac{4\kappa^2\mathcal{N}(\lambda)\log(2/\delta)}{n}}$$

$$\leq \frac{4\kappa A\log(2/\delta)}{n\lambda^{\alpha/2}} + \sqrt{\frac{4\kappa^2 Q^2\log(2/\delta)}{n\lambda^{1/\beta}}},$$

where the last inequality holds from Lemma 11 and (32) that $\mathcal{N}_\infty(\lambda) \leq A^2\lambda^{-\alpha}$ and $\mathcal{N}(\lambda) \leq Q^2\lambda^{-1/\beta}$. Thus we complete the proof. $\qquad\square$

**Lemma 15.** *For any $\delta \in (0,1)$, with probability at least $1 - \delta$, there holds*

$$\mathcal{U}_{n,\lambda} \leq \|C_K - C_{K,n}\|_{HS} \leq \frac{C_{\mathcal{U}}}{\sqrt{n}}\log\frac{2}{\delta},$$

*where $C_{\mathcal{U}} = 6\kappa^2$.*

*Proof.* Recall the definition of $C_K$ and $C_{K,n}$, we have

$$C_{K,n} = \frac{1}{n} \sum_{i=1}^n K_{\boldsymbol{x}_i} \otimes K_{\boldsymbol{x}_i} \quad \text{and} \quad C_K = \mathbb{E}_{\boldsymbol{x}}[K_{\boldsymbol{x}} \otimes K_{\boldsymbol{x}}],$$

Note that $\|K_{\boldsymbol{x}}\|_{\mathcal{H}}^2 \leq \kappa^2$, then applying Lemma 30 with $\widetilde{B} = 2\kappa^2$ and $\widetilde{\sigma} = \kappa^2$, we can obtain that

$$\|C_K - C_{K,n}\|_{HS} \leq \frac{4\kappa^2 \log(2/\delta)}{n} + \sqrt{\frac{2\kappa^2 \log(2/\delta)}{n}} \leq \frac{6\kappa^2}{\sqrt{n}} \log \frac{2}{\delta}.$$

Thus we complete the proof. $\qquad\square$

Now we bound $\mathcal{P}_{n,\lambda}$ to control the sample variance. Compared to that in Lin & Cevher (2020a) and Lin & Cevher (2020b), we do not impose the uniform boundedness assumption on the regression function $f_\rho$. Inspired by Fischer & Steinwart (2020) and Zhang et al. (2024), we adopt a truncation technique and the $L^q$-embedding property of the interpolation space in Lemma 4 to address the potential unboundedness of $f_\rho$, which yields a more general result. While the proof strategies are indeed similar, there are two important differences: (i) our statements and arguments are formulated under a general source condition via an index function, whereas Fischer & Steinwart (2020); Zhang et al. (2024) work with Hölder source conditions; (ii) the resulting bounds and conditions on the sample size $n$ differ in several details, reflecting our more general assumptions (e.g., the dependence on the source function).

The following lemma provides a bound on the truncation level of $\mathcal{P}_{n,\lambda}$.

**Lemma 16.** *Under Assumptions 1-4, denote $\xi_i^t = C_{K,\lambda}^{-1/2}(y - f_\lambda(\boldsymbol{x}))K_{\boldsymbol{x}}\mathbb{I}_{\boldsymbol{x} \in \mathcal{X}_t}$ with $\mathcal{X}_t = \{\boldsymbol{x} \in \mathcal{X} : |f_\rho(\boldsymbol{x})| \leq t\}$, then for any $\delta \in (0,1)$, with probability at least $1 - \delta$, there holds*

$$\left\| \frac{1}{n} \sum_{i=1}^n \xi_i^t - \mathbb{E}\xi^t \right\|_{\mathcal{H}} \leq C_{\mathcal{P}}^t \left( \frac{\lambda^{\min\{0, \frac{s-\alpha}{2}\}} + t + 1}{n\lambda^{\frac{\alpha}{2}}} + \frac{\phi(\lambda)\lambda^{-\frac{\alpha}{2}} + \lambda^{-\frac{1}{2\beta}}}{\sqrt{n}} \right) \log \frac{2}{\delta},$$

*where $C_{\mathcal{P}}^t = \left(8(EA\phi(\kappa^2)\max\{\kappa^{-s}, \kappa^{-\alpha}\} + 1 + M) + 4\sigma + 4\sqrt{2}c_\tau R\right)(2A + Q)$.*

*Proof.* We apply Lemma 30 to prove this lemma. Note that

$$\mathbb{E}\|\xi_i^t - \mathbb{E}[\xi^t]\|_{\mathcal{H}}^p \leq \mathbb{E}_{\xi_i^t}\mathbb{E}_{\xi^t}\|\xi_i^t - \xi^t\|_{\mathcal{H}}^p \leq 2^{p-1}\mathbb{E}_{\xi_i^t}\mathbb{E}_{\xi^t}(\|\xi_i^t\|_{\mathcal{H}}^p + \|\xi^t\|_{\mathcal{H}}^p) = 2^p\mathbb{E}\|\xi^t\|_{\mathcal{H}}^p. \qquad (33)$$

Thus we only need to bound $\mathbb{E}\|\xi^t\|_{\mathcal{H}}^p$. From the definition of $\xi^t$, we have

$$\mathbb{E}\|\xi^t\|_{\mathcal{H}}^p \leq \int_{\mathcal{X} \times \mathcal{Y}} \|C_{K,\lambda}^{-1/2}(y - f_\lambda(\boldsymbol{x}))K_{\boldsymbol{x}}\mathbb{I}_{\boldsymbol{x} \in \mathcal{X}_t}\|_{\mathcal{H}}^p d\rho(\boldsymbol{x}, y)$$

$$\leq 2^{p-1} \int_{\mathcal{X} \times \mathcal{Y}} \|C_{K,\lambda}^{-1/2}K_{\boldsymbol{x}}\|_{\mathcal{H}} \left(|y - f_\rho(\boldsymbol{x})|^p\mathbb{I}_{\boldsymbol{x} \in \mathcal{X}_t} + |f_\rho(\boldsymbol{x}) - f_\lambda(\boldsymbol{x})|^p\mathbb{I}_{\boldsymbol{x} \in \mathcal{X}_t}\right) d\rho(\boldsymbol{x}, y)$$

$$= 2^{p-1} \int_{\mathcal{X}} \|C_{K,\lambda}^{-1/2}K_{\boldsymbol{x}}\|_{\mathcal{H}}^p \int_{\mathcal{Y}} |\varepsilon|^p\mathbb{I}_{\boldsymbol{x} \in \mathcal{X}_t} d\rho(\varepsilon|\boldsymbol{x})d\rho_{\mathcal{X}}(\boldsymbol{x})$$

$$+ 2^{p-1} \int_{\mathcal{X}} \|C_{K,\lambda}^{-1/2}K_{\boldsymbol{x}}\|_{\mathcal{H}}^p |f_\rho(\boldsymbol{x}) - f_\lambda(\boldsymbol{x})|^p\mathbb{I}_{\boldsymbol{x} \in \mathcal{X}_t} d\rho_{\mathcal{X}}(\boldsymbol{x})$$

$$\leq 2^{p-1} \int_{\mathcal{X}} \|C_{K,\lambda}^{-1/2}K_{\boldsymbol{x}}\|_{\mathcal{H}}^p \int_{\mathcal{Y}} |\varepsilon|^p\mathbb{I}_{\boldsymbol{x} \in \mathcal{X}_t} d\rho(\varepsilon|\boldsymbol{x})d\rho_{\mathcal{X}}(\boldsymbol{x})$$

$$+ 2^{p-1}\|(f_\rho - f_\lambda)\mathbb{I}_{\boldsymbol{x} \in \mathcal{X}_t}\|_{\infty}^{p-2} \int_{\mathcal{X}} \|C_{K,\lambda}^{-1/2}K_{\boldsymbol{x}}\|_{\mathcal{H}}^p |f_\rho(\boldsymbol{x}) - f_\lambda(\boldsymbol{x})|^2\mathbb{I}_{\boldsymbol{x} \in \mathcal{X}_t} d\rho_{\mathcal{X}}(\boldsymbol{x}).$$

(34)

For the first term in the right side of (34), from Assumption 1 and definition 1, we have

$$
2^{p-1} \int_{\mathcal{X}} \|C_{K,\lambda}^{-1/2} K_{\boldsymbol{x}}\|_{\mathcal{H}}^p \int_{\mathcal{Y}} |\varepsilon|^p \mathbb{I}_{\boldsymbol{x} \in \mathcal{X}_t} d\rho(\varepsilon|\boldsymbol{x}) d\rho_{\mathcal{X}}(\boldsymbol{x})
$$

$$
\leq 2^{p-1} \int_{\mathcal{X}} \|C_{K,\lambda}^{-1/2} K_{\boldsymbol{x}}\|_{\mathcal{H}}^p \int_{\mathcal{Y}} |\varepsilon|^p d\rho(\varepsilon|\boldsymbol{x}) d\rho_{\mathcal{X}}(\boldsymbol{x})
$$

$$
\leq \frac{1}{2} 2^{p-1} p! \sigma^2 M^{p-2} \int_{\mathcal{X}} \|C_{K,\lambda}^{-1/2} K_{\boldsymbol{x}}\|_{\mathcal{H}}^p d\rho_{\mathcal{X}}(\boldsymbol{x}) \tag{35}
$$

$$
\leq \frac{1}{2} 2^{p-1} p! \sigma^2 M^{p-2} \left( \sup_{\boldsymbol{x} \in \mathcal{X}} \|C_{K,\lambda}^{-1/2} K_{\boldsymbol{x}}\|^{p-2} \right) \int_{\mathcal{X}} \|C_{K,\lambda}^{-1/2} K_{\boldsymbol{x}}\|^2 d\rho_{\mathcal{X}}(\boldsymbol{x})
$$

$$
= \frac{1}{2} p! \left( \sqrt{2\mathcal{N}(\lambda)\sigma^2} \right)^2 \left( 2M\sqrt{\mathcal{N}_\infty(\lambda)} \right)^{p-2}.
$$

For the second term in the right side of (34), we have

$$
\int_{\mathcal{X}} \|C_{K,\lambda}^{-1/2} K_{\boldsymbol{x}}\|_{\mathcal{H}}^p |f_\rho(\boldsymbol{x}) - f_\lambda(\boldsymbol{x})|^2 \mathbb{I}_{\boldsymbol{x} \in \mathcal{X}_t} d\rho_{\mathcal{X}}(\boldsymbol{x})
$$

$$
\leq \left( \sup_{\boldsymbol{x} \in \mathcal{X}} \|C_{K,\lambda}^{-1/2} K_{\boldsymbol{x}}\|^p \right) \int_{\mathcal{X}} |f_\rho(\boldsymbol{x}) - f_\lambda(\boldsymbol{x})|^2 \mathbb{I}_{\boldsymbol{x} \in \mathcal{X}_t} d\rho_{\mathcal{X}}(\boldsymbol{x}) \tag{36}
$$

$$
\leq (\mathcal{N}_\infty(\lambda))^{p/2} \|f_\rho - f_\lambda\|_\rho^2 \leq c_\tau^2 R^2 (\mathcal{N}_\infty(\lambda))^{p/2} \phi^2(\lambda),
$$

where the last inequality holds from Proposition 2 with $\gamma = 0$.

On the other hand, from Lemma 10, we have

$$
\|(f_\rho - f_\lambda)\mathbb{I}_{\boldsymbol{x} \in \mathcal{X}_t}\|_\infty \leq \|f_\rho \mathbb{I}_{\boldsymbol{x} \in \mathcal{X}_t}\|_\infty + \|f_\lambda\|_\infty
$$

$$
\leq EA\phi(\kappa^2) \max\{\kappa^{-s}, \kappa^{-\alpha}\} \lambda^{\min\{0, \frac{s-\alpha}{2}\}} + t. \tag{37}
$$

Plugging (34), (35), (36) and (37) into (33), we have

$$
\mathbb{E}\|\xi_i^t - \mathbb{E}[\xi^t]\|_{\mathcal{H}}^p \leq \frac{1}{2} p! \left( 4(EA\phi(\kappa^2) \max\{\kappa^{-s}, \kappa^{-\alpha}\} \lambda^{\min\{0, \frac{s-\alpha}{2}\}} + t + M) \sqrt{\mathcal{N}_\infty(\lambda)} \right)^{p-2}
$$

$$
\left( (2\sqrt{2}\sigma + 4c_\tau R) \left( \sqrt{\mathcal{N}_\infty(\lambda)}\phi(\lambda) + \sqrt{\mathcal{N}(\lambda)} \right) \right)^2
$$

Applying Lemma 30 with $\widetilde{B} = 4(EA\phi(\kappa^2) \max\{\kappa^{-s}, \kappa^{-\alpha}\} \lambda^{\min\{0, \frac{s-\alpha}{2}\}} + t + M) \sqrt{\mathcal{N}_\infty(\lambda)}$ and $\widetilde{\sigma} = (2\sqrt{2}\sigma + 4c_\tau R) \left( \sqrt{\mathcal{N}_\infty(\lambda)}\phi(\lambda) + \sqrt{\mathcal{N}(\lambda)} \right)$, Assimption 2, and Lemma 11, we can obtain that with probability at least $1 - \delta$, there holds

$$
\left\| \frac{1}{n} \sum_{i=1}^n \xi_i^t - \mathbb{E}\xi^t \right\|_{\mathcal{H}} \leq C_{\mathcal{P}}^t \left( \frac{\lambda^{\min\{0, \frac{s-\alpha}{2}\}} + t + 1}{n\lambda^{\frac{\alpha}{2}}} + \frac{\phi(\lambda)\lambda^{-\frac{\alpha}{2}} + \lambda^{-\frac{1}{2\beta}}}{\sqrt{n}} \right) \log \frac{2}{\delta},
$$

where $C_{\mathcal{P}}^t = \left( 8(EA\phi(\kappa^2) \max\{\kappa^{-s}, \kappa^{-\alpha}\} + 1 + M) + 4\sigma + 4\sqrt{2}c_\tau R \right) (2A + Q)$. Thus we complete the proof.

$\square$

By combining the truncation technique and the bound in Lemma 16, we can derive a bound on $\mathcal{P}_{n,\lambda}$ as follows.

**Lemma 17.** *Under Assumptions 1-4, if $n \geq (\lambda^\alpha \phi(\lambda))^{-\frac{\alpha}{s+\alpha}}$, then for any $\delta \in (0,1)$, with probability at least $1 - \delta$, there holds*

$$
\mathcal{P}_{n,\lambda} \leq C_{\mathcal{P}} \left( \frac{\lambda^{\min\{0, \frac{s-\alpha}{2}\}} + 1}{n\lambda^{\frac{\alpha}{2}}} + \frac{\phi(\lambda)\lambda^{-\frac{\alpha}{2}} + \lambda^{-\frac{1}{2\beta}}}{\sqrt{n}} + \phi(\lambda) \right) \log \frac{4}{\delta},
$$

*where $C_{\mathcal{P}} = C_{\mathcal{P}}^t + ARc_\tau(C_qR)^{q/2} + A\sigma(C_qR)^q + 1$.*

*Proof.* Recall that $\mathcal{P}_{n,\lambda} = \left\| C_{K,\lambda}^{-1/2} \left[ (S_{K,n}^* \mathbf{y} - C_{k,n} f_\lambda) - (S_K^* f_\rho - C_K f_\lambda) \right] \right\|_{\mathcal{H}}$, for $i \in \{1, \dots, n\}$, let

$$\xi_i = C_{K,\lambda}^{-1/2}(y_i - f_\lambda(\boldsymbol{x}_i)) K_{\boldsymbol{x}_i},$$

and note that

$$\int_{\mathcal{X} \times \mathcal{Y}} y K_{\boldsymbol{x}} d\rho(\boldsymbol{x}, y) = \int_{\mathcal{Y}} y d\rho(y|\boldsymbol{x}) \int_{\mathcal{X}} K_{\boldsymbol{x}} d\rho_{\mathcal{X}}(\boldsymbol{x}) = \int_{\mathcal{X}} f_\rho(\boldsymbol{x}) K_{\boldsymbol{x}} d\rho_{\mathcal{X}}(\boldsymbol{x}),$$

then we have

$$\mathbb{E}[\xi] = C_{K,\lambda}^{-1/2} \int_{\mathcal{X}} (f_\rho(\boldsymbol{x}) - f(\lambda)) K_{\boldsymbol{x}} d\rho_{\mathcal{X}}(\boldsymbol{x}).$$

Thus $\mathcal{P}_{n,\lambda} = \| \frac{1}{n} \sum_{i=1}^n (\xi_i - \mathbb{E}\xi) \|_{\mathcal{H}}$. Let $\mathcal{X}_t = \{\boldsymbol{x} \in \mathcal{X} : |f_\rho(\boldsymbol{x})| \leq t\}$ and $\mathcal{X}_t' = \mathcal{X} \setminus \mathcal{X}_t$, where $t$ is the truncation level. Assume that for some $q \geq 2$,

$$[\mathcal{H}]_\rho^s \hookrightarrow L_\rho^q(\mathcal{X}). \tag{38}$$

We will prove (38) later. According to Assumption 3 and (38), there exists a constant $C_q$ such that $\|f\|_{L^q(\mathcal{X}, \rho_{\mathcal{X}})} \leq C_q \|f\|_{[\mathcal{H}]_\rho^s} \leq C_q R$. Thus by Markov's inequality, we have

$$\mathbb{P}(\mathcal{X}_t') = \mathbb{P}(\{\boldsymbol{x} \in \mathcal{X} : |f_\rho(\boldsymbol{x})| > t\}) \leq \frac{\|f_\rho\|_{L_\rho^q(\mathcal{X})}}{t^q} \leq \frac{(C_q R)^q}{t^q}.$$

We decompose $\xi$ as $\xi = \xi \mathbb{I}_{\boldsymbol{x} \in \mathcal{X}_t} + \xi \mathbb{I}_{\boldsymbol{x} \in \mathcal{X}_t'}$, then by the triangle inequality, we have

$$\left\| \frac{1}{n} \sum_{i=1}^n (\xi_i - \mathbb{E}\xi) \right\|_{\mathcal{H}} \leq \left\| \frac{1}{n} \sum_{i=1}^n (\xi_i \mathbb{I}_{\boldsymbol{x}_i \in \mathcal{X}_t} - \mathbb{E}[\xi \mathbb{I}_{\boldsymbol{x} \in \mathcal{X}_t}]) \right\|_{\mathcal{H}} + \left\| \frac{1}{n} \sum_{i=1}^n \xi_i \mathbb{I}_{\boldsymbol{x}_i \in \mathcal{X}_t'} \right\|_{\mathcal{H}}$$
$$+ \|\mathbb{E}[\xi \mathbb{I}_{\boldsymbol{x} \in \mathcal{X}_t'}]\|_{\mathcal{H}}. \tag{39}$$

For the first term in the right side of (39), we can bound it by Lemma 16 with $t \leq n\lambda^{\frac{\alpha}{2}} \phi(\lambda)$, i.e.,

$$\left\| \frac{1}{n} \sum_{i=1}^n \xi_i^t - \mathbb{E}\xi^t \right\|_{\mathcal{H}} \leq C_{\mathcal{P}}^t \left( \frac{\lambda^{\min\{0, \frac{s-\alpha}{2}\}} + 1}{n\lambda^{\frac{\alpha}{2}}} + \frac{\phi(\lambda)\lambda^{-\frac{\alpha}{2}} + \lambda^{-\frac{1}{2\beta}}}{\sqrt{n}} + \phi(\lambda) \right) \log \frac{4}{\delta} \tag{40}$$

holds with probability at least $1 - \delta/2$. For the second term in the right side of (39), we have

$$\mathbb{P}\left( \left\| \frac{1}{n} \sum_{i=1}^n \xi_i \mathbb{I}_{\boldsymbol{x}_i \in \mathcal{X}_t'} \right\|_{\mathcal{H}} \geq \frac{1}{\sqrt{n\lambda^{\frac{1}{\beta}}}} \right) \leq \mathbb{P}(\exists \boldsymbol{x}_i s.t. \boldsymbol{x}_i \in \mathcal{X}_t')$$
$$= 1 - (\mathbb{P}(\mathcal{X}_t))^n = 1 - (\mathbb{P}(|f_\rho(\boldsymbol{x})| \leq t))^n \leq 1 - \left( 1 - \frac{(C_q R)^q}{t^q} \right)^n. \tag{41}$$

Let (41) be less than $\delta/2$, we have $t \gtrsim C_q R(\frac{n}{-\log(1-\delta/2)})^{1/q}$ when $n$ is large enough. For the third term in the right side of (39), we have

$$\|\mathbb{E}[\xi \mathbb{I}_{\boldsymbol{x} \in \mathcal{X}_t'}]\|_{\mathcal{H}} \leq \mathbb{E}\|\xi \mathbb{I}_{\boldsymbol{x} \in \mathcal{X}_t'}\|_{\mathcal{H}} \leq \mathbb{E}\left( \|C_{K,\lambda}^{-1/2} K_{\boldsymbol{x}}\|_{\mathcal{H}} |(y - f_\lambda(\boldsymbol{x})) \mathbb{I}_{\boldsymbol{x} \in \mathcal{X}_t'}| \right)$$
$$\leq \sup_{\boldsymbol{x} \in \mathcal{X}} \|C_{K,\lambda}^{-1/2} K_{\boldsymbol{x}}\|_{\mathcal{H}} \left( \mathbb{E}|(f_\rho(\boldsymbol{x}) - f_\lambda(\boldsymbol{x})) \mathbb{I}_{\boldsymbol{x} \in \mathcal{X}_t'}| + \mathbb{E}|\varepsilon \mathbb{I}_{\boldsymbol{x} \in \mathcal{X}_t'}| \right)$$
$$\leq A\lambda^{-\frac{\alpha}{2}} \left( \|f_\rho - f_\lambda\|_\rho (\mathbb{P}(\mathcal{X}_t'))^{1/2} + \mathbb{E}(\mathbb{I}_{\boldsymbol{x} \in \mathcal{X}_t'} \mathbb{E}(|\varepsilon| \mid \boldsymbol{x})) \right)$$
$$\leq A\lambda^{-\frac{\alpha}{2}} \left( \|f_\rho - f_\lambda\|_\rho (\mathbb{P}(\mathcal{X}_t'))^{1/2} + \mathbb{E}(\mathbb{I}_{\boldsymbol{x} \in \mathcal{X}_t'} \mathbb{E}^{1/2}(|\varepsilon|^2 \mid \boldsymbol{x})) \right) \tag{42}$$
$$\leq A\lambda^{-\frac{\alpha}{2}} \left( \|f_\rho - f_\lambda\|_\rho (\mathbb{P}(\mathcal{X}_t'))^{1/2} + \sigma \mathbb{P}(\mathcal{X}_t') \right)$$
$$\leq A\lambda^{-\frac{\alpha}{2}} \left( c_\tau R\phi(\lambda) \frac{(C_q R)^{q/2}}{t^{q/2}} + \sigma \frac{(C_q R)^q}{t^q} \right),$$

where the fourth inequality holds from Cauchy-Schwarz inequality and Lemma 11, the fifth and sixth inequality holds from the Jesen's inequality and Assumption 1, and the last inequality holds from Proposition 2 with $\gamma = 0$ and the bound of $\mathbb{P}(\mathcal{X}_t')$.

When $t \geq n^{1/q}$, we can obtain that

$$\|\mathbb{E}[\xi \mathbb{I}_{\boldsymbol{x} \in \mathcal{X}_t'}]\|_{\mathcal{H}} \leq (ARc_\tau(C_qR)^{q/2} + A\sigma(C_qR)^q)\left(\frac{\phi(\lambda)\lambda^{-\frac{\alpha}{2}}}{\sqrt{n}} + \frac{1}{n\lambda^{\frac{\alpha}{2}}}\right) \tag{43}$$

Ignore the logarithmic factor and constants, the truncation level $t$ exists when $n\lambda^{\frac{\alpha}{2}}\phi(\lambda) > n^{1/q}$, that is $[\mathcal{H}]_\rho^s \hookrightarrow L_\rho^q(\mathcal{X})$ holds for

$$q > q_\lambda = \frac{2\log n}{2\log n + \alpha\log\lambda + \log\phi(\lambda)}. \tag{44}$$

If $s > \alpha$, according to Lemma 1 and assumption 3, we have $[\mathcal{H}]_\rho^s \hookrightarrow [\mathcal{H}]_\rho^\alpha \hookrightarrow L_\rho^\infty(\mathcal{X}) \hookrightarrow L_\rho^q(\mathcal{X}) \hookrightarrow L_\rho^{q_\lambda}(\mathcal{X})$. If $s < \alpha$, from Theorem 4, we have $[\mathcal{H}]_\rho^s \hookrightarrow L_\rho^{q_s}(\mathcal{X})$ with $q_s = \frac{2\alpha}{\alpha-s}$, then (44) holds when $\frac{2\alpha}{\alpha-s} > \frac{2\log n}{2\log n + \alpha\log\lambda + \log\phi(\lambda)}$, that is

$$n \geq (\lambda^\alpha\phi(\lambda))^{-\frac{\alpha}{s+\alpha}}. \tag{45}$$

Combining (39)-(42), if $n$ satisfying (45), then with probability at least $1 - \delta$, there holds that

$$\left\|\frac{1}{n}\sum_{i=1}^n(\xi_i - \mathbb{E}\xi)\right\|_{\mathcal{H}} \leq C_{\mathcal{P}}\left(\frac{\lambda^{\min\{0, \frac{s-\alpha}{2}\}} + 1}{n\lambda^{\frac{\alpha}{2}}} + \frac{\phi(\lambda)\lambda^{-\frac{\alpha}{2}} + \lambda^{-\frac{1}{2\beta}}}{\sqrt{n}} + \phi(\lambda)\right)\log\frac{4}{\delta},$$

where $C_{\mathcal{P}} = C_{\mathcal{P}}^t + ARc_\tau(C_qR)^{q/2} + A\sigma(C_qR)^q + 1$. Thus we complete the proof $\qquad\square$

In the following lemma, we establish bounds for $\mathcal{T}_{n,\lambda}$ under various conditions on the index function $\phi$, which is not considered in Lin & Cevher (2020a).

**Lemma 18.** *Under Assumption 4, then for any $\lambda \in [0, \kappa^2]$, the following holds*

1) *if $\phi : [0, \kappa^2] \to \mathbb{R}^+$ is non-decreasing, and $\phi(0) = 0$, $\phi(\kappa^2) < \infty$, then*

$$\mathcal{T}_{n,\lambda} \lesssim \phi(\lambda)\mathcal{Q}_{n,\lambda}.$$

2) *if $\phi : [0, \kappa^2] \to \mathbb{R}^+$ is Lipschitz continuous with constant 1, then*

$$\mathcal{T}_{n,\lambda} \lesssim \phi(\lambda)\mathcal{Q}_{n,\lambda} + \lambda^{1/2}\mathcal{S}_{n,\lambda}.$$

3) *if $\phi = \psi\vartheta$, where $\psi : [0, \kappa^2] \to \mathbb{R}^+$ is non-decreasing, and $\phi(0) = 0$, $\phi(\kappa^2) < \infty$, and $\vartheta : [0, \kappa^2] \to \mathbb{R}^+$ is non-decreasing and Lipschitz continuous with constant 1, and $\vartheta(0) = 0$, and there exists $c' > 0$ such that*

$$c'\frac{\lambda^\tau}{\vartheta(\lambda)\lambda^{1/2}} \leq \inf_{t \in [\lambda, \kappa^2]}\frac{t^\tau}{\vartheta(t)t^{1/2}}, \quad \forall \lambda \in [0, \kappa^2],$$

   *then*

$$\mathcal{T}_{n,\lambda} \lesssim \phi(\lambda)\mathcal{Q}_{n,\lambda} + \lambda^{1/2}\mathcal{S}_{n,\lambda} + \vartheta(\lambda)\psi(\mathcal{U}_{n,\lambda}).$$

*Proof.* From the definition of $f_\lambda$ and Assumption 4, we have

$$\|O_{K,n,\lambda}^{1/2}r_\lambda(O_{K,n})Pf_\lambda\|_{\mathcal{H}} = \|O_{K,n,\lambda}^{1/2}r_\lambda(O_{K,n})Pg_\lambda(C_K)S_K^*f_\rho\|_{\mathcal{H}}$$

$$=\|O_{K,n,\lambda}^{1/2}r_\lambda(O_{K,n})Pg_\lambda(C_K)S_K^*\phi(L_K)g\|_{\mathcal{H}}$$

$$\leq R\|O_{K,n,\lambda}^{1/2}r_\lambda(O_{K,n})Pg_\lambda(C_K)S_K^*\phi(S_KS_K^*)\|$$

$$=R\|O_{K,n,\lambda}^{1/2}r_\lambda(O_{K,n})Pg_\lambda(C_K)\phi(S_K^*S_K)S_K\|$$

$$=R\|O_{K,n,\lambda}^{1/2}r_\lambda(O_{K,n})Pg_\lambda(C_K)\phi(C_K)C_K^{1/2}\|,$$

where the last inequality is from Lemma 27. Then we bound $\|O_{K,n,\lambda}^{1/2}r_\lambda(O_{K,n})Pg_\lambda(C_K)\phi(C_K)C_K^{1/2}\|$ for three case of $\phi(\cdot)$:

**(1) When $\phi(\cdot)$ is non-decreasing.**

$$\|O_{K,n,\lambda}^{1/2} r_\lambda(O_{K,n}) P g_\lambda(C_K) \phi(C_K) C_K^{1/2}\|$$

$$=\|O_{K,n,\lambda}^{1/2} r_\lambda(O_{K,n}) P C_{K,n,\lambda}^{1/2} C_{K,n,\lambda}^{-1/2} C_{K,n}^{1/2} C_{K,n}^{-1/2} g_\lambda(C_K) \phi(C_K) C_K^{1/2}\|$$

$$\leq\|O_{K,n,\lambda}^{1/2} r_\lambda(O_{K,n}) P C_{K,n,\lambda}^{1/2}\|\|C_{K,n,\lambda}^{-1/2} C_{K,n}^{1/2}\|\|C_{K,n}^{-1/2} C_K^{1/2}\|\|g_\lambda(C_K) \phi(C_K)\| \tag{46}$$

$$\leq\mathcal{Q}_{n,\lambda} \sup_{t\in[0,\kappa^2]} |(t+\lambda) r_\lambda(t)| \sup_{t\in[0,\kappa^2]} |g_\lambda(t)\phi(t)|$$

$$\leq 2F_\tau \lambda \mathcal{Q}_{n,\lambda} \sup_{t\in[0,\kappa^2]} |g_\lambda(t)\phi(t)|,$$

where the second inequality holds by $\|O_{K,n,\lambda}^{1/2} r_\lambda(O_{K,n}) P C_{K,n,\lambda}^{1/2}\| = \|O_{K,n,\lambda} r_\lambda(O_{K,n})\|$, $\|C_{K,n}^{-1/2} C_K^{1/2}\| \leq 1$ and the definition of $\mathcal{Q}_{n,\lambda}$, the last inequality holds by the property of filter function in (13).

When $t \in [0, \lambda]$, by the non-decreasing property of $\phi(\cdot)$ and (13), we have

$$\|g_\lambda(C_K)\phi(C_K)\| \leq \sup_{t\in[0,\kappa^2]} |g_\lambda(t)\phi(t)| \leq E\phi(\lambda)\lambda^{-1}.$$

For $t \in (\lambda, \kappa^2]$, from Lemma 28, there exists some $c'_\phi < \infty$ such that $\phi(t)t^{-1} \leq c'_\phi \phi(\lambda)\lambda^{-1}$, then we have

$$\|g_\lambda(C_K)\phi(C_K)\| \leq \sup_{t\in[0,\kappa^2]} |g_\lambda(t)t\phi(t)t^{-1}| \leq Ec'_\phi\phi(\lambda)\lambda^{-1}$$

Combining the above two inequalities and plugging into (46), we have

$$\|O_{K,n,\lambda}^{1/2} r_\lambda(O_{K,n}) P g_\lambda(C_K) \phi(C_K) C_K^{1/2}\| \leq 2(c'_\phi + 1) E F_\tau \phi(\lambda) \mathcal{Q}_{n,\lambda}. \tag{47}$$

**(2) When $\phi(\cdot)$ is Lipschitz continuous with constant 1 and satisfies that $P\phi(A)P = \phi(PAP)$ for any operator $A$ and projection operator $P$.**

$$\|O_{K,n,\lambda}^{1/2} r_\lambda(O_{K,n}) P g_\lambda(C_K) \phi(C_K) C_K^{1/2}\| = \|O_{K,n,\lambda}^{1/2} r_\lambda(O_{K,n}) P g_\lambda(C_K) \phi(C_{K,n}) C_K^{1/2}\|$$

$$+ \|O_{K,n,\lambda}^{1/2} r_\lambda(O_{K,n}) P g_\lambda(C_K)(\phi(C_K) - \phi(C_{K,n})) C_K^{1/2}\|. \tag{48}$$

For the first term in the right side of (48),

$$\|O_{K,n,\lambda}^{1/2} r_\lambda(O_{K,n}) P g_\lambda(C_K) \phi(C_{K,n}) C_K^{1/2}\|$$

$$=\|P^{1/2} C_{K,n,\lambda}^{1/2} P^{1/2} r_\lambda(O_{K,n}) P g_\lambda(C_K) \phi(C_{K,n}) C_K^{1/2}\|$$

$$\leq\|r_\lambda(O_{K,n}) P \phi(C_{K,n}) P\|\|C_{K,n,\lambda}^{1/2} C_{K,\lambda}^{-1/2}\|\|C_{K,\lambda}^{1/2} g_\lambda(C_K) C_K^{1/2}\| \tag{49}$$

$$\leq\mathcal{Q}_{n,\lambda}\|r_\lambda(O_{K,n}) \phi(O_{K,n})\|\|C_{K,\lambda}^{1/2} g_\lambda(C_K) C_K^{1/2}\|$$

$$\leq\mathcal{Q}_{n,\lambda} \sup_{t\in[0,\kappa^2]} |r_\lambda(t)\phi(t)| \sup_{t\in[0,\kappa^2]} |g_\lambda(t)t^{1/2}(t+\lambda)^{1/2}|$$

$$\leq 2c_\tau E\phi(\lambda)\mathcal{Q}_{n,\lambda},$$

where the second inequality uses $P\phi(C_{K,n})P = \phi(PC_{K,n}P) = \phi(O_{K,n})$, and the last inequality holds by Lemma 8 and the property of filter function in (13) that

$$\sup_{t\in[0,\kappa^2]} |g_\lambda(t)t^{1/2}(t+\lambda)^{1/2}| \leq \sup_{t\in[0,\kappa^2]} |g_\lambda(t)(t+\lambda^{1/2}t^{1/2})| \leq 2E.$$

For the second term on the right side of (48),

$$\|O_{K,n,\lambda}^{1/2} r_\lambda(O_{K,n}) P g_\lambda(C_K)(\phi(C_K) - \phi(C_{K,n})) C_K^{1/2}\|$$

$$\leq \|O_{K,n,\lambda}^{1/2} r_\lambda(O_{K,n}) P C_{K,n}^{1/2}\| \|C_{K,n}^{-1/2} C_K^{1/2}\| \|C_K^{-1/2}(\phi(C_K) - \phi(C_{K,n}))\| \|C_K^{1/2} g_\lambda(C_K)\|$$

$$\leq \|O_{K,n,\lambda}^{1/2} r_\lambda(O_{K,n}) O_{K,n}^{1/2}\| \|C_{K,n}^{-1/2} C_K^{1/2}\| \|C_K^{-1/2}(\phi(C_K) - \phi(C_{K,n}))\| \|C_K^{1/2} g_\lambda(C_K)\|$$

$$\leq \|O_{K,n,\lambda}^{1/2} r_\lambda(O_{K,n}) O_{K,n}^{1/2}\| \|C_{K,n}^{-1/2} C_K^{1/2}\| \|C_K^{-1/2}(C_K - C_{K,n})\| \|C_K^{1/2} g_\lambda(C_K)\| \qquad (50)$$

$$\leq \mathcal{S}_{n,\lambda} \sup_{t \in [0,\kappa^2]} |r_\lambda(t) t^{1/2}(t+\lambda)^{1/2}| \sup_{t \in [0,\kappa^2]} |g_\lambda(t) t^{1/2}|$$

$$\leq 2 E F_\tau \lambda^{1/2} \mathcal{S}_{n,\lambda},$$

where the third inequality holds by the Lipschitz continuity of $\phi(\cdot)$, and the last inequality holds by the property of the filter function in (13) and (13) that

$$\sup_{t \in [0,\kappa^2]} |g_\lambda(t) t^{1/2}| \leq E \lambda^{-1/2},$$

and

$$\sup_{t \in [0,\kappa^2]} |r_\lambda(t) t^{1/2}(t+\lambda)^{1/2}| \leq \sup_{t \in [0,\kappa^2]} |r_\lambda(t)(t + \lambda^{1/2} t^{1/2})| \leq 2 F_\tau.$$

Plugging (49) and (50) into (48), we have

$$\|O_{K,n,\lambda}^{1/2} r_\lambda(O_{K,n}) P g_\lambda(C_K) \phi(C_K) C_K^{1/2}\| \leq 2 c_\tau E \phi(\lambda) \mathcal{Q}_{n,\lambda} + 2 E F_\tau \lambda^{1/2} \mathcal{S}_{n,\lambda}. \qquad (51)$$

**(3) When $\phi = \psi\vartheta$, where $\psi(\cdot)$ is non-decreasing, and $\vartheta(\cdot)$ is Lipschitz continuous with constant 1 and satisfies that $P\vartheta(A)P = \vartheta(PAP)$ for any operator $A$ and projection operator $P$.**

$$\|O_{K,n,\lambda}^{1/2} r_\lambda(O_{K,n}) P g_\lambda(C_K) \phi(C_K) C_K^{1/2}\| = \|O_{K,n,\lambda}^{1/2} r_\lambda(O_{K,n}) P g_\lambda(C_K)[\phi(C_{K,n})$$

$$+ (\vartheta(C_K) - \vartheta(C_{K,n}))\psi(C_K) + \vartheta(C_{K,n})(\psi(C_K) - \psi(C_{K,n}))] C_K^{1/2}\|$$

$$\leq \|O_{K,n,\lambda}^{1/2} r_\lambda(O_{K,n}) P g_\lambda(C_K) \phi(C_{K,n}) C_K^{1/2}\| \qquad (52)$$

$$+ \|O_{K,n,\lambda}^{1/2} r_\lambda(O_{K,n}) P g_\lambda(C_K)(\vartheta(C_K) - \vartheta(C_{K,n}))\psi(C_K) C_K^{1/2}\|$$

$$+ \|O_{K,n,\lambda}^{1/2} r_\lambda(O_{K,n}) P g_\lambda(C_K) \vartheta(C_{K,n})(\psi(C_K) - \psi(C_{K,n})) C_K^{1/2}\|$$

For the first term in the right side of (52), it is the same as (49). For the second term in the right side of (52), we have

$$\|O_{K,n,\lambda}^{1/2} r_\lambda(O_{K,n}) P g_\lambda(C_K)(\vartheta(C_K) - \vartheta(C_{K,n}))\psi(C_K) C_K^{1/2}\|$$

$$\leq \|O_{K,n,\lambda}^{1/2} r_\lambda(O_{K,n}) P g_\lambda(C_K)(\vartheta(C_K) - \vartheta(C_{K,n})) C_K^{1/2}\| \|\psi(C_K)\| \qquad (53)$$

$$\leq 2 E F_\tau \psi(\kappa^2) \lambda^{1/2} \mathcal{S}_{n,\lambda},$$

where the last inequality follows the same argument as (50) and uses the Lipschitz property of $\vartheta(\cdot)$ and the non-decreasing property of $\psi(\cdot)$ that $\|\psi(C_K)\| \leq \psi(\kappa^2)$. For the third term in the right side of (52), we have

$$\|O_{K,n,\lambda}^{1/2} r_\lambda(O_{K,n}) P g_\lambda(C_K) \vartheta(C_{K,n})(\psi(C_K) - \psi(C_{K,n})) C_K^{1/2}\|$$

$$\leq \|O_{K,n,\lambda}^{1/2} r_\lambda(O_{K,n}) P^2 g_\lambda(C_K) \vartheta(C_{K,n})(\psi(C_K) - \psi(C_{K,n})) C_K^{1/2}\|$$

$$\leq \|O_{K,n,\lambda}^{1/2} r_\lambda(O_{K,n}) P \vartheta(C_{K,n}) P\| \|\psi(C_K) - \psi(C_{K,n})\| \|C_K^{1/2} g_\lambda(C_K)\| \qquad (54)$$

$$\leq \|O_{K,n,\lambda}^{1/2} r_\lambda(O_{K,n}) \vartheta(O_{K,n})\| \|\psi(C_K) - \psi(C_{K,n})\| \|C_K^{1/2} g_\lambda(C_K)\|,$$

where the last inequality holds by $P\vartheta(C_{K,n})P = \vartheta(PC_{K,n}P) = \vartheta(O_{K,n})$. We first bound $\|O_{K,n,\lambda}^{1/2} r_\lambda(O_{K,n}) \vartheta(O_{K,n})\|$. When $t \in [0,\lambda]$, by the non-decreasing property of $\vartheta(\cdot)$ and (13), we have

$$|r_\lambda(t)| \vartheta(t) t^{1/2} \leq \vartheta(\lambda) |r_\lambda(t)| t^{1/2} \leq F_\tau \vartheta(\lambda) \lambda^{1/2}.$$

When $t \in [\lambda, \kappa^2]$, note that

$$c' \frac{\lambda^\tau}{\vartheta(\lambda)\lambda^{1/2}} \leq \inf_{t \in [\lambda, \kappa^2]} \frac{t^\tau}{\vartheta(t)t^{1/2}}, \quad \forall \lambda \in [0, \kappa^2].$$

Thus we have

$$|r_\lambda(t)|\vartheta(t)t^{1/2} \leq c'^{-1}\vartheta(\lambda)\lambda^{1/2-\tau}|r_\lambda(t)|t^\tau \leq c'^{-1}F_\tau\vartheta(\lambda)\lambda^{1/2},$$

where the second inequality uses the property of the filter function in (13). By combining the above two cases, we can obtain that

$$\|O_{K,n,\lambda}^{1/2}r_\lambda(O_{K,n})\vartheta(O_{K,n})\| \leq \sup_{t \in [0,\kappa^2]} |r_\lambda(t)\vartheta(t)(t+\lambda)^{1/2}| \leq 2c'_\tau\vartheta(\lambda)\lambda^{1/2}, \tag{55}$$

where $c'_\tau = F_\tau / \min(1, c')$.

Next we bound $\|\psi(C_K) - \psi(C_{K,n})\|$. From Lemma 28, there exists some $c_\psi < \infty$ such that

$$\|\psi(C_K) - \psi(C_{K,n})\| \leq c_\psi\psi(\|C_K - C_{K,n}\|) = c_\psi\psi(\mathcal{U}_{n,\lambda}). \tag{56}$$

For the last term $\|C_K^{1/2}g_\lambda(C_K)\|$, from (13), we have

$$\|C_K^{1/2}g_\lambda(C_K)\| \leq \sup_{t \in [0,\kappa^2]} |g_\lambda(t)t^{1/2}| \leq E\lambda^{-1/2}. \tag{57}$$

Plugging (55), (56) and (57) into (54), we have

$$\|O_{K,n,\lambda}^{1/2}r_\lambda(O_{K,n})Pg_\lambda(C_K)\vartheta(C_{K,n})(\psi(C_K) - \psi(C_{K,n}))C_K^{1/2}\| \leq 2c'_\tau c_\psi E\vartheta(\lambda)\psi(\mathcal{U}_{n,\lambda}). \tag{58}$$

Combining (49), (53) and (58) and plugging into (52), we have

$$\begin{aligned} &\|O_{K,n,\lambda}^{1/2}r_\lambda(O_{K,n})Pg_\lambda(C_K)\phi(C_K)C_K^{1/2}\| \\ &\leq 2c_\tau E\phi(\lambda)\mathcal{Q}_{n,\lambda} + 2EF_\tau\psi(\kappa^2)\lambda^{1/2}\mathcal{S}_{n,\lambda} + 2c'_\tau c_\psi E\vartheta(\lambda)\psi(\mathcal{U}_{n,\lambda}). \end{aligned} \tag{59}$$

Thus we complete the proof by combining (47), (51) and (59).

$\square$

Now we give the error bound of the randomized projection operator terms. Let

$$\widehat{\mathcal{N}}(\lambda) = \text{Tr}(C_{K,n}C_{K,n,\lambda}^{-1})$$

be the empirical average effective dimension.

**Lemma 19.** *Under Assumption 2, given a fixed input set $\{x_1, \ldots, x_n\}$, if $n \gtrsim \kappa^2 \log \frac{\kappa^2}{\delta}$, and $\frac{1}{n}\log\frac{n}{\delta} \lesssim \lambda < \kappa^2$, then for any $\delta \in (0,1)$, with probability at least $1 - \delta$, there holds*

$$\widehat{\mathcal{N}}(\lambda) = Tr(C_{K,n}C_{K,n,\lambda}^{-1}) \lesssim \lambda^{-\frac{1}{\beta}}. \tag{60}$$

*Moreover, there exists a subset $U_n \in \mathbb{R}^{m \times n}$ such that for any $\mathbf{G} \in U_n$, if the projection dimension satisfies that*

$$m \gtrsim \lambda^{-\frac{1}{\beta}}\log\frac{3}{\delta}, \tag{61}$$

*then with probability at least $1 - \delta$, there holds*

$$\mathcal{A}_{K,n,P} = \|C_{K,n}^{1/2}(I - P)\| \lesssim m^{-\frac{\beta}{2}}. \tag{62}$$

*Proof.* From Proposition 1 in Rudi et al. (2015), if $n \gtrsim \kappa^2 \log \frac{\kappa^2}{\delta}$, and $\frac{1}{n}\log\frac{n}{\delta} \lesssim \lambda < \kappa^2$, then with probability at least $1 - \delta$, there holds

$$\frac{|\widehat{\mathcal{N}}(\lambda) - \mathcal{N}(\lambda)|}{\mathcal{N}(\lambda)} \leq 1.65.$$

Thus we can obtain (60) by combining the above inequality and Assumption 2.

The proof of (61) and (62) is a direct result of Lemma 17 in Lin & Cevher (2020a) given that (61) holds. Thus we complete the proof. $\square$

**Lemma 20** (**Projection Error for Randomized Sketches**). *Under Assumption 2, let* $\mathbf{G} \in \mathbb{R}^{m \times n}$ *is a random matrix satisfying* (15)*, and* $P$ *be the projection operator with its range being the closure of the linear span of* $\{S^*_{K,n}\mathbf{G}^\top\}$*. If* $n \gtrsim \kappa^2 \log \frac{\kappa^2}{\delta} \vee \lambda^{-\alpha} \log(4/\delta)$*, and* $\frac{1}{n} \log \frac{n}{\delta} \lesssim \lambda < \kappa^2$*, the projection dimension satisfies that*

$$m \gtrsim \lambda^{-\frac{1}{\beta}} \log \frac{6}{\delta}, \tag{63}$$

*then for any* $\delta \in (0,1)$*, with probability at least* $1 - \delta$*, there holds*

$$\mathcal{A}_{K,P} = \|C_K^{1/2}(I - P)\| \lesssim m^{-\frac{\beta}{2}}. \tag{64}$$

*Proof.* Note that

$$
\begin{aligned}
\mathcal{A}_{K,P} = \|C_K^{1/2}(I - P)\| &\leq \|C_{K,\lambda}^{1/2}(I - P)\| \\
&\leq \|C_{K,\lambda}^{1/2} C_{K,n,\lambda}^{-1/2}\| \|C_{K,n,\lambda}^{1/2}(I - P)\| \leq \mathcal{Q}_{n,\lambda} \|C_{K,n,\lambda}^{1/2}(I - P)\|,
\end{aligned} \tag{65}
$$

And for any $f \in \mathcal{H}_K$ and $\|f\|_{\mathcal{H}} = 1$,

$$
\begin{aligned}
\|C_{K,n,\lambda}^{1/2}(I - P)f\|_{\mathcal{H}}^2 = \langle C_{K,n,\lambda}(I - P)f, (I - P)f \rangle_{\mathcal{H}} &= \||C_{K,n}^{1/2}(I - P)f\|_{\mathcal{H}}^2 + \lambda\|(I - P)f\|_{\mathcal{H}}^2 \\
&\leq \mathcal{A}_{K,n,P}^2 + \lambda.
\end{aligned}
$$

From Lemma 13, if $n \gtrsim \lambda^{-\alpha} \log(4/\delta)$, then with probability at least $1 - \delta/2$, there holds $\mathcal{Q}_{n,\lambda} \leq \sqrt{2}$. Combining the above inequality, Lemma 19 and (65), with probability at least $1 - \delta$, we have

$$\mathcal{A}_{K,P} \leq \sqrt{2}(\mathcal{A}_{K,n,P}^2 + \lambda)^{1/2} \lesssim \lambda^{1/2},$$

provided that $m \gtrsim \lambda^{-\frac{1}{\beta}} \log \frac{6}{\delta}$. Thus we complete the proof. $\qquad\square$

**Lemma 21** (**Projection Error for Plain Nyström**). *Under Assumption 2, let* $P$ *be the projection operator with its range being the closure of the linear span of* $\{K(\boldsymbol{x}_{i_j}, \cdot)\}_{i=1}^m$*, where* $\{i_j\}_{j=1}^m$ *is a subset of* $\{1, 2, \ldots, n\}$ *with cardinality* $m$ *selected uniformly at random without replacement. If the projection dimension satisfies that*

$$m \gtrsim \log \frac{4\kappa^2}{\lambda\delta} \vee \lambda^{-\alpha} \log(4\kappa^2/\lambda\delta), \tag{66}$$

*then for any* $\delta \in (0,1)$*, with probability at least* $1 - \delta$*, there holds*

$$\mathcal{A}_{K,P} = \|C_K^{1/2}(I - P)\| \lesssim m^{-\frac{1}{2\alpha}}. \tag{67}$$

**Lemma 22** (**Projection Error for ALS Nyström**). *Under Assumption 2, let* $P$ *be the projection operator with its range being the closure of the linear span of* $\{K(\widetilde{\boldsymbol{x}}_{i_j}, \cdot)\}_{i=1}^m$*, where* $\{\widetilde{\boldsymbol{x}}_{i_j}\}_{j=1}^m$ *is selected by ALS Nyström sub-sampling method. For any* $\delta \in (0,1)$*, with probability at least* $1 - \delta$*, there holds*

$$\mathcal{A}_{K,P} = \|C_K^{1/2}(I - P)\| \lesssim m^{-\frac{\beta}{2}}, \tag{68}$$

*when the following conditions are satisfied:*

1. *There exists a* $L \geq 1$ *and a* $\lambda_0 > 0$ *such that* $\{\hat{l}_i(t)\}_{i=1}^n$ *are* $L$*-approximate leverage scores for any* $t \geq \lambda_0$*;*

2. $n \gtrsim \kappa^2 + \kappa^2 \log \frac{2\kappa^2}{\delta}$*;*

3. $\lambda_0 \vee \frac{19\kappa^2}{n} \log \frac{2n}{\delta} \leq \lambda \leq \kappa^2$*;*

4. $m \gtrsim \log \frac{8n}{\delta} \vee \lambda^{-\frac{1}{\beta}} \log \frac{8n}{8}$*.*

The proof of Lemma 21 and 22 can be derived with a similar argument as Lemmas 20 and 21 in Lin & Cevher (2020a) and Lemmas 6 and 7 in Rudi et al. (2015).

**Remark 6.** *Ignore the log factor terms, for randomized sketches and ALS Nyström methods, Lemmas 20 and 22 show that a sharp bound on $\mathcal{A}_{K,P}$ holds provided $m \gtrsim \lambda^{-\frac{1}{\beta}}$. For the plain Nyström method, Lemma 21 gives the corresponding condition $m \gtrsim \lambda^{-\frac{1}{\alpha}}$. This is from Assumption 2, the average effective dimension satisfies $\mathcal{N}(\lambda) \lesssim \lambda^{-1/\beta}$, and by Assumption 3 and Lemma 11, the maximal effective dimension satisfies $\mathcal{N}_\infty(\lambda) \lesssim \lambda^{-1/\alpha}$. The conditions above actually can be written as $m \gtrsim \mathcal{N}(\lambda)$ for randomized sketches and ALS Nyström; $m \gtrsim \mathcal{N}_\infty(\lambda)$ for plain Nyström. Moreover, in the proof of Corollary 3, we explicitly obtain the optimal rates by choosing $m$ so that the projection term is no larger than the bias and variance terms under these conditions. Thus, the minimal projection dimension is proportional to the (average or maximal) effective dimension to guarantee the optimal rates.*

### E.5    PROOF OF THEOREM 2

*Proof.* Suppose that the conditions in Theorem 2 hold.

**Approximation Error Term.** From Proposition 1, we have

$$\|f_\lambda - f_\rho\|_{[\mathcal{H}]_\rho^\gamma} \lesssim \lambda^{-\frac{\gamma}{2}} \phi(\lambda).$$

**Random Projection Error Term.** From Proposition 3, we have

$$\|(I - P)f_\lambda\|_{[\mathcal{H}]_\rho^\gamma} \lesssim \begin{cases} \mathcal{A}_{K,P}^{2-\gamma} \lambda^{\frac{s}{2}-1}, & \text{if } s < 2; \\ \mathcal{A}_{K,P}^{2-\gamma}, & \text{if } s \geq 2. \end{cases}$$

**Empirical Error Term.** From Proposition 4 and Lemma 12-17, we have

$$\|f_{n,\lambda}^{rp} - f_\lambda\|_{[\mathcal{H}]_\rho^\gamma}$$
$$\lesssim \begin{cases} \lambda^{-\frac{\gamma}{2}} \left[ \dfrac{\lambda^{\min\{0, \frac{s-\alpha}{2}\}+1}}{n\lambda^{\frac{\alpha}{2}}} + \dfrac{\phi(\lambda)\lambda^{-\frac{\alpha}{2}} + \lambda^{-\frac{1}{2\beta}}}{\sqrt{n}} + \phi(\lambda) + \lambda^{\frac{s}{2}-1}(\mathcal{A}_{K,P}^2 + \lambda^{\frac{1}{2}}\mathcal{A}_{K,P}) + \mathcal{T}_{n,\lambda} \right] \log\frac{C}{\delta}, & \text{if } s < 2; \\ \lambda^{-\frac{\gamma}{2}} \left[ \dfrac{\lambda^{\min\{0, \frac{s-\alpha}{2}\}+1}}{n\lambda^{\frac{\alpha}{2}}} + \dfrac{\phi(\lambda)\lambda^{-\frac{\alpha}{2}} + \lambda^{-\frac{1}{2\beta}}}{\sqrt{n}} + \phi(\lambda) + \mathcal{A}_{K,P}^2 + n^{-\frac{1}{4}}\mathcal{A}_{K,P} + \mathcal{T}_{n,\lambda} \right] \log\frac{C}{\delta}, & \text{if } s \geq 2. \end{cases}$$

To plug in the bounds of $\mathcal{T}_{n,\lambda}$, we consider three cases of $\phi(\cdot)$ separately, and combine the above three error terms to prove the desired results. For ease of notation, we denote the term related to random projection error term $\mathcal{A}_{K,P}$ as

$$\Delta_1 = \lambda^{\frac{s}{2}-1}(\mathcal{A}_{K,P}^2 + \lambda^{\frac{1}{2}}\mathcal{A}_{K,P} + \lambda^{\frac{\gamma}{2}}\mathcal{A}_{K,P}^{2-\gamma}),$$
$$\Delta_2 = \mathcal{A}_{K,P}^2 + n^{-\frac{1}{4}}\mathcal{A}_{K,P} + \lambda^{\frac{\gamma}{2}}\mathcal{A}_{K,P}^{2-\gamma}.$$

**(1)** $\phi(\cdot)$ **is non-decreasing.**

When $s < 2$, combining Proposition 1-4, and the similarly bounds in Lemma 12-18, with probability at least $1 - \delta$, we have

$$\|f_{n,\lambda}^{rp} - f_\rho\|_{[\mathcal{H}]_\rho^\gamma}$$
$$\lesssim \lambda^{-\frac{\gamma}{2}} \left[ \frac{\lambda^{\min\{0, \frac{s-\alpha}{2}\}} + 1}{n\lambda^{\frac{\alpha}{2}}} + \frac{\phi(\lambda)\lambda^{-\frac{\alpha}{2}} + \lambda^{-\frac{1}{2\beta}}}{\sqrt{n}} + \phi(\lambda) + \Delta_1 \right] \log\frac{4}{\delta} \tag{69}$$
$$\lesssim \lambda^{-\frac{\gamma}{2}} \left[ \frac{1}{n\lambda^{\max\{\frac{\alpha}{2}, \alpha-\frac{s}{2}\}}} + \frac{1}{n\lambda^{\frac{\alpha}{2}}} + \frac{1}{\sqrt{n}\lambda^{\frac{1}{2\beta}}} + \phi(\lambda)\left(\frac{1}{\sqrt{n}\lambda^{\frac{\alpha}{2}}} + 1\right) + \Delta_1 \right] \log\frac{4}{\delta}.$$

When $s \geq 2$, combining Proposition 1-4, and the similarly bounds in Lemma 12-18, with probability at least $1 - \delta$, we have

$$\|f_{n,\lambda}^{rp} - f_\rho\|_{[\mathcal{H}]_\rho^\gamma}$$
$$\lesssim \lambda^{-\frac{\gamma}{2}} \left[ \frac{\lambda^{\min\{0, \frac{s-\alpha}{2}\}} + 1}{n\lambda^{\frac{\alpha}{2}}} + \frac{\phi(\lambda)\lambda^{-\frac{\alpha}{2}} + \lambda^{-\frac{1}{2\beta}}}{\sqrt{n}} + \phi(\lambda) + \Delta_2 \right] \log\frac{6}{\delta} \tag{70}$$
$$= \lambda^{-\frac{\gamma}{2}} \left[ \frac{1}{n\lambda^{\max\{\frac{\alpha}{2}, \alpha-\frac{s}{2}\}}} + \frac{1}{n\lambda^{\frac{\alpha}{2}}} + \frac{1}{\sqrt{n}\lambda^{\frac{1}{2\beta}}} + \phi(\lambda)\left(\frac{1}{\sqrt{n}\lambda^{\frac{\alpha}{2}}} + 1\right) + \Delta_2 \right] \log\frac{6}{\delta}.$$

**(2) $\phi(\cdot)$ is Lipschitz continuous with constant 1 and satisfies that $P\phi(A)P = \phi(PAP)$ for any operator $A$ and projection operator $P$.**

When $s < 2$, combining Proposition 1-4, and the similarly bounds in Lemma 12-22, with probability at least $1 - \delta$, we have

$$\|f_{n,\lambda}^{rp} - f_\rho\|_{[\mathcal{H}]_\rho^\gamma}$$

$$\lesssim \lambda^{-\frac{\gamma}{2}} \left[ \frac{\lambda^{\min\{0, \frac{s-\alpha}{2}\}} + 1}{n\lambda^{\frac{\alpha}{2}}} + \frac{\phi(\lambda)\lambda^{-\frac{\alpha}{2}} + \lambda^{-\frac{1}{2\beta}}}{\sqrt{n}} + \frac{\lambda^{1/2}}{n\lambda^{\frac{\alpha}{2}}} + \frac{\lambda^{1/2}}{\sqrt{n}\lambda^{\frac{1}{2\beta}}} + \phi(\lambda) + \Delta_1 \right] \log \frac{6}{\delta} \tag{71}$$

$$\lesssim \lambda^{-\frac{\gamma}{2}} \left[ \frac{1}{n\lambda^{\max\{\frac{\alpha}{2}, \alpha - \frac{s}{2}\}}} + \frac{1}{n\lambda^{\frac{\alpha}{2}}} + \frac{1}{\sqrt{n}\lambda^{\frac{1}{2\beta}}} + \phi(\lambda)\left( \frac{1}{\sqrt{n}\lambda^{\frac{\alpha}{2}}} + 1 \right) + \Delta_1 \right] \log \frac{6}{\delta}.$$

When $s \geq 2$, combining Proposition 1-4, and the similarly bounds in Lemma 12-22, with probability at least $1 - \delta$, we have

$$\|f_{n,\lambda}^{rp} - f_\rho\|_{[\mathcal{H}]_\rho^\gamma}$$

$$\lesssim \lambda^{-\frac{\gamma}{2}} \left[ \frac{\lambda^{\min\{0, \frac{s-\alpha}{2}\}} + 1}{n\lambda^{\frac{\alpha}{2}}} + \frac{\phi(\lambda)\lambda^{-\frac{\alpha}{2}} + \lambda^{-\frac{1}{2\beta}}}{\sqrt{n}} + \frac{\lambda^{1/2}}{n\lambda^{\frac{\alpha}{2}}} + \frac{\lambda^{1/2}}{\sqrt{n}\lambda^{\frac{1}{2\beta}}} + \phi(\lambda) + \Delta_2 \right] \log \frac{8}{\delta} \tag{72}$$

$$\lesssim \lambda^{-\frac{\gamma}{2}} \left[ \frac{1}{n\lambda^{\max\{\frac{\alpha}{2}, \alpha - \frac{s}{2}\}}} + \frac{1}{n\lambda^{\frac{\alpha}{2}}} + \frac{1}{\sqrt{n}\lambda^{\frac{1}{2\beta}}} + \phi(\lambda)\left( \frac{1}{\sqrt{n}\lambda^{\frac{\alpha}{2}}} + 1 \right) + \Delta_2 \right] \log \frac{8}{\delta}.$$

**(3) $\phi = \psi\vartheta$, where $\psi(\cdot)$ is non-decreasing, and $\vartheta(\cdot)$ is Lipschitz continuous with constant 1 and satisfies that $P\vartheta(A)P = \vartheta(PAP)$ for any operator $A$ and projection operator $P$.**

When $s < 2$, combining Proposition 1-4, and the similarly bounds in Lemma 12-22, with probability at least $1 - \delta$, we have

$$\|f_{n,\lambda}^{rp} - f_\rho\|_{[\mathcal{H}]_\rho^\gamma}$$

$$\lesssim \lambda^{-\frac{\gamma}{2}} \left[ \frac{\lambda^{\min\{0, \frac{s-\alpha}{2}\}} + 1}{n\lambda^{\frac{\alpha}{2}}} + \frac{\phi(\lambda)\lambda^{-\frac{\alpha}{2}} + \lambda^{-\frac{1}{2\beta}}}{\sqrt{n}} + \frac{\lambda^{1/2}}{n\lambda^{\frac{\alpha}{2}}} + \frac{\lambda^{1/2}}{\sqrt{n}\lambda^{\frac{1}{2\beta}}} + \phi(\lambda) + \vartheta(\lambda)\psi(\mathcal{U}_{n,\lambda}) + \Delta_1 \right] \log \frac{8}{\delta}$$

$$\lesssim \lambda^{-\frac{\gamma}{2}} \left[ \frac{1}{n\lambda^{\max\{\frac{\alpha}{2}, \alpha - \frac{s}{2}\}}} + \frac{1}{n\lambda^{\frac{\alpha}{2}}} + \frac{1}{\sqrt{n}\lambda^{\frac{1}{2\beta}}} + \phi(\lambda)\left( \frac{1}{\sqrt{n}\lambda^{\frac{\alpha}{2}}} + 1 \right) + \vartheta(\lambda)\psi(\frac{1}{\sqrt{n}}) + \Delta_1 \right] \log \frac{8}{\delta}. \tag{73}$$

When $s \geq 2$, combining Proposition 1-4, and the similarly bounds in Lemma 12-22, with probability at least $1 - \delta$, we have

$$\|f_{n,\lambda}^{rp} - f_\rho\|_{[\mathcal{H}]_\rho^\gamma}$$

$$\lesssim \lambda^{-\frac{\gamma}{2}} \left[ \frac{\lambda^{\min\{0, \frac{s-\alpha}{2}\}} + 1}{n\lambda^{\frac{\alpha}{2}}} + \frac{\phi(\lambda)\lambda^{-\frac{\alpha}{2}} + \lambda^{-\frac{1}{2\beta}}}{\sqrt{n}} + \frac{\lambda^{1/2}}{n\lambda^{\frac{\alpha}{2}}} + \frac{\lambda^{1/2}}{\sqrt{n}\lambda^{\frac{1}{2\beta}}} + \phi(\lambda) + \vartheta(\lambda)\psi(\mathcal{U}_{n,\lambda}) + \Delta_2 \right] \log \frac{8}{\delta}$$

$$\lesssim \lambda^{-\frac{\gamma}{2}} \left[ \frac{1}{n\lambda^{\max\{\frac{\alpha}{2}, \alpha - \frac{s}{2}\}}} + \frac{1}{n\lambda^{\frac{\alpha}{2}}} + \frac{1}{\sqrt{n}\lambda^{\frac{1}{2\beta}}} + \phi(\lambda)\left( \frac{1}{\sqrt{n}\lambda^{\frac{\alpha}{2}}} + 1 \right) + \vartheta(\lambda)\psi(\frac{1}{\sqrt{n}}) + \Delta_2 \right] \log \frac{8}{\delta}. \tag{74}$$

Thus we complete the proof by combining (69)-(74).

$\square$

### E.6 Proof of Theorem 3

*Proof.* By the choice $\lambda = \Phi^{-1}(n^{-1})$ with $\Phi(t) = (\phi(t)/\phi(1))^2 t^{\frac{1}{\beta}}$, we have

$$\left(\frac{\lambda^{\frac{1}{2\beta}}\phi(\lambda)}{\phi(1)}\right)^2 = \frac{1}{n} \implies \lambda = \left(\frac{\phi(1)}{\sqrt{n}\phi(\lambda)\lambda^{-\frac{s}{2}}}\right)^{\frac{2\beta}{s\beta+1}}$$

We consider three cases of $\phi(\cdot)$. In this case, $\Phi(\cdot)$ is non-decreasing. Note that $\Phi(0) = 0$ and $\Phi(1) = 1$, thus $\lambda \in [0, 1]$. Since $\phi(t)t^{-s/2}$ is non-decreasing, let $\lambda = 1$, thus we have

$$\lambda = \left(\frac{\phi(1)}{\sqrt{n}\phi(\lambda)\lambda^{-\frac{s}{2}}}\right)^{\frac{2\beta}{s\beta+1}} \geq n^{-\frac{\beta}{s\beta+1}}.$$

Note that $s \geq \alpha - \frac{1}{\beta}$ and $\lambda \in [n^{-\frac{\beta}{s\beta+1}}, 1]$,

$$\frac{1}{\sqrt{n}\lambda^{\frac{\alpha}{2}}} \leq \frac{1}{\sqrt{n \cdot n^{\frac{\alpha\beta}{s\beta+1}}}} \leq 1.$$

When $s \geq \alpha$, we have $\lambda^{\max\{\frac{\alpha}{2}, \alpha-\frac{s}{2}\}} = \lambda^{\frac{\alpha}{2}}$, and

$$\frac{1}{n\lambda^{\frac{\alpha}{2}}} \leq \frac{1}{\sqrt{n}} \leq \frac{1}{\sqrt{n\lambda^{\frac{1}{\beta}}}} = \frac{\phi(\lambda)}{\phi(1)}.$$

When $r < \alpha/2$, we have $\lambda^{\max\{\frac{\alpha}{2}, \alpha-\frac{s}{2}\}} = \lambda^{\alpha-\frac{s}{2}}$, and

$$\frac{1}{n\lambda^{\alpha-\frac{s}{2}}} \leq \frac{1}{\sqrt{n\lambda^{\beta}}} = \frac{\phi(\lambda)}{\phi(1)}.$$

For case (3), since $\psi(t)$ is non-decreasing, and note that $\lambda \geq n^{-1/2}$, then

$$\psi(\frac{1}{\sqrt{n}})\vartheta(\lambda) \leq \psi(\lambda)\vartheta(\lambda) = \phi(\lambda).$$

Combining the above results and the three cases in Theorem 2, we can obtain that

$$\|f_{n,\lambda}^{rp} - f_\rho\|_{[\mathcal{H}]_\rho^\gamma}^2 \lesssim \lambda^{-\gamma}(\phi(\lambda) + \Delta)^2 \log^2 \frac{8}{\delta}, \tag{75}$$

where $\Delta = \begin{cases} \Delta_1, & \text{if } s < 2; \\ \Delta_2, & \text{if } s \geq 2. \end{cases}$

Thus we complete the proof. $\square$

### E.7 Proof of Corollary 2

*Proof.* Note that $\phi(t) = \kappa^{-(s-2)_+} t^{\frac{s}{2}}$, when $s < 2$, then $\phi$ is non-decreasing, when $s \geq 2$, then $\phi$ is Lipschitz continuous with constant 1 over $[0, \kappa^2]$, thus we can apply Theorem 2. We focus on three cases of $s$.

**(1) When $s < \alpha - \frac{1}{\beta}$, and thus $s < 2$,** let $\lambda = n^{-\frac{1}{\alpha}}$, we need to bound $\Delta_1$.

Let

$$m \gtrsim \begin{cases} n^{\frac{1}{\alpha\beta}}, & \text{for Randomized Sketches and ALS Nyström;} \\ n, & \text{for Plain Nyström.} \end{cases}$$

Then according to Lemma 20-22, with probability at least $1 - \delta/5$, we have $\mathcal{A}_{K,P} \lesssim n^{-\frac{1}{2\alpha}}$. Thus we have

$$\Delta_1 = \lambda^{\frac{s}{2}-1}(\mathcal{A}_{K,P}^2 + \lambda^{\frac{1}{2}}\mathcal{A}_{K,P} + \lambda^{\frac{\gamma}{2}}\mathcal{A}_{K,P}^{2-\gamma}) \lesssim n^{-\frac{s}{2\alpha}}.$$

Combining the above result and (75), we have

$$\|f_{n,\lambda}^{rp} - f_\rho\|_{[\mathcal{H}]_\rho^\gamma}^2 \lesssim n^{-\frac{s-\gamma}{\alpha}} \log^2 \frac{10}{\delta}.$$

**(2) When $s \geq \alpha - \frac{1}{\beta}$ and $s < 2$,** let $\lambda = n^{-\frac{\beta}{s\beta+1}}$, we need to bound $\Delta_1$.

Let

$$m \gtrsim \begin{cases} n^{\frac{1}{s\beta+1}}, & \text{for Randomized Sketches and ALS Nyström;} \\ n^{\frac{\alpha\beta}{s\beta+1}}, & \text{for Plain Nyström.} \end{cases}$$

Then according to Lemma 20-22, with probability at least $1 - \delta/5$, we have $\mathcal{A}_{K,P} \lesssim n^{-\frac{\beta}{2(s\beta+1)}}$. Thus we have

$$\Delta_1 = \lambda^{\frac{s}{2}-1}(\mathcal{A}_{K,P}^2 + \lambda^{\frac{1}{2}}\mathcal{A}_{K,P} + \lambda^{\frac{\gamma}{2}}\mathcal{A}_{K,P}^{2-\gamma}) \lesssim n^{-\frac{s\beta}{2(s\beta+1)}}.$$

Combining the above result and (75), we have

$$\|f_{n,\lambda}^{rp} - f_\rho\|_{[\mathcal{H}]_\rho^\gamma}^2 \lesssim n^{-\frac{(s-\gamma)\beta}{s\beta+1}} \log^2 \frac{10}{\delta}.$$

**(3) When $s \geq 2$,** let $\lambda = n^{-\frac{\beta}{s\beta+1}}$, we need to bound $\Delta_1$.

Let

$$m \gtrsim \begin{cases} n^{\frac{s-\gamma}{(2-\gamma)(s\beta+1)}}, & \text{for Randomized Sketches and ALS Nyström;} \\ n^{\frac{\alpha\beta(s-\gamma)}{(2-\gamma)(s\beta+1)}}, & \text{for Plain Nyström.} \end{cases}$$

Then according to Lemma 20-22, with probability at least $1 - \delta/5$, we have $\mathcal{A}_{K,P} \lesssim n^{-\frac{\beta(s-\gamma)}{(4-2\gamma)(s\beta+1)}}$. We prove the first two terms in $\Delta_2$ are dominated by the last term. Note that

$$\mathcal{A}_{K,P}^2 = n^{-\frac{\beta(s-\gamma)}{(2-\gamma)(s\beta+1)}} = n^{-\frac{s\beta}{2(s\beta+1)}} n^{-\frac{\beta(s-2)}{2(2-\gamma)(s\beta+1)}} \leq n^{-\frac{s\beta}{2(s\beta+1)}} = \lambda^{\frac{\gamma}{2}}\mathcal{A}_{K,P}^{2-\gamma},$$

where the inequality holds since $s \geq 2$ and $\gamma \leq 1$. On the other hand,

$$
\begin{aligned}
n^{-\frac{1}{4}}\mathcal{A}_{K,P} = n^{-\frac{1}{4}-\frac{\beta(s-\gamma)}{(4-2\gamma)(s\beta+1)}} &= n^{-\frac{s\beta}{2(s\beta+1)}} n^{-\left(\frac{1}{4}+\frac{s\beta}{2(s\beta+1)}-\frac{\beta(s-\gamma)}{(4-2\gamma)(s\beta+1)}\right)} \\
&= n^{-\frac{s\beta}{2(s\beta+1)}} n^{-\frac{(s-2)\beta\gamma+2-\gamma}{4(2-\gamma)(s\beta+1)}} \\
&\leq n^{-\frac{s\beta}{2(s\beta+1)}} = \lambda^{\frac{\gamma}{2}}\mathcal{A}_{K,P}^{2-\gamma},
\end{aligned}
$$

where the inequality holds since $s \geq 2$ and $\gamma \leq 1$. Thus we have

$$\Delta_2 = \mathcal{A}_{K,P}^2 + n^{-\frac{1}{4}}\mathcal{A}_{K,P} + \lambda^{\frac{\gamma}{2}}\mathcal{A}_{K,P}^{2-\gamma} \lesssim n^{-\frac{s\beta}{2(s\beta+1)}}.$$

Combining the above result and (75), we have

$$\|f_{n,\lambda}^{rp} - f_\rho\|_{[\mathcal{H}]_\rho^\gamma}^2 \lesssim n^{-\frac{(s-\gamma)\beta}{s\beta+1}} \log^2 \frac{10}{\delta}.$$

Finally, we verify the conditions on $n$. The first restriction on $n$ arises from Lemma 13 and 19–22.

- When $s < \alpha - \frac{1}{\beta}$, with $\lambda = n^{-1/\alpha}$, we require

$$n \gtrsim \kappa^2 + \kappa^2 \log \frac{2\kappa^2}{\delta} \vee \lambda^{-\alpha},$$

which simplifies to

$$n \gtrsim \kappa^2 + \kappa^2 \log \frac{2\kappa^2}{\delta}.$$

- When $s \geq \alpha - \frac{1}{\beta}$, with $\lambda = n^{-\frac{\beta}{s\beta+1}}$, we require

$$n \gtrsim \kappa^2 + \kappa^2 \log \frac{2\kappa^2}{\delta} \vee n^{\frac{\alpha\beta}{s\beta+1}},$$

which also reduces to

$$n \gtrsim \kappa^2 + \kappa^2 \log \frac{2\kappa^2}{\delta}$$

since $s\beta + 1 \geq \alpha\beta$.

The second restriction on $n$ arises from Lemma 17

- When $s < \alpha - \frac{1}{\beta}$, with $\phi(\lambda) \asymp \lambda^{\frac{s}{2}}$ and $\lambda = n^{-1/\alpha}$, we require

$$n \geq (\lambda^\alpha \phi(\lambda))^{-\frac{\alpha}{s+\alpha}} \gtrsim n^{\frac{s/2+\alpha}{s+\alpha}},$$

which holds since $s > 0$.

- When $s \geq \alpha - \frac{1}{\beta}$, with $\lambda = n^{-\frac{\beta}{s\beta+1}}$, we require

$$n \geq (\lambda^\alpha \phi(\lambda))^{-\frac{\alpha}{s+\alpha}} \gtrsim n^{\frac{\alpha\beta(s/2+\alpha)}{(s\beta+1)(s+\alpha)}},$$

which also reduces to

$$\alpha\beta(s/2 + \alpha) \leq (s\beta + 1)(s + \alpha) \implies \alpha(s/2 + \alpha) \leq (s + 1/\beta)(s + \alpha).$$

the latter holds since $s\beta + 1 \geq \alpha\beta$ and $s > 0$.

Combining the two restrictions, so we only need that $n$ is sufficiently large, i.e., $n \geq n_0$ for some $n_0 \geq 0$.

Thus we complete the proof.

$\square$

### E.8 AUXILIARY LEMMAS

This section collects some auxiliary lemmas that are used in the proofs of the main results.

**Lemma 23.** *For any $\lambda > 0$ and $a \in [0, 1]$, there holds*

$$\frac{\lambda^{a-1}}{2} \leq \sup_{t \geq 0} \frac{t^a}{t + \lambda} \leq \lambda^{a-1}.$$

*Proof.* In order to prove the lemma, we consider the function $f(t) = \frac{t^a}{t+\lambda}$ for $t \geq 0$. The derivative of $f(t)$ is given by

$$f'(t) = \frac{at^{a-1}(t + \lambda) - t^a}{(t + \lambda)^2} = \frac{at^{a-1}\lambda - (1 - a)t^a}{(t + \lambda)^2}.$$

For $a = 0$, we have $f'(t) = \frac{-1}{(t+\lambda)^2} < 0$, so $f(t)$ is decreasing and $\sup_{t \geq 0} f(t) = f(0) = \lambda^{a-1}$. For $a = 1$, we have $f'(t) = \frac{\lambda}{(t+\lambda)^2} > 0$, so $f(t)$ is increasing and $\sup_{t \geq 0} f(t) = \lim_{t \to \infty} f(t) = 1 = \lambda^{a-1}$. For the general case $0 < a < 1$, we have $f'(t) = 0$ if and only if $t^* = \frac{a}{1-a}\lambda$. Note that $f(0) = 0$ and $\lim_{t \to \infty} f(t) = 0$, thus $f(t)$ has a maximum at $t^*$ and

$$\sup_{t \geq 0} f(t) = f(t^*) = a^a(1 - a)^{1-a}\lambda^{a-1}.$$

Since $1/2 \leq a^a(1 - a)^{1-a} \leq 1$ for $0 < a < 1$, thus we complete the proof. $\square$

**Lemma 24** (Cordes Inequality). *Let $A$ and $B$ be positive bounded linear operators on a separable Hilbert space. Then, for any $0 < \tau \leq 1$, we have*

$$\|A^\tau B^\tau\| \leq \|AB\|^\tau.$$

**Lemma 25** (lemma E.2 in Blanchard & Krämer (2010)). *For any self-adjoint and positive semi-definite operators $A$ and $B$, if there exists some $\eta \in [0,1]$ such that*

$$\|(A + \lambda I)^{-1/2}(B - A)(A + \lambda I)^{-1/2}\| \leq 1 - \eta,$$

*then we have*

$$\|(A + \lambda I)^{1/2}(B + \lambda I)^{-1/2}\| \leq \frac{1}{\sqrt{\eta}}.$$

**Lemma 26** (Lemma 12 in Lin & Cevher (2020a)). *Let $P$ be an orthogonal projection on a Hilbert space $\mathcal{H}$, and $A$, $B$ be two self-definite positive operators on $\mathcal{H}$. Then for any $a, b \in [0, 1/2]$, there holds*

$$\|A^a(I - P)A^b\| \leq \|A - B\|^{a+b} + \|B^{1/2}(I - P)B^{1/2}\|^{a+b}.$$

**Lemma 27.** *Let $\mathcal{H}$ and $\mathcal{K}$ be two separable Hilbert spaces and $S : \mathcal{H} \to \mathcal{K}$ be a compact operator with its adjoint operator $S^*$. Then, for any piecewise continuous function $f : [0, \|L\|] \to \mathbb{R}^+$, we have*

$$f(SS^*)S = Sf(S^*S)$$

**Lemma 28** (Lemma 5.8 in Lin et al. (2020a)). *Suppose $\psi$ is an operator monotone index function on $[0, b]$, with $b > 1$. Then there is a constant $c_\psi < \infty$ depending on $b - a$, such that for any pair $B_1, B_2, \|B_1\|, \|B_2\| \leq a$, of non-negative self-adjoint operators on some Hilbert space, it holds,*

$$\|\psi(B_1) - \psi(B_2)\| \leq c_\psi \psi(\|B_1 - B_2\|)$$

*Moreover, there is $c'_\psi > 0$ such that*

$$c'_\psi \frac{\lambda}{\psi(\lambda)} \leq \frac{\sigma}{\psi(\sigma)}$$

*whenever $0 < \lambda < \sigma \leq a < b$.*

**Lemma 29** (Lemma 5.9 in Lin et al. (2020a)). *Let $h : [0, a] \to \mathbb{R}_+$ be Lipschitz continuous with constant 1 and $h(0) = 0$. Then for any pair $B_1, B_2, \|B_1\|, \|B_2\| \leq a$, of non-negative self-adjoint operators on some Hilbert space, it holds,*

$$\|h(B_1) - h(B_2)\|_{HS} \leq \|B_1 - B_2\|_{HS}$$

**Lemma 30** (Bennett inequality for sum of random vectors). *Let $z_1, \ldots, z_n$ be a sequence of i.i.d random variables on a separable Hilbert space $\mathcal{H}$, if there exists $\widetilde{\sigma}, \widetilde{B} \geq 0$ such that*

$$\mathbb{E}\|z - \mathbb{E}[z]\|_{\mathcal{H}}^p \leq \frac{1}{2}p!\widetilde{B}^{p-2}\widetilde{\sigma}^2, \quad \forall p \geq 2, \tag{76}$$

*for any $0 \leq i \leq n$, then for any $\delta \in (0,1]$, there holds*

$$\left\|\frac{1}{n}\sum_{i=1}^{n} z_i - \mathbb{E}z_i\right\|_{\mathcal{H}} \leq \frac{2\widetilde{B}\log(2/\delta)}{n} + \sqrt{\frac{2\widetilde{\sigma}^2\log(2/\delta)}{n}},$$

*with probability at least $1 - \delta$. In particular, (76) holds if*

$$\|z\|_{\mathcal{H}} \leq \widetilde{B}/2, a.s. \quad and \quad \mathbb{E}\|z\|_{\mathcal{H}}^2 \leq \widetilde{\sigma}^2$$

**Lemma 31** (Proposition 5 in Rudi & Rosasco (2017)). *Let $\mathcal{H}$ and $\mathcal{K}$ be two separable Hilbert spaces and $(v_1, z_1), \ldots, (v_n, z_n) \in \mathcal{H} \times \mathcal{K}$ for $n \geq 1$ be i.i.d. random variables such that there exists some constant $\tau$ such that $\|v\|_{\mathcal{H}} \leq \tau$ and $\|z\|_{\mathcal{H}} \leq \tau$ almost everywhere. Let $Q = \mathbb{E}v \otimes v$ and $T = \mathbb{E}v \otimes z$ and $T_n = \frac{1}{n}\sum_{i=1}^{n} v_i \otimes z_i$, then for any $\delta \in (0,1]$, the following holds with probability at least $1 - \delta$,*

$$\left\|(Q + \lambda I)^{-1/2}(T - T_n)\right\|_{HS} \leq \frac{4\tau\sqrt{\mathcal{Q}_\infty(\lambda)}\log(2/\delta)}{n} + \sqrt{\frac{4\tau^2\mathcal{Q}(\lambda)\log(2/\delta)}{n}},$$

*where $\mathcal{Q}_\infty(\lambda) = \sup_{v \in \mathcal{H}}\|(Q + \lambda I)^{-1/2}v\|^2$ and $\mathcal{Q}(\lambda) = Tr((Q + \lambda I)^{-1}Q)$.*

**Lemma 32** (Proposition 6 in Rudi & Rosasco (2017)). *Let $v_1, \ldots, v_n$ be a sequence of i.i.d random variables on a separable Hilbert spaces $\mathcal{H}$ such that $Q = \mathbb{E}v \otimes v$ is trace class, and for any $\lambda > 0$ there exists a constant $\mathcal{Q}_\infty(\lambda) < \infty$ such that $\langle v, (Q + \lambda I)^{-1}v\rangle \leq \mathcal{Q}_\infty(\lambda)$ almost everywhere. Let $Q_n = \frac{1}{n}\sum_{i=1}^{n} v_i \otimes v_i$, then for any $\delta \in (0,1]$, the following holds with probability at least $1 - \delta$,*

$$\left\|(Q + \lambda I)^{-1/2}(Q - Q_n)(Q + \lambda I)^{-1/2}\right\| \leq \frac{2\mathcal{Q}_\infty(\lambda)\log(2/\delta)}{n} + \sqrt{\frac{2\mathcal{Q}_\infty(\lambda)\log(2/\delta)}{n}}.$$

# F  NUMERICAL EXPERIMENTS

In this section, we provide some simulated examples to illustrate our theoretical findings and real-world applications to demonstrate the effectiveness of SARP.

## F.1  SIMULATED EXAMPLES

Our aim is to verify the optimal rates of SARP derived in Corollary 2. We consider three types of spectral algorithms:

(1) Kernel Ridge Regression (KRR) with $g_\lambda(t) = (t + \lambda)^{-1}$ and $\tau = 1$;

(2) Gradient Flow (GF) with $g_\lambda(t) = t^{-1}(1 - e^{-t/\lambda})$ and $\tau$ being any positive number;

(3) Spectral Cut-off (SC) with $g_\lambda(t) = t^{-1}\mathbb{I}_{[t \geq \lambda]}$ and $\tau$ being any positive number.

For the simulation setup, we suppose the input space $\mathcal{X} = [0, 1]$ and the marginal distribution $\rho_\mathcal{X}$ is uniform on $\mathcal{X}$. The kernel function is chosen as

$$K(x, x') = \min\{x, x'\}.$$

As introduced in Example 3 of Section C.3, the corresponding RKHS is the first order Sobolev space $\mathcal{H}^1([0, 1])$ with the corresponding eigenfunction-eigenvalue pairs given by

$$e_k(x) = \sqrt{2}\sin\frac{(2k-1)\pi x}{2}, \quad \mu_k = \left(\frac{2}{(2k-1)\pi}\right)^2, \quad \forall k \geq 1.$$

It is easy to verify that Assumption 2 holds with $\beta = 2$, and Section C.3 has shown that the RKHS is benign with $\alpha = 1/\beta = 0.5$ since the eigenfunctions are uniformly bounded. We set the regression function as

$$f_\rho(x) = \sum_{k=1}^{\infty} \frac{1}{k^{s+0.5}} e_k(x)$$

for some $s > 0$. The output $y$ is generated by the following model

$$y = f_\rho(x) + \varepsilon,$$

where $\varepsilon$ follows the standard Gaussian distribution.

We investigate two distinct scenarios: a mis-specified case with $s = 0.5$ and two well-specified cases with $s = 1.5$ and $s = 2.5$. For each scenario, we vary the training sample size $n$ across 10 values evenly from $\{1000, 2000, \ldots, 10000\}$. The regularization parameter is set according to the theoretical optimal form $\lambda = Cn^{-\frac{\beta}{s\beta+1}}$. For each generated dataset, we perform a grid search of 10 values for $C$ from $10^{-7}$ to $10^2$. We select the value of $C$ that gives the smallest error on an independent validation set and then report the corresponding performance. For the random projection methods, we use Gaussian sketching (GAU), ROS sketching (ROS), Nyström with uniform sampling (Ny-plain), and Nyström with approximate leverage scores (Ny-ALS) (Rudi et al., 2018). For each scenario, we select the projection dimension $m$ based on the theoretical requirements outlined in Corollary 2. Here we fix $m$ as the smallest integer that satisfies the conditions in Corollary 2 for different $(n, s, \beta, \gamma, \alpha)$ values. We further investigate the effect of projection dimension $m$ on the performance of SARP in the real applications in next section.

Suppose the solution of SARP is $f_{n,\lambda}^{rp} = \sum_{i=1}^{n} a_i K(\boldsymbol{x}_i, \cdot)$, then learning rates $\|f_{n,\lambda}^{rp} - f_\rho\|_{[\mathcal{H}]_\rho^\gamma}^2$ can be computed as

$$\|f_{n,\lambda}^{rp} - f_\rho\|_{[\mathcal{H}]_\rho^\gamma}^2 = \|L_K^{-\frac{\gamma}{2}} f_{n,\lambda}^{rp} - f_\rho\|_\rho = \left\| L_K^{-\frac{\gamma}{2}} \left( \sum_{i=1}^{n} a_i \sum_{k=1}^{\infty} \mu_k e_k(\boldsymbol{x}_i) e_k - \sum_{k=1}^{\infty} \frac{1}{k^{s+0.5}} e_k \right) \right\|_\rho^2$$

$$= \sum_{k=1}^{\infty} \left( \mu_k \sum_{i=1}^{n} a_i e_k(\boldsymbol{x}_i) - \frac{1}{k^{s+0.5}} \right)^2 \mu_k^{-\gamma}$$

$$= \sum_{k=1}^{\infty} \left( \sum_{i=1}^{n} a_i e_k(\boldsymbol{x}_i) - \frac{1}{\mu_k k^{s+0.5}} \right)^2 \mu_k^{2-\gamma}.$$

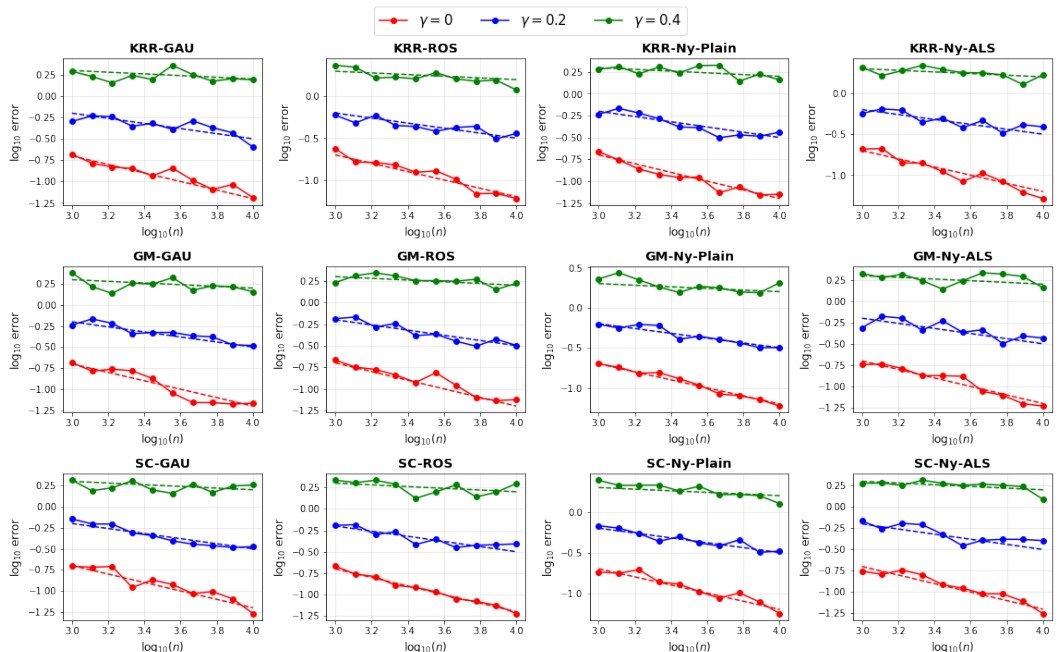

Figure 2: Log empirical learning rates of SARP with different spectral algorithms and projection methods for the mis-specified case when $s = 0.5$. The dashed lines represent the theoretical slopes. From left to right, the columns correspond to GAU, ROS, Ny-plain, and Ny-ALS, respectively. From top to bottom, the rows correspond to KRR, GF, and SC, respectively. The red, blue, and green lines represent the learning rates with $\gamma = 0, 0.2, 0.4$, respectively.

In this paper, we use the following truncation to approximate the learning rate

$$\|f_{n,\lambda}^{rp} - f_\rho\|_{[\mathcal{H}]_\rho^\gamma}^2 \approx \sum_{k=1}^{T} \left( \sum_{i=1}^{n} a_i e_k(\boldsymbol{x}_i) - \frac{1}{\mu_k k^{s+0.5}} \right)^2 \mu_k^{2-\gamma}. \tag{77}$$

Here we set $T = 10000$. For the mis-specified case where $s = 0.5$, we evaluate the soblev learning rates using $\gamma \in \{0, 0.2, 0.4\}$; for the well-specified case ($s = 1.5$ and $s = 2.5$), we use $\gamma \in \{0, 0.4, 1\}$. The theoretical slope for the true learning rate is given by $-\frac{(s-\gamma)\beta}{s\beta+1}$. To empirically validate the learning rates derived in Corollary 2, we plot the test error against the training sample size $n$ on a log-log scale. All the results are averaged over 50 independent trials.

As shown in Figure 2, 3, and 4, the empirical learning rates of SARP align well with the theoretical findings in Corollary 2 for both mis-specified and well-specified cases. Since the theoretical learning rates are independent of the choice of spectral algorithms and projection methods once the number of projection dimensions is chosen optimally, the empirical results also exhibit this consistency. For different Sobolev norms, we can see that the learning curves flatten as $\gamma$ increases, and the $L_{\rho_\mathcal{X}}^2$ error with $\gamma = 0$ converges the fastest, which is consistent with our theoretical findings.

We note that the regression function $f_\rho$ is not uniformly bounded when $s = 0.5$. Indeed, for $k = 4a + 1$, $a \in \mathbb{Z}$, we have $e_k(1/2) = 1$, which implies

$$f_\rho(1/2) = \sum_{k=1}^{\infty} \frac{1}{k} e_k(1/2) \geq \sum_{a=0}^{\infty} \frac{1}{(4a+1)} = \infty.$$

This example illustrates that the assumption of uniform boundedness of $f_\rho$ is not necessary for deriving the optimal learning rates of SARP.

**Remark 7.** *In real applications, where the parameters $\beta$ and $s$ are unknown, one can choose both $\lambda$ and $m$ by grid search on an independent validation set or by cross-validation. This type of tuning procedure is standard in the kernel methods and random projection literature (Rudi et al., 2015; Yang et al., 2017).*

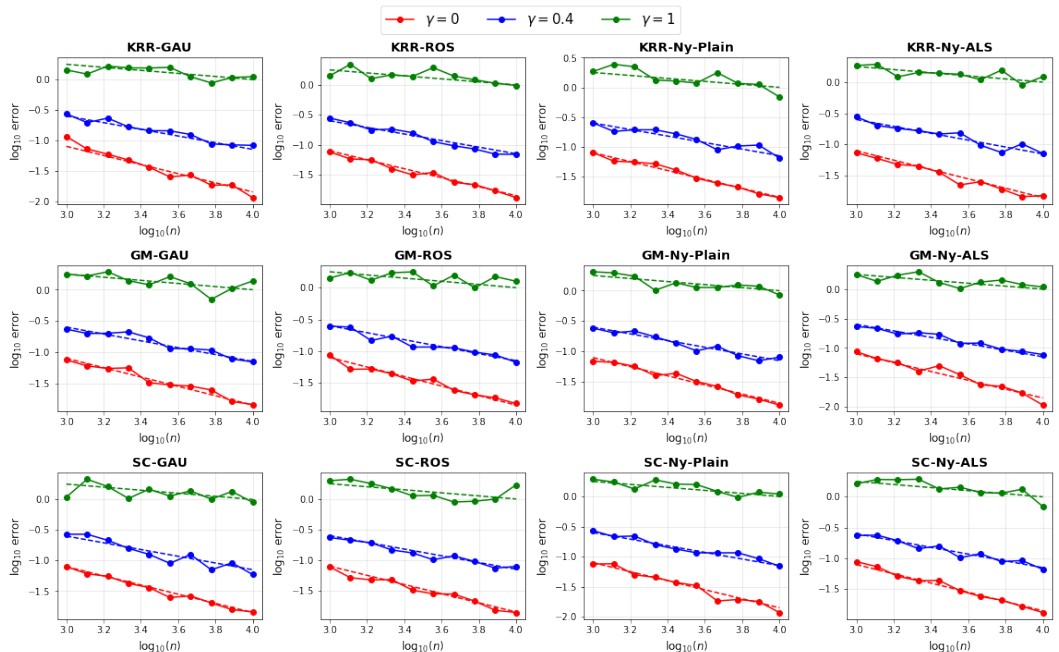

Figure 3: Log empirical learning rates of SARP with different spectral algorithms and projection methods for the well-specified case when $s = 1.5$. The dashed lines represent the theoretical slopes. From left to right, the columns correspond to GAU, ROS, Ny-plain, and Ny-ALS, respectively. From top to bottom, the rows correspond to KRR, GF, and SC, respectively. The red, blue, and green lines represent the learning rates with $\gamma = 0, 0.4, 1$, respectively.

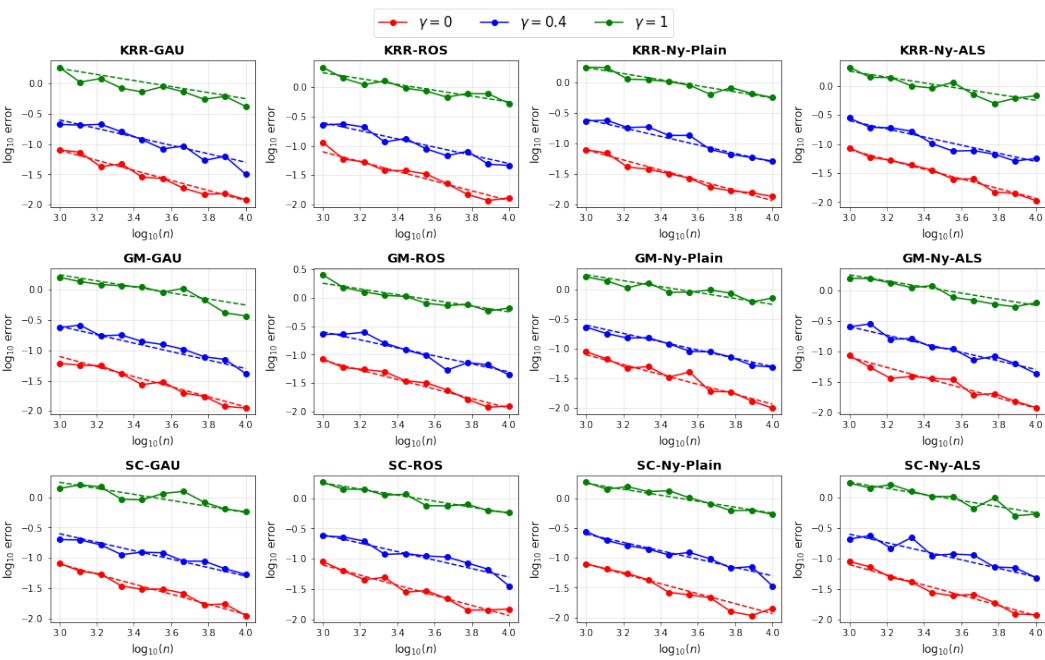

Figure 4: Log empirical learning rates of SARP with different spectral algorithms and projection methods for the well-specified case when $s = 2.5$. The dashed lines represent the theoretical slopes. From left to right, the columns correspond to GAU, ROS, Ny-plain, and Ny-ALS, respectively. From top to bottom, the rows correspond to KRR, GF, and SC, respectively. The red, blue, and green lines represent the learning rates with $\gamma = 0, 0.4, 1$, respectively.

## F.2 REAL APPLICATIONS

In this section, we apply SARP to two real-world datasets from the UCI repository: the Handwritten Digits USPS dataset[1] and the cpusmall dataset[2] . The USPS dataset contains 7291 training and 2007 test images of size $16 \times 16$ pixels, while the cpusmall dataset consists of 8192 samples with 12 CPU characteristic features, where the target variable represents the percentage of time that CPUs run in user mode. For both datasets, we randomly select $80\%$ of the samples for training and the remaining $20\%$ for testing. Note that the USPS dataset corresponds to a classification task, whereas the cpusmall dataset corresponds to a regression task.

We implement SARP using a Gaussian RBF kernel with bandwidth 24, combined with KRR, GF, and SC as the spectral algorithms, and GAU, ROS, Ny-plain, and Ny-ALS as the projection methods. The regularization parameter $\lambda$ is chosen from $10^{-6}, 10^{-5}, \ldots, 10^{0}$ via the NNI toolbox[3] . The projection dimension $m$ is varied over $20, 50, 100, 200, 300, 400, 500$. For each combination of $(\lambda, m)$, we run SARP independently 20 times and report the average performance. Classification error rate is used as the evaluation metric for the USPS dataset, while mean squared error (MSE) is used for the cpusmall dataset.

Figure 5 illustrates the performance of SARP with different spectral algorithms and projection methods on the USPS and cpusmall datasets. The results show that as the projection dimension $m$ increases, SARP's performance improves across all combinations of spectral algorithms and projection methods. Notably, when $m$ reaches 600, SARP achieves stable performance, demonstrating its ability to reduce computational costs while maintaining high accuracy. Among the various projection methods, the sketching methods (GAU and ROS) and Nyström methods (Ny-plain and Ny-ALS) exhibit comparable performance, with no significant difference between the two types. However, Ny-ALS performs slightly better than Ny-plain, which aligns with our theoretical findings that Ny-ALS requires a smaller projection dimension than Ny-plain to achieve optimal learning rates.

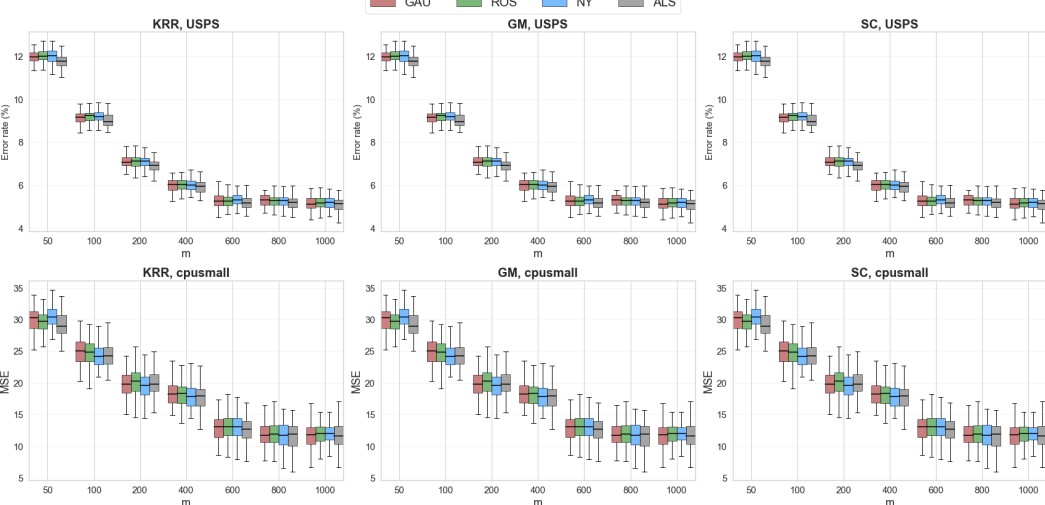

Figure 5: Performance of SARP with different spectral algorithms and projection methods on the USPS and cpusmall datasets. The top and bottom rows correspond to the USPS and cpusmall datasets, respectively. The Left and right columns correspond to KRR, GF, and SC, respectively. The red, blue, green, and gray lines represent the performance of SARP with GAU, ROS, Ny-plain, and Ny-ALS, respectively.

---

[1] https://www.kaggle.com/datasets/bistaumanga/usps-dataset/data
[2] https://www.csie.ntu.edu.tw/~cjlin/libsvmtools/datasets/regression.html
[3] https://github.com/microsoft/nni

