# OpenReview forum: "Random Projections for Spectral Algorithms in Mis-specified Setting: Sobolev Norm Learning Rates and Minimax Optimality"
_ICLR.cc/2026/Conference — ICLR 2026 Conference Withdrawn Submission_

### Official Review · Reviewer_i6nj · 2025-10-20

**Soundness:** 1
**Presentation:** 2
**Contribution:** 2
**Rating:** 2
**Confidence:** 4

**Summary:**

This paper investigates the generalization error of scalar-valued regression estimators derived from spectral algorithms within RKHS, specifically when combined with Random Projection techniques (including Nystrom methods). The authors aim to provide a unified analysis that incorporates several refinements previously studied in isolation: handling the misspecified case (by taking into account the embedding index $\alpha$), relaxing assumptions on the target function's boundedness, using general source conditions, applying general spectral algorithms, incorporating random projections, and analyzing convergence in Sobolev norms. The main results presented are convergence rates (upper and lower bounds) under these combined conditions, with a claim that these rates avoid the saturation effect typically seen in kernel methods.

**Strengths:**

* The paper tackles an important and relevant problem for the ICLR community: understanding the theoretical properties of scalable kernel methods (like those using Nyström or other random projections) under realistic assumptions (mis-specification, general smoothness).
* The paper attempts to cover a wide range of modern theoretical aspects relevant to kernel regression, potentially offering a more complete picture.

**Weaknesses:**

Despite the interesting goal, the paper suffers from significant weaknesses, including potentially erroneous central claims and numerous presentation errors. These issues undermine the reliability of the results and require major revisions.

**Major Issues**

* **Erroneous Claim on Saturation Effect**: The paper claims that the derived rates ``do not exhibit saturation effect.'' This contradicts established theoretical results for kernel methods, which shows that saturation is unavoidable for KRR (Zhang, 2023). The error appears to stem from Lemma 8, specifically the application of Eq. (13), which seems to ignore the condition $\nu \leq \tau$. A correct application likely leads back to the standard saturation condition $s \leq \tau + \gamma$ (see [1]).
* **Issues with General Source Condition (GSC)**: The paper emphasizes the GSC as a significant generalization, but its presentation is flawed ($\tau$ used before definition, parameter $s$ missing from $f_{\rho} \in \Omega_{\phi,R}$). More critically, the analysis later appears to rely on the property $P\phi(A)P=\phi(PAP)$ (P projection, A operator). It's questionable if any commonly used non-linear $\phi$ (like the Hölder case) satisfies this under the paper's assumptions. This potentially invalidates the applicability of the results.

**Other Issues**

* **Uniformly Bounded Eigenfunctions (UBE)**: Example 1 (line 1041) suggests that the UBE assumption is a common or mild condition, citing prior work that makes this claim. This is mathematically incorrect and spreading this misconception is problematic. See [2] for a counter-example of the claim.
* **Lack of Precision Regarding Measures**: Examples 2 \& 3 claim certain RKHSs are "benign" without specifying the crucial dependence on the underlying probability measure. These properties often hold only for specific measures (like the uniform measure). Similarly, Section C.3.2 discussing Sobolev embeddings needs to specify the assumed measure. Without this, the claims are ill-defined or potentially false.
* **Lower bound**: The theorem statement claims only a bound on the effective dimension is needed, but the proof appears to use lower and upper bounds on eigenvalues. The theorem statement should precisely list all necessary assumptions. Assumption 3 and the mis-specification parameter $\alpha$ are mentioned in the theorem and discussion but seem irrelevant to the lower bound analysis. As for previous lower bound (e.g. Zhang 2024) the bound is valid for any smoothness level irrespective of $\alpha$.
* **Typos** (064-083)  stiuded, projectins (twice), (097) ,our, (143) the reproducing property holds that, (263) that leverage operator spectral, (326) plain Nyström, (350) an projection, (772-1487-1546-1563-1698-1713-) various typos with norms, (808) wwe, (1111) determined by $r$, (1131) by allowing the index function by allowing the index function, (1442) Assumption 2 (should it be 4?)...
* **Redundant Proofs & Lack of Attribution**: Section C.6 and the proof of Lemma 4 appear to reproduce technical results already established in the literature (Zhang 2024) without clear attribution or justification for their inclusion. This makes the paper appear more technically dense than necessary and obscures the novel contributions. Similarly, how different are the proofs of Lemma 16 and 17 compared to Theorem 13 and 15 in Zhang 2024? Lemma 23 can be found in Fischer & Steinwart.

[1] Blanchard and Mucke. Optimal rates for regularization of statistical inverse learning problems, 2018.
[2] Minh, Niyogi, and Yao. Mercer’s theorem, feature maps, and smoothing, 2006.

**Questions:**

* **Mercer Assumptions:** The analysis assumes the domain $\mathcal{X}$ is compact and the kernel $K$ is Mercer. However, much of the modern analysis of kernel methods relies on weaker conditions, often only requiring the kernel to be square-integrable w.r.t. the measure to ensure the integral operator is Hilbert-Schmidt \citep{fischer2020sobolev}. Could you clarify why the more restrictive classical Mercer conditions are invoked or needed?

* Could the authors detail how they go from $||O_{K,n,\lambda}^{1/2} g_{\lambda}(O_{K,n})P C_{K,n,\lambda}^{1/2}||$ to $||O_{K,n,\lambda} g_{\lambda}(O_{K,n})||$ (1682-1684), I could not convinced myself of this step.

* Have the authors considered extending their analysis to the vector-valued setting (relevant for conditional mean embeddings [1]) or incorporating recently proposed relaxations of noise assumptions [2]?

[1] Li, Meunier, Mollenhauer, Gretton. Optimal rates for regularized conditional mean embedding learning. 2022
[2] Mollenhauer, Mucke, Meunier, retton. Regularized least squares learning with heavy-tailed noise is minimax optimal. 2025.

---

> ### Author Response · Authors · 2025-11-18
> **Responses to Reviewer i6nj**
>
> **Thank you for your valuable feedback and helpful comments on our work! Our point-by-point responses to your comments are given below.**
>
>
> *Weakness 1: The paper claims that the derived rates "do not exhibit saturation effect." This contradicts established theoretical results for kernel methods, which shows that saturation is unavoidable for KRR (Zhang, 2023). The error appears to stem from Lemma 8, specifically the application of Eq. (13), which seems to ignore the condition $\nu \leq \tau$. A correct application likely leads back to the standard saturation condition $s \\leq \tau+\gamma$ (see [1]).*
>
> **Answer**: We thank the reviewer for this careful observation and apologize for the confusion caused by our current presentation. We agree that there is a mistake in our current argument regarding the saturation effect for SARP.
>
> For $s>2\tau$, a **saturation effect** occurs, the learning rate no longer improves with $s$ anymore and coincides with the rate at $s=2\tau$. This phenomenon is well known in the theory of inverse problems and has been proved for KRR ($\tau=1$) in [1] and for general spectral algorithms ($\tau>0$) in [2]. In our setting, we also cannot avoid this effect, because our argument requires $s\le 2\tau+\gamma$. Specifically, in the proof of Lemma 8, we use the second inequality in Definition 3 (property of filter function in (13)), which is only valid for $\nu \in [0,\tau]$.  In Lemma 8, we plug in $\nu = (s-\gamma)/2$, so we must have
>
> $$
> \frac{s-\gamma}{2} \le \tau
> \quad\Longleftrightarrow\quad s \le 2\tau + \gamma.
> $$
>
> Therefore, Lemma 8 and Proposition 2 are only valid for $s \le 2\tau + \gamma$, and all statements that depend on them are restricted to this range as well. Our claim that SARP completely overcomes the saturation effect is incorrect. In particular, for $s>2\tau$ the classical saturation phenomenon for KRR and SA still persists, in contrast to what we stated in the current draft.
>
> Despite this mistake being ignored, we emphasize that the underlying proofs do not break down:
> the technical arguments in Lemma 8 and Proposition 2 remain correct once the restriction $s \le 2\tau + \gamma$ is made explicit. And it does not affect the remaining proofs. This correction does not change our main contributions, which concern SARP under more general conditions (mis-specified setting, general source condition, absence of uniform boundedness on $f_\rho$, and Sobolev norm analysis), together with minimax-optimal rates and sharp requirements on the projection dimension.
>
> **In the revised version, we have **removed all claims and discussion about overcoming the saturation effect for SARP**, and we have explicitly **imposed the restriction $s \in (0, 2\tau]$** in our paper. In this way,  all the statements are fully consistent with the classical saturation theory, while the main contributions of our paper remain unchanged.**
>
> *Weakness 2: The paper emphasizes the GSC as a significant generalization, but its presentation is flawed ( $\tau$ used before definition, parameter $s$ missing from $f_\rho \in \Omega_{\phi, R}$ ). More critically, the analysis later appears to rely on the property $P \phi(A) P=\phi(P A P)$ (P projection, A operator). It's questionable if any commonly used non-linear $\phi$ (like the Hölder case) satisfies this under the paper's assumptions. This potentially invalidates the applicability of the results.*
>
> **Answer**: We thank the reviewer for carefully checking our use of the general source condition (GSC) and for pointing out both the presentation issues and the concern about $P\phi(A)P=\phi(PAP)$.
>
> First, we agree that the current presentation of the GSC is flawed. In particular, $\tau$ is used before being defined, and the smoothness parameter $s$ is not clearly linked to the set $\Omega_{\varphi,R}$. In the revised version, we will have fixed the notation and restated Assumption 4. First, we define the index function set as
> $$
> \mathcal{C}\_s=\{\phi: [0,\kappa^2] \\rightarrow \\mathbb{R}^+, \phi \ \\text{ is non-decreasing },  \phi(0)=0, \phi(\kappa^2)<\infty, \\text{ and } \frac{\phi^2}{t\^{s}} \ \\text{ is non-decreasing for some } s\in (0,2\tau]\}.
> $$
> Then we correct the Assumption 4 as for $R \geq 0$, assume the smoothness condition can be expressed as
> $$
> f\_{\rho}\in \Omega\_{\phi, s, R}:=\\{f \in L^2\_{\rho_{\cal X}}: f=\phi(L_K)g, \|g\|\_{\rho}\leq R, \phi \in  \mathcal{C}\_s \\},
> $$
>
> where $\phi(\cdot)$ is the  index function. For $\lambda \in [0, \kappa^2]$, we further assume that there exists $c>0$ such that $c\lambda^{\tau}/\phi(\lambda)\leq \inf_{t \in [\lambda, \kappa^2]}t^{\tau}/\phi(t)$, **where $\tau$ is the qualification parameter will be defined in Definition 3**.

---

> ### Author Response · Authors · 2025-11-18
> **Responses to Reviewer i6nj**
>
> Theorem 2 further relies on the property $P \phi(A) P=\phi(P A P)$ (P projection, A operator). Actually, the Hölder source condition is a special case of the general source condition with \$\\phi(t)=t^{s/2}\$, and it satisfies \$P\\phi(A)P=\\phi(PAP)\$. To prove this, we only need to show that $P^{a}=P$ for any $a \in \mathbb{R}$.  The proof when $a \in Z$ is trivial since $P^2=P$. When $a \notin Z$, the identity $P^2=P$ implies that the eigenvalues of $P$ are only 0 and 1. Then there exists an invertible matrix $B$ such that $P=BDB\^{-1}$, where $D$ is a diagonal matrix with 1 or 0 in the diagonal. **Hence $P^a=BD^aB^{-1}=BDB^{-1}=P$**.
>
> **In the revised version, we have clarified this definition of the GSC, the role of \$s\$ and \$\\tau\$, and proved the above property in Assumption 4 and the related discussion in Appendix C.4.**
>
> *Weakness 3: Example 1 (line 1041) suggests that the UBE assumption is a common or mild condition, citing prior work that makes this claim. This is mathematically incorrect and spreading this misconception is problematic. See [2] for a counter-example of the claim.*
>
> **Answer**: We thank the reviewer for pointing out this issue and apologize for the confusion caused by our current presentation. In full generality, UBE is a relatively strong assumption, and there are explicit counter-examples as highlighted in [3]. Our main SARP results do not assume UBE as a standing hypothesis on the whole framework. UBE only appears when we specialize to the benign case and want to provide simple, verifiable examples in which the embedding assumption is satisfied.
>
> **In the revised version, we have removed any wording suggesting that UBE is a mild assumption and replaced Examples 1 and 2 with the exact kernel class satisfying the UBE assumption. We have also cited [3] in the discussion around Example 1 to clearly acknowledge that UBE fails in some cases.**
>
> *Weakness 4: Examples 2 & 3 claim certain RKHSs are "benign" without specifying the crucial dependence on the underlying probability measure. These properties often hold only for specific measures (like the uniform measure). Similarly, Section C.3.2 discussing Sobolev embeddings needs to specify the assumed measure. Without this, the claims are ill-defined or potentially false.*
>
> **Answer**: We thank the reviewer for pointing out this important issue. We agree that our current presentation of Examples 2 and 3, as well as Section C.3.2, is not precise enough regarding the dependence on the underlying probability measure $\rho_{\cal X}$.
>
> **In the revised version, we have clearly stated that the benign property is not a generic kernel property, but also depends crucially on $\rho_{\cal X}$. We have explicitly specified the measure assumptions, such as $\rho_{\cal X}$ being the uniform distribution on a bounded domain in Example 2 and 3. More generally in Section C.3.2, $\rho_{\cal X}$ is equivalent to the Lebesgue measure $\rho^{\prime}$ on $\cal X$, $\rho_{\cal X} \ll \rho^{\prime}$, $\rho^{\prime}  \ll \rho_{\cal X}$, and there exists constants $c, C>0$ such that $\frac{d\rho_{\cal X}}{d \rho^{\prime}} \in [c,C]$ is $\rho^{\prime}$-almost surely satisfied.**  These assumptions are exactly what ensure that the Sobolev embeddings and benign statements we use hold for the given RKHS.
>
> *Weakness 5: The theorem statement claims only a bound on the effective dimension is needed, but the proof appears to use lower and upper bounds on eigenvalues. The theorem statement should precisely list all necessary assumptions. Assumption 3 and the mis-specification parameter $\alpha$ are mentioned in the theorem and discussion but seem irrelevant to the lower bound analysis. As for previous lower bound (e.g. Zhang 2024) the bound is valid for any smoothness level irrespective of $\alpha$.*
>
> **Answer**: We thank the reviewer for this careful observation. We agree that our current proof of the lower bound does use both **upper and lower bounds** on the eigenvalues, not just an upper bound on the effective dimension. More precisely, in the proof we construct a least-favourable kernel operator $L_K$ whose eigenvalues $(\mu_k)_k$ satisfy $q_1 k^{-\beta} \leq \mu_k \leq q_2 k^{-\beta}$. This is indeed a **stronger** assumption than Assumption 2, which only requires ${\cal N}(\lambda)\lesssim \lambda^{-\frac{1}{\beta}}$. However, the two-sided bound implies Assumption 2 and in fact yields the sharper equivalence ${\cal N}(\lambda)\asymp \lambda^{-\frac{1}{\beta}}$. So the least-favourable distribution we construct is still inside the model class defined by Assumption 2.
>
> **For clarity, in the revised version, we have added an extra assumption before the lower bound theorem. We have stated that this is an enhanced version of Assumption 2 (proof added in Appendix C.5), which is used only for the construction of the lower bound**.

---

> ### Author Response · Authors · 2025-11-18
> **Responses to Reviewer i6nj**
>
> We also confirm that Assumption 3 and the parameter $\alpha$ do not play a role in the lower bound analysis. **In the revision, we have removed Assumption 3 and $\alpha$ from the statement of the lower bound theorem and explicitly stated that the result holds irrespective of $\alpha$ (consistent with [4])**.
>
> *Weakness 6: Typos (064-083) stiuded, projectins (twice), (097) ,our, (143) the reproducing property holds that, (263) that leverage operator spectral, (326) plain Nyström, (350) an projection, (772-1487-1546-1563-1698-1713-) various typos with norms, (808) wwe, (111) determined by $r,(1131)$ by allowing the index function by allowing the index function, (1442) Assumption 2 (should it be 4?)...*
>
> **Answer**: We thank the reviewer for carefully pointing out these typos and inconsistencies. **We have gone through the manuscript and corrected all of the instances listed by the reviewer. We have also carried out an additional proofreading pass to fix further minor typographical and notation errors**. These changes only affect the presentation and do not impact any of the technical results.
>
> **Weakness 7: Section C. 6 and the proof of Lemma 4 appear to reproduce technical results already established in the literature (Zhang 2024) without clear attribution or justification for their inclusion. This makes the paper appear more technically dense than necessary and obscures the novel contributions. Similarly, how different are the proofs of Lemma 16 and 17 compared to Theorem 13 and 15 in Zhang 2024? Lemma 23 can be found in Fischer \& Steinwart.**
>
> **Answer**: We thank the reviewer for raising this concern and for pointing out the overlap with existing results. Regarding Section C.6 and Lemma 4, our intention was not to present new technical contributions, but to collect auxiliary results that are used several times later, so that the proofs can be read in a self-contained way. We agree that this motivation is not clearly stated and that the attribution to [4,5] is incomplete. In the revised version, we will explicitly label Section C.6 and Lemma 4 as recalling known results for completeness and add clear citations to [4,5] where appropriate.
>
> Concerning Lemmas 16 and 17, we already state at the beginning of these lemmas that the proofs are inspired by Zhang (2024), and we will make this connection even more explicit. While the proof strategies are similar at a high level, **there are two important differences**:
>
> (i) our statements and arguments are formulated under a general source condition (via an index function), whereas Zhang (2024) works with Hölder-type source conditions;
>
> (ii) the resulting bounds and conditions on the sample size $n$ differ in several details, reflecting our more general assumptions (e.g., the dependence on the source function). We also emphasize that the focus of the two papers is different: [4,5] studies the full spectral algorithm (SA), while our work analyzes SARP.
>
> **We have revised Section C.6, Lemma 4, Lemmas 16 and 17 accordingly and added more discussion around them, so that known results are clearly identified and attributed, and the novel technical contributions stand out more clearly.**
>
> *Question 1: The analysis assumes the domain $\cal X$  is compact and the kernel $K$  is Mercer. However, much of the modern analysis of kernel methods relies on weaker conditions, often only requiring the kernel to be square-integrable w.r.t. the measure to ensure the integral operator is Hilbert-Schmidt \\citep{fischer2020sobolev}. Could you clarify why the more restrictive classical Mercer conditions are invoked or needed?*
>
> **Answer**: We thank the reviewer for raising this important point about the compactness assumption on $\cal X$ and Mercer kernel. In the current version, we assume that $\cal X$  is compact and $K$ is a Mercer kernel, mainly to stay within the standard textbook setting [6]. This guarantees the spectral decomposition of the integral operator $L_K$ and $C_K$ in (5). Our analysis, however, does not use any geometric property of compact sets. It only relies on the spectral properties of $L_K$, which are needed to define the effective dimension, the interpolation spaces, and to apply standard operator inequalities.
>
> The compactness of $\cal X$ is therefore not essential. A more general and still standard setting is: ${\cal X}$ is a Borel subset of $\mathbb{R}^d$ (not necessarily compact), $\rho_{\cal X}$ is a Borel probability measure on ${\cal X}$ and $\cal H$ is a separable RKHS on $\cal X$ with a bounded, measurable, symmetric, positive definite kernel $K$. Thus $K$ is also square-integrable w.r.t. the measure $\rho_{\cal X}$. Under these conditions, the integral operator $L_K$ is Hilbert-Schmidt, hence compact and self-adjoint, and it admits a spectral decomposition.  All our capacity and source conditions, as well as the operator inequalities we use, remain valid, so the learning rates and the required projection dimension do not change.

---

> ### Author Response · Authors · 2025-11-18
> **Responses to Reviewer i6nj**
>
> **In the revised version, we have replaced the compactness and Mercer assumption with the more general assumption and clarified that our results do not rely on classical compactness and Mercer conditions.**
>
> *Question 2: Could the authors detail how they go from $\left\\|O\_{K, n, \lambda}^{1 / 2} g\_\lambda\left(O\_{K, n}\right) P C\_{K, n, \lambda}^{1 / 2}\right\\|$ to $\left\\|O\_{K, n, \lambda} g\_\lambda\left(O\_{K, n}\right)\right\\|(1682-1684)$, I could not convinced myself of this step.*
>
> **Answer**: We apologize for the confusion caused by this step and thank the reviewer for pointing it out. Below, we provide the detailed argument.
>
> Since $C_{K,n}$ and $O_{K,n}=PC_{K,n}P$ are self-adjoint, then any continuous function $f$ and $g$, we have $f(O_{K,n})g(O_{K,n})=g(O_{K,n})f(O_{K,n})$. So we have
>
> $$
> \\|O_{K,n,\lambda}^{1/2}g_{\lambda}(O_{K,n})PC_{K,n,\lambda}^{1/2}\\|=\\|g_{\lambda}(O_{K,n})O_{K,n,\lambda}^{1/2}PC_{K,n,\lambda}^{1/2}\\|.
> $$
>
> Note that for any $a \in {\cal H}$, $\\langle P^2a, a \\rangle\_{{\cal H}}=\\|Pa\\|\_{{\cal H}}^2\leq \|a\|\_{{\cal H}}^2=\\langle a, a\\rangle\_{{\cal H}}$ for $\\|P\\|\leq 1$. So we have
>
> $$
> P(C_{K,n}+\lambda I) P \preceq PC_{K,n}P+\lambda I =O_{K,n,\lambda}.
> $$
>
> Let $B=g_{\lambda}(O_{K,n})O_{K,n,\lambda}^{1/2}$, we have
>
> $$
> \\|BPC\_{K,n,\lambda}^{1/2}\\|\^2=\\|BPC\_{K,n,\lambda}PB\^*\\| \\leq \\|BO_{K,n,\lambda}^{1/2}\\|^2
> $$
>
>
> where $B^*=O_{K,n,\lambda}^{1/2}g_{\lambda}(O_{K,n})$.
>
> Combining the above, we have
>
> $$
> \left\\|O_{K, n, \lambda}^{1 / 2} g_\lambda\left(O_{K, n}\right) P C_{K, n, \lambda}^{1 / 2}\right\\|=\left\\| g_\lambda\left(O_{K, n}\right)O_{K, n, \lambda}\\right\|=\left\\|O_{K, n, \lambda} g_\lambda\left(O_{K, n}\right)\right\\|.
> $$
>
> **In the revised version, we have inserted this detailed derivation in the proof to make this step fully transparent.**
>
> *Question 3: Have the authors considered extending their analysis to the vector-valued setting (relevant for conditional mean embeddings [1]) or incorporating recently proposed relaxations of noise assumptions [2]?*
>
> **Answer**:  We thank the reviewer for these insightful suggestions. In the present paper ,we have not pursued extensions to the vector-valued setting or to more relaxed noise assumptions, but we agree that both directions are natural and important.
>
> Regarding the vector-valued setting (e.g. conditional mean embeddings as in [7]), many of our structural assumptions and techniques formally extend if one replaces the real-valued interpolation space with the vector-valued interpolation space. The capacity and source conditions can be formulated in terms of the spectrum of the corresponding integral operator, and the random projection step can, in principle, be adapted to this setting. A full treatment, however, would require checking all steps carefully in the vector-valued framework, which is beyond the scope of the current paper.
>
> As for relaxations of the noise assumption (e.g. the heavy-tailed or robust frameworks proposed in [8]), our proofs currently rely on a Bernstein-type condition to obtain exponential concentration for the sample variance term. In principle, one could combine our SARP analysis with robust concentration results tailored to heavy-tailed noise (i.e., the Fuk–Nagaev inequality), while keeping the rest of the argument unchanged. We therefore expect that our results can be extended to certain heavy-tailed noise settings, but working out the details would require a separate, careful study.
>
> **In the revised version, we have added a short paragraph in the conclusion and Appendix C.6 to explicitly mention these two directions as promising avenues for future work.**
>
> *[1] Yicheng Li, Haobo Zhang, and Qian Lin. On the saturation effect of kernel ridge regression.*
>
> *[2] Weihao Lu, Yicheng Li, Qian Lin, et al. On the saturation effects of spectral algorithms in large dimensions.*
>
> *[3] Minh, Niyogi, and Yao. Mercer’s theorem, feature maps, and smoothing.*
>
> *[4] Haobo Zhang, Yicheng Li, and Qian Lin. On the optimality of misspecified spectral algorithms.*
>
> *[5] Simon Fischer and Ingo Steinwart. Sobolev norm learning rates for regularized least-squares
> algorithms*
>
> *[6] Martin J Wainwright. High-dimensional statistics: A non-asymptotic viewpoint.*
>
> *[7] Li, Meunier, Mollenhauer, Gretton. Optimal rates for regularized conditional mean embedding learning.*
>
> *[8] Mollenhauer, Mucke, Meunier, Retton. Regularized least squares learning with heavy-tailed noise is minimax optimal.*

---

> ### Comment · Reviewer_i6nj · 2025-11-19
>
> - **Answer to Weakness 1:** The authors should verify their proofs regarding the saturation level. Saturation typically occurs at  2 tau + gamma rather than 2 tau when measuring error in the gamma-norm. Previous literature claiming saturation at 2 tau often overlooked this distinction. For reference, please consult Optimal Rates For Regularization Of Statistical Inverse Learning Problems (Blanchard & Mücke), which clarifies the interplay between the source condition and the norm used for error analysis.
>
> - **Answer to Weakness 2:** I thank the authors for the attempt to clarify, but the provided proof contains a fundamental algebraic error. It is mathematically incorrect to claim that showing P^a = P is sufficient to obtain P phi(A)P = phi(PAP) when phi(t) =  t^s/2. This logic implicitly assumes that powers distribute over matrix products, i.e., (PAP)^a = P^aA^aP^a, which is false for non-commuting operators. As a counter-example, consider phi(t) =  t^2, then P phi(A)P = PA^2P is not equal to phi(PAP)  = PAPAP. Since the paper relies on this identity holding for general operators where P does not commute with A, the central assumption of the paper is never satisfied. Due to this critical mathematical flaw, I am reducing my score.
>
> - **Answer to Weakness 5:**  I thank the authors for the revision, but a new error appears to have been introduced in the statement of Theorem 1. To properly characterize the saturation effect, the lower bound must be established for any value of s>0, not restricted to s in (0, 2 tau] (or s in (0, 2 tau + gamma]). The definition of saturation is precisely that the convergence rate stops improving even as the regularity s increases beyond the threshold. If the lower bound is only stated for s<= 2 tau + gamma, it does not prove saturation; it only proves optimality within the unsaturated regime. See, for example, Zhang et al. 2024, Theorem 2, where the lower bound covers the full range of s.

---

### Official Review · Reviewer_3MZ4 · 2025-10-28

**Soundness:** 2
**Presentation:** 3
**Contribution:** 3
**Rating:** 6
**Confidence:** 4

**Summary:**

This paper provides a comprehensive and improved analysis of the generalization performance of RP-based spectral algorithms under general
conditions, without increasing computational complexity. The focus is on generalization performance in mis-specified settings (where the target function may not lie in the RKHS) under general conditions, without assuming uniform boundedness on the regression function. Matching upper and minimax lower bounds in Sobolev norms are established to show the optimality of the algorithm. Numerical experiments confirm theoretical rates and practical benefits.

**Strengths:**

* This paper is well-written and the structure is clear, despite the technical complexities. A table summarizing the related results is provided, allowing for a easy comparision.

* This paper presents a comprehensive analysis of the spectral algorithms with random projection, unifying Radomized Sketches, plain Nystrom and ALS Nystrom. The settings are general regarding the source condition, embedding properties and algorithms. Moreover, the assumptions are discussed in detail.

* The theory in this paper is strong, proving the minimax optimal rates of SARP. Moreover, the required projection dimension is less than or equal to that required in the literature, providing theoretical guarantees for reducing computational costs.

* Empirical experiments are provided to validate the theory.

**Weaknesses:**

* Assumption 4 cannot hold for $\phi(u) = u^{s/2}$ when $s >2\tau$. Consequently, the saturation effect still holds for KRR, in contrast to the claim in the paper. See the proof of Lemma 8 and Proposition 2.

* The technical contribution in this paper seems to be marginal. The proof idea and steps seem to be standard. A overview of the technical novelty can be provided and emphasized.

* It would improve the contribution of this paper to propose practical criteria for choosing the projection dimension.

**Questions:**

1. While the proof in this paper is erroneous, is it possible that indeed SARP does not suffer from the saturation effect? Do we have supporting empirical evidence?


1. Is the current requirement of the projection dimension $m$ minimal for the optimal rates? Is it possible to establish a lower bound for it, or what are the difficulties here?

2. Can you weaken requirement that $\mathcal{X}$ is compact? What will be the impact of non-compactness to the effectiveness of random projection methods?

---

> ### Author Response · Authors · 2025-11-18
> **Responses to Reviewer 3MZ4**
>
> **Thank you for your valuable feedback and helpful comments on our work! Our point-by-point responses to your comments are given below.**
>
> *Weakness 1: Assumption 4 cannot hold for $\phi(u)=u^{s/2}$ when $s>2\tau$. Consequently, the saturation effect still holds for KRR, in contrast to the claim in the paper. See the proof of Lemma 8 and Proposition 2.*
>
> **Answer**:  We thank the reviewer for this careful observation and apologize for the confusion caused by our current presentation. We agree that there is a mistake in our current argument regarding the saturation effect for SARP.
>
> For $s>2\tau$, a **saturation effect** occurs, the learning rate no longer improves with $s$ anymore and coincides with the rate at $s=2\tau$. This phenomenon is well known in the theory of inverse problems and has been proved for KRR ($\tau=1$) in [1] and for general spectral algorithms ($\tau>0$) in [2]. In our setting, we also cannot avoid this effect, because our argument requires $s\le 2\tau+\gamma$. Specifically, in the proof of Lemma 8, we use the second inequality in Definition 3 (property of filter function in (13)), which is only valid for $\nu \in [0,\tau]$.  In Lemma 8, we plug in $\nu = (s-\gamma)/2$, so we must have
>
> $$
> \frac{s-\gamma}{2} \le \tau
> \quad\Longleftrightarrow\quad s \le 2\tau + \gamma.
> $$
>
> Therefore, Lemma 8 and Proposition 2 are only valid for $s \le 2\tau + \gamma$, and all statements that depend on them are restricted to this range as well. Our claim that SARP completely overcomes the saturation effect is incorrect. In particular, for $s>2\tau$ the classical saturation phenomenon for KRR and SA still persists, in contrast to what we stated in the current draft.
>
> Despite this mistake being ignored, we emphasize that the underlying proofs do not break down:
> the technical arguments in Lemma 8 and Proposition 2 remain correct once the restriction $s \le 2\tau + \gamma$ is made explicit. And it does not affect the remaining proofs. And this correction does not change our main contributions, which concern SARP under more general conditions (mis-specified setting, general source condition, absence of uniform boundedness on $f_\rho$, and Sobolev norm analysis), together with minimax-optimal rates and sharp requirements on the projection dimension.
>
> **In the revised version, we have removed all claims and discussion about overcoming the saturation effect for SARP, and we have explicitly imposed the restriction $s \in (0, 2\tau]$ in our paper. In this way,  all the statements are fully consistent with the classical saturation theory, while the main contributions of our paper remain unchanged.**
>
> *Weakness 2: The technical contribution in this paper seems to be marginal. The proof idea and steps seem to be standard. A overview of the technical novelty can be provided and emphasized.*
>
> **Answer**: We thank the reviewer for this comment and for suggesting better highlight the technical novelties of our work. Although our analysis shares some high-level ideas and tools with the existing spectral-regularization literature, it also contains several non-trivial technical contributions of its own. In particular:
>
> - **Sobolev-norm minimax lower bounds under general source condition and mis-specified case**: We derive minimax lower bounds in Sobolev norms under a general source condition, and in the mis-specified setting. This goes beyond previous lower-bound results, which typically work in $L^2_\rho$ or ${\cal H}$ norm in the well-specified case or pure H\"older source conditions. A key technical novelty is the construction of an appropriate family of functions $(f_i)_{i\in\mathcal{I}}$ that satisfy the general source condition with index function $\phi$. Unlike the H\"older case $\varphi(u)=u^{s/2}$, where one can directly choose $f_i$ along eigenfunctions with prescribed polynomial decay, the general source framework requires us to carefully design $f_i = \phi(L_K) g_i$ with suitable $g_i$ so that: (1) the Sobolev norms can be controlled; (2) the functions remain well separated in the risk metric; (3) the induced prior fits within the assumptions.
> - **Different error decomposition**: Compared with the existing SA literature, our upper-bound proof has to handle an additional error term induced by random projections, which is controlled by the operator similarity parameter $\mathcal{A}\_{K,P}$. This term also enters the ​empirical error term​, so the analysis cannot be reduced to a direct adaptation of classical SA arguments. Compared with previous SARP works, our error decomposition is also different, especially in the treatment of the empirical error term. We explicitly distinguish the regimes $s<2$ and $s\ge 2$, and our analysis of the term $T_{n,\lambda}$ is carried out separately for three different types of source conditions, with tailored bounds in each case. These three cases and their corresponding bounds are new.

---

> ### Author Response · Authors · 2025-11-18
> **Responses to Reviewer 3MZ4**
>
> At the same time, our overall decomposition is simpler and more transparent than in [3], while still yielding optimal rates under more general assumptions.
> - **Sharper error control of the operator similarity:** By using the embedding condition, we refine the analysis of operator similarity (Lemmas 11–17) and obtain sharper error bounds in the final results. In particular, we derive tighter estimates of (i) the similarity between the covariance operator $C_K$ and the sample covariance operator $C_{K,n}$, leading to the condition on $n$ from $n \gtrsim \lambda^{-1}$ to $n \gtrsim \lambda^{-\alpha}$, and (ii) the sample-variance term, improving it roughly from ${\cal O}(n^{-1}\lambda^{\min\{-1/2,\, s/2-1\}}+n^{1/2}\lambda^{(s-1)/2})$ to ${\cal O}(n^{-1}\lambda^{\min\{-\alpha/2,\, s/2-\alpha\}}+n^{1/2}\lambda^{(s-\alpha)/2})$. Different from the analysis in [1], we further do not require the uniform boundedness of $f_{\rho}$.
>
> **In the revised version, we have highlighted and added more details on the technical novelty in the Remark 1,4,5, comparison in Appendix B, and the beginning of proofs of Lemma 16-18.**
>
> *Weakness 3: It would improve the contribution of this paper to propose practical criteria for choosing the projection dimension.*
>
> **Answer**: We appreciate this suggestion and agree that providing more practical criteria for choosing the projection dimension $m$ would strengthen the contribution of the paper.
>
> In our simulation, when the parameters $\beta$ and $s$ are known, we can set the projection dimension $m$ according to the theoretical optimal form of different RP methods in Corollary 2, i.e., $m=Cn^{\frac{1}{s\beta+1}}$ for randomized sketches when $s\in [\alpha-1/\beta,2]$. The constant $C$ can be either fixed or chosen by grid search or cross-validation. For the real applications when the parameters $\beta$ and $s$ are unknown, we can also adopt the grid search for $m$ on an independent validation dataset or a cross-validation method. Specifically, we can perform a hyperparameter search over a grid of different values for $m$, such as ${10}^{0.5(a+1)}$ with $a=\{0,0.5,1,\ldots,\}$ (smaller than the total sample size). The result corresponding to the smallest error measure on an independent validation dataset is selected and recorded. This type of tuning procedure is standard in the kernel methods and random projection literature [4,5].
>
> In the revised version, we have added a short discussion on describing these practical choices in the experiments section and have briefly summarized the grid-search strategy for $m$ in Remark 7.
>
> *Question 1: While the proof in this paper is erroneous, is it possible that indeed SARP does not suffer from the saturation effect? Do we have supporting empirical evidence?*
>
> **Answer**: We apologize that our discussion of the saturation effect caused confusion. As the reviewer correctly pointed out, once we use the second inequality in Definition 3, the proof of Lemma 8 implicitly requires $(s-\gamma)/2 \le \tau$, namely $s \le 2\tau+\gamma$.  With this restriction, Lemma 8 and Proposition 2 remain mathematically correct, but they no longer imply that  SARP overcomes the classical saturation effect.
>
> **In the revised version, we have removed all claims and discussion about overcoming the saturation effect for SARP, and we have explicitly imposed the restriction $s \in (0, 2\tau]$ in our paper. In this way,  all the statements are fully consistent with the classical saturation theory, while the main contributions of our paper remain unchanged.**
>
> *Question 2: Is the current requirement of the projection dimension  minimal for the optimal rates? Is it possible to establish a lower bound for it, or what are the difficulties here?*
>
> **Answer**: We thank the reviewer for this insightful question about the sharpness of the projection dimension requirement. In view of the existing literature [3,4,5], our condition on the projection dimension can be regarded as a minimal order for attaining the optimal rates. More precisely, our bounds show that it is sufficient to take $m \gtrsim \mathcal{N}(\lambda)$ for randomized sketches and ALS  Nystrom or $\mathcal{N}\_\infty(\lambda)$  for plain Nystrom up to logarithmic factors. In this sense, our condition $m \gtrsim \mathcal{N}(\lambda)$ (or $\mathcal{N}\_\infty(\lambda)$) matches the best-known requirements in the literature. However, a fully general information-theoretic lower bound on \$m\$ for SARP is still an open problem and lies beyond the scope of this paper.
>
> **In the revised version, we have discussed this point after Corollary 2 to make clear how our requirement compares to existing sharp results on RP-based kernel methods. And in the conclusion, we have added a future direction to investigate the lower bound on \$m\$ for SARP.**

---

> ### Author Response · Authors · 2025-11-18
> **Responses to Reviewer 3MZ4**
>
> *Question 3: Can you weaken requirement that $\cal X$  is compact? What will be the impact of non-compactness to the effectiveness of random projection methods?*
>
> **Answer**:  We thank the reviewer for raising this important point about the compactness assumption on $\cal X$. In the current version, we assume that $\cal X$  is compact and $K$ is a Mercer kernel, mainly to stay within the standard textbook setting [6]. This guarantees the spectral decomposition of the integral operators $L_K$ and $C_K$ in (5). Our analysis, however, does not use any geometric property of compact sets. It only relies on the spectral properties of $L_K$, which are needed to define the effective dimension, the interpolation spaces, and to apply standard operator inequalities.
>
> The compactness of $\cal X$ is therefore not essential. A more general and still standard setting is: ${\cal X}$ is a Borel subset of $\mathbb{R}^d$ (not necessarily compact), $\rho\_{\cal X}$ is a Borel probability measure on ${\cal X}$ and $\cal H$ is a separable RKHS on $\cal X$ with a bounded, measurable, symmetric, positive definite kernel $K$. Under these conditions, the integral operator $L\_K$ is Hilbert-Schmidt, hence compact and self-adjoint, and it admits a spectral decomposition.  All our capacity and source conditions, as well as the operator inequalities we use, remain valid, so the learning rates and the required projection dimension do not change.
>
> In more challenging settings where $\cal X$ is non-compact and $K$ is unbounded, one may need additional tail or moment assumptions on $\rho_{\cal X}$ to control the effective dimension. However,  this may increase the required projection dimension $m$. And this phenomenon is inherent to the underlying kernel method and is not specific to random projections.
>
> **In the revised version, we have replaced the compactness assumption with the more general assumption and clarified that our results do not rely on classical compactness and Mercer conditions.**
>
> *[1] Yicheng Li, Haobo Zhang, and Qian Lin. On the saturation effect of kernel ridge regression.*
>
> *[2] Weihao Lu, Yicheng Li, Qian Lin, et al. On the saturation effects of spectral algorithms in large dimensions.*
>
> *[3] Junhong Lin and Volkan Cevher. Convergences of regularized algorithms and stochastic gradient
> methods with random projections*.
>
> *[4] Alessandro Rudi, Raffaello Camoriano, and Lorenzo Rosasco. Less is more: Nyström computational
> regularization*.
>
> *[5] Yun Yang, Mert Pilanci, and Martin J. Wainwright. Randomized sketches for kernels: Fast and
> optimal nonparametric regression*.
>
> *[6] Martin J Wainwright. High-dimensional statistics: A non-asymptotic viewpoint.*

---

### Official Review · Reviewer_TgQw · 2025-11-01

**Soundness:** 3
**Presentation:** 2
**Contribution:** 2
**Rating:** 4
**Confidence:** 3

**Summary:**

This paper develops a comprehensive study about the generalization performance of RP-based spectral algorithms under more general conditions.

The main modification of assumptions in this paper includes two parts. Firstly, it proposes to impose classical Bernstein condition on the noise $\epsilon$ for the purpose of replacing uniform boundedness of $f_p,$ which may be difficult to satisfy in practice. Then, the paper expands traditional H\"older source condition to assumption 4 that covers more general cases.

Under the new and general assumptions, this paper first derives a minimax lower bound for the learning rate in Sobolev norms, which is the first to generalize current conclusions to both well-specified and mis-specified cases. Afterwards, it presents the sharp learning rates for general spectral algorithms with random projections (SARP), which recovers previous result for spectral algorithms without RP. Furthermore, it finds the optimal learning rates that up to a logarithmic factor of the minimax lower bounds. Finally, paper applies above theorem to three algorithms, including randomized sketches, plain and ALS Nystr\"om sub-sampling.

**Strengths:**

1. This paper is a solid work with in-depth theoretical proofs.

2. This paper is the first to establish bounds under general conditions, which have not been fully solved by previous works.

3. This work includes a large number of appendices to explain their findings, including Table 2, which provides a detailed comparison of various conditions.

**Weaknesses:**

1. Although this paper is built on more general conditions, all bounds in this paper recover previous results without getting a tighter bound. This may weaken the contribution of this work.

2. The lack of thorough discussion about specific cases of generalization of conditions (e.g. mis-specified setting and uniformly boundedness). Since one of the most important contribution of this paper is the generalization of conditions, a more thorough discussion about it can greatly demonstrate the contribution of this work. It's not very clear to me what new situations are covered in practice because of mis-specified setting and no uniformly boundedness on $f_p$.

3. Lack of comparison about conditions with previous works. There are some other works also built on more general conditions according to Table 2. Moreover, this work introduces embedding as a condition, which is not required in most other works. Thus, the general condition of this work is in doubt and needs more discussions to prove.

**Questions:**

1. Why projection dimension needed for optimality is proportional to the empirical effective dimension? The conclusion is not so straightforward to me based on current presentation.

2. How to select appropriate projection dimension and regularization parameter in practice?

---

> ### Author Response · Authors · 2025-11-18
> **Responses to Reviewer TgQw**
>
> **Thank you for your valuable feedback and helpful comments on our work! Our point-by-point responses to your comments are given below**.
>
> *Weakness 1: Although this paper is built on more general conditions, all bounds in this paper recover previous results without getting a tighter bound. This may weaken the contribution of this work*.
>
> **Answer**: We thank the reviewer for raising this concern about our contributions. We agree that our bounds do not improve the minimax exponents compared to the best existing results in overlapping regimes. This is natural, since in those regimes the known rates are already minimax optimal, and a strictly tighter rate would contradict existing lower bounds. Our aim is therefore not to beat the minimax rates, but to
>
> (i) extend optimal rates to previously ​**uncovered mis-specified setting**​;
>
> (ii) obtain them under ​**weaker and more realistic assumptions**​;
>
> (iii) show that random projections can reach these optimal rates with a ​no larger projection dimension (and can even be smaller in some cases) than those required in [1,2,3], leading to **better statistical–computational trade-offs**.
>
> More concretely:
>
> - **New regimes beyond previous analyses**: Existing state-of-the-art SARP results [1] obtain optimal rates only for $s \\ge 1 - 1/\\beta$. In contrast, we extend the analysis to $s \ge \alpha - 1/\beta$. In the benign case $\alpha = 1/\beta$, we obtain minimax-optimal rates for ​all $s \in (0,2\tau]$​, which covers the entire mis-specified regimes that were not treated before.
> - **Weaker assumptions and stronger norms**: Our analysis works under ​weaker and more realistic assumptions​: we impose a noise-level Bernstein condition, do not assume a uniform bound on $f_\rho$, and use a general source condition where the classical Hölder source condition is a special case. Under these more general assumptions, we establish the minimax lower and upper bounds in Sobolev norms, which cover the usual $L^2\_\rho$ norm and RKHS norm.
> - **Efficient computational improvement**: On the computational side, our results show that the projection dimension \$m\$ required to achieve the optimal learning rates is of the same order or smaller than in previous work on SARP and related methods ([1,2,3]). This means that we can preserve minimax optimality over a broader source range while reducing the computational cost in certain cases. The details of computational improvement can be found in Remark 3 and Appendix B.
>
> We hope this clarifies that the contribution is not just to restate known rates, but to significantly broaden the scope and practicality of minimax-optimal results for SARP.  **We have revised the introduction, Remark 3, and the comparison section to clearly highlight these three aspects of our contribution**.
>
> *Weakness 2: The lack of thorough discussion about specific cases of generalization of conditions (e.g. mis-specified setting and uniformly boundedness). Since one of the most important contribution of this paper is the generalization of conditions, a more thorough discussion about it can greatly demonstrate the contribution of this work. It's not very clear to me what new situations are covered in practice because of mis-specified setting and no uniformly boundedness on $f_{\rho}$.*
>
> **Answer**: We thank the reviewer for pointing out that our discussion of what is new behind the generalized conditions is too brief. We agree on this and have made these practical scenarios more explicit and easier to understand.
>
> - **Mis-specified setting**: Assumption 4 characterizes the regularity of the target function $f_{\rho}$ via the index function $\phi$. And the parameter $s$ quantifies the smoothness of $f\_{\rho}$ relative to the RKHS ${\cal H}\_K$. Specifically, if $s>1$, then $f\_{\rho}$ is smoother than the functions in ${\cal H}\_K$ (well-specified case), while if $s\in (0,1]$, then $f\_{\rho}$ is less smooth than the functions in ${\cal H}\_K$ (mis-specified case). The derived optimal rates allow source indices $s >\alpha - 1/\\beta$, covering part of the mis-specified case. Practically, for example, we use a very smooth kernel (Gaussian or high-smoothness Matérn) while the true regression function has only finite Sobolev regularity. Such *kernel too smooth, truth function less smooth* situations are common in practice when a Gaussian kernel is used as a default and still performs well. Our theory shows that this is reasonable since SARP can still achieve the optimal rates under this mis-specified setting.

---

> ### Author Response · Authors · 2025-11-18
> **Responses to Reviewer TgQw**
>
> - **Uniformly boundedness**: Previous work on SARP usually need the uniformly boundedness of $f\_{\rho}$ [1], restrict their analysis in a smaller function space (i.e., $[\cal H]\_\rho^s \cap L\_\rho^\infty({\cal X})$, not the entire $[\cal H]\_\rho^s$, when taking Holder condition as example). We extend the optimal lower and upper bound of SARP for $f\_{\rho} \in [\cal H]\_\rho^s$. From a practical point of view, a uniform bound on $f\_\rho$ is very hard to verify from finite data. In our synthetic example in Appendix F, the regression function diverges when $s=0.5$, so $f_\rho$ is not uniformly bounded, yet SARP still achieves the optimal Sobolev learning rates.
>
> **In the revised version, we have added a short paragraph after Assumptions 1 and 4, and expanded the discussion in Appendix B to clearly highlight these two types of new scenarios covered by our generalized conditions**.
>
> *Weakness 3: Lack of comparison about conditions with previous works. There are some other works also built on more general conditions according to Table 2. Moreover, this work introduces embedding as a condition, which is not required in most other works. Thus, the general condition of this work is in doubt and needs more discussions to prove.*
>
> **Answer**: We thank the reviewer for raising this point about the comparison of assumptions with previous work. In fact, we have provided a detailed, side-by-side comparison of assumptions in Appendix B and Table 2.
>
> While some works listed in Table 2 also use general source or capacity conditions, they typically focus on a **specific kernel method or spectral algorithm**.  In contrast, our paper is explicitly about **SARP (spectral algorithms with random projections)**. To the best of our knowledge, we are the first to systematically relax the usual assumptions (well-specified setting, Holder source condition, uniformly bounded $f_\rho$) for SARP, while still proving minimax-optimal Sobolev learning rates.
>
> Regarding the embedding assumption, we agree that it can look stronger at first sight. However, it is introduced exactly to provide the regularity needed to analyze the ​**mis-specified setting in Sobolev norms**​. In particular, we use it to derive a sharp bound on the maximal effective dimension ${\cal N}_{\infty}(\lambda)$ (Lemma 11). This bound is then used to obtain tighter operator similarity estimates (Lemmas 12–17), which allow us to handle a wider range of source conditions in the SARP analysis.  **We have clarified this role in Remark 5**. Embedding assumptions of this type have been used before for particular kernel methods or specific spectral algorithms, but, to the best of our knowledge, our work is the first to incorporate such an embedding condition into a general SARP framework in the mis-specified setting.
>
> **In the revised version, we have made these points explicit by (i) strengthening the discussion in Appendix B and around Table 2, and (ii) refining Remark 5 to clearly explain the role of the embedding assumption.**
>
> *Question 1: Why projection dimension needed for optimality is proportional to the empirical effective dimension? The conclusion is not so straightforward to me based on current presentation*.
>
> **Answer**: We apologize for the confusion caused by our current presentation of the relationship between the projection dimension and the effective dimension. In fact, our results identify the **minimal** projection dimension that still achieves the minimax rate, and this minimal dimension naturally scales with a suitable effective dimension.
>
> In the SARP framework, the extra error term caused by random projections is the projection error term and the empirical error term, which is controlled by the operator similarity measure ${\cal A}\_{K,P}$. For randomized sketches and ALS Nyström methods, Lemmas 20 and 22 show that a sharp bound on ${\cal A}\_{K,P}$ holds provided
>
> $$
> m \gtrsim \lambda^{-\frac{1}{\beta}}.
> $$
>
> For the plain Nyström method, Lemma 21 gives the corresponding condition
>
> $$
> m \gtrsim \lambda^{-\frac{1}{\alpha}};
> $$
>
> This is from Assumption 2, the average effective dimension satisfies ${\cal N}(\lambda) \lesssim \lambda^{-1/\\beta}$, and by Assumption 3 and Lemma 11, the maximal effective dimension satisfies ${\cal N}\_{\infty}(\lambda) \lesssim \lambda^{-1/\\alpha}$. The conditions above actually can be written as **$m \gtrsim {\cal N}(\lambda)$ for randomized sketches and ALS Nyström; $m \gtrsim {\cal N}_{\infty}(\lambda)$ for plain Nyström**. This is why the minimal projection dimension is **proportional to the (average or maximal) effective dimension**. Moreover, in the proof of Corollary 2 in  Appendix E.7, we explicitly obtain the optimal rates under the above conditions on $m$.
>
> **In the revised version, we have made this logic explicit in the discussion after Corollary 2 and Remark 6**.

---

> ### Author Response · Authors · 2025-11-18
> **Responses to Reviewer TgQw**
>
> *Question 2: How to select appropriate projection dimension and regularization parameter in practice?*
>
> **Answer**: We thank the reviewer for raising this practical question about how to choose the projection dimension and the regularization parameter. In our simulation, the regularization parameter is chosen according to the theoretical optimal form $\lambda = C n^{-\frac{\beta}{s\beta+1}}$. For each generated dataset, we perform a grid search of 10 values for $C$  from ${10}^{-7}$ to ${10}^{2}$. We select the value of $C$ that gives the smallest error on an independent validation set and then report the corresponding performance. The projection dimension $m$ is fixed according to the theoretical orders for different random projection methods given in Corollary 2, and the experimental results in Figures 2–4 match these theoretical predictions. We also study the effect of the projection dimension $m$  on real data in Appendix F.2. In real applications, where the parameters $\beta$ and $s$ are unknown, one can choose both $\lambda$ and $m$ by grid search on an independent validation set or by cross-validation. This type of tuning procedure is standard in the kernel methods and random projection literature [2,3].
>
> **In the revised version, we have added a short discussion on describing these practical choices in the experiments section and have briefly summarized the grid-search strategy for $\lambda$ and $m$ in Remark 7**.
>
> *[1] Junhong Lin and Volkan Cevher. Convergences of regularized algorithms and stochastic gradient
> methods with random projections*.
>
> *[2] Alessandro Rudi, Raffaello Camoriano, and Lorenzo Rosasco. Less is more: Nyström computational
> regularization*.
>
> *[3] Yun Yang, Mert Pilanci, and Martin J. Wainwright. Randomized sketches for kernels: Fast and
> optimal nonparametric regression*.

---

### Official Review · Reviewer_DTFG · 2025-11-01

**Soundness:** 3
**Presentation:** 2
**Contribution:** 3
**Rating:** 8
**Confidence:** 4

**Summary:**

This paper presents a comprehensive theoretical analysis of Spectral Algorithms with Random Projections (SARP), establishing their minimax optimality under very general conditions, particularly in the common scenario where the true function may not belong to the model's hypothesis space (the "mis-specified setting").

**Strengths:**

By applying recent developments in spectral algorithms, the authors analyzed random projection methods and derived convergent results.

This is an interesting research direction that merits further investigation in future work.

**Weaknesses:**

I did not check the entire proof. The paper would benefit from a more careful revision of the writing.

1. Can you provide more examples such that assumption 2 holds?

2. Could you be more careful on the assumption 3? It is ambiguous for non-expert.

This is an interesting topic, however, it needs to be more careful on presentation.

**Questions:**

Same to the weakness.

---

> ### Author Response · Authors · 2025-11-18
> **Responses to Reviewer TgQw**
>
> **Thank you for your valuable feedback and helpful comments on our work! Our point-by-point responses to your comments are given below**.
>
> *Weakness 1: Can you provide more examples such that assumption 2 holds?*
>
> **Answer**: We thank the reviewer for asking for more concrete examples where Assumption 2 holds. Assumption 2 is the usual capacity condition, which is widely used in the literature on kernel methods [1,2]. Let \$\\mu\_k\$ be the eigenvalues of \$L\_K\$, then
>
> $$
> {\cal N}(\lambda)=  \sum_{k=1}^{\infty}\frac{\mu_k}{\mu_k+\lambda}\leq \frac{1}{\lambda}\sum_{k=1}^{\infty}\mu_k=\frac{\text{Tr}(L_K)}{\lambda},
> $$
>
> so Assumption 2 always holds with \$\\beta=1\$ and \$Q^2=\\mathrm{Tr}(L\_K)\$.
>
> Below we list several standard examples of RKHSs for which Assumption 2 holds, together with the corresponding values of \$\\beta\$.
>
> - **Finite rank RKHSs:**  Suppose the kernel has finite rank \$C\_R\$, i.e., the eigenvalues are non-zero only up to some index \$C\_R<\\infty\$. Examples include the linear kernel, polynomial kernels, approximate kernels with a fixed number of random features, and neural tangent kernels with finite network width. In this case
> $$
> {\cal N}(\lambda)=  \sum_{k=1}^{\infty}\frac{\mu_k}{\mu_k+\lambda}\leq C_R,
> $$
> so Assumption 2 holds, for instance, with \$\\beta=1\$ and \$Q^2=C\_R\$.
>
> - **Finite smoothness RKHSs**: If the eigenvalues decay polynomially, \$\\mu\_k \\le c,k^{-\\beta}\$ for some \$\\beta>0\$, then
> $$
> {\cal N}(\lambda)=  \sum_{k=1}^{\infty}\frac{\mu_k}{\mu_k+\lambda}\leq \sum_{k=1}^{\infty}\frac{q_2}{q_2+k^\beta\lambda}\leq \int_{0}^{\infty}\frac{q_2}{q_2+t^\beta\lambda}dt \leq \lambda^{-1/\beta}\int_0^\infty \frac{q_2}{q_2+t^\beta}d t\leq \left (1+\frac{q_2}{b-1} \right )\lambda^{-\frac{1}{\beta}},
> $$
>
> so Assumption 2 holds with this value of \$\\beta\$ and \$Q^2=C\_\\beta\$. A classical example is the Sobolev RKHS \$\\mathcal{H}^r([0,1]^d)\$ with smoothness parameter \$r>d/2\$, for which \$\\mu\_k \\asymp k^{-2r/d}\$ and Assumption 2 holds with \$\\beta = 2r/d\$.
>
> - **Gaussian RKHSs**: If \$K\$ is a Gaussian kernel and the input domain is compact, then the eigenvalues decay exponentially fast, which is stronger than any polynomial decay. Consequently, Assumption 2 holds for any $\beta \geq 1$.
>
> **In the revised version, we have added these examples and the explanation in Appendix C.5**.
>
> *Weakness 2: Could you be more careful on the assumption 3? It is ambiguous for non-expert*.
>
> **Answer**: We thank the reviewer for pointing out that Assumption 3 is not sufficiently clear for non-expert readers. We agree and will clarify both the notation and the intuition. Concretely, Assumption 3 is an **embedding condition** on the interpolation space \$[{\\cal H}]\_{\\rho}^{\\alpha}\$. In the revised version, we have restated it as follows:
>
> Assume that there exists an order $\alpha \in [1/\beta,1]$ and a constant $A>0$, such that such that the canonical inclusion map
> $$
> i^\alpha: [\mathcal{H}]^\alpha_\rho \rightarrow L^\infty_\rho (\mathcal{X}), \quad  i^\alpha(f)=f,
> $$
> is bounded, i.e.
> $$
> \\|f\\|_{\\infty} \\leq A \\|f\\|\_{[\mathcal{H}]^\alpha\_\rho},  \quad \forall f \in [{\cal H}]\_{\rho}^{\alpha}.
> $$
> Equivalently, $[\cal H]\_\rho^\alpha$ is continuously embedded into $L\_\rho^\infty(\cal X)$ and the operator norm of the embedding satisfies $\\|i^\alpha\\|\le A$.
>
> The operator $i^\alpha : [\cal H]\_\rho^\alpha \to L\_\rho^\infty(\cal X)$ in Assumption 3 is simply the identity map $i^\alpha(f)=f$. The boundedness condition means that the sup-norm of every function $f \in [\cal H]\_\rho^\alpha$  can be controlled by its $[\cal H]\_\rho^\alpha$-norm.  The case $\alpha=1$ corresponds to the embedding of the RKHS itself, which holds in our setting because the kernel $K$ is bounded, while the case $\alpha=1/\beta$ corresponds to the **benign RKHSs** discussed in Appendix C.3, for which we obtain optimal rates for the mis-specified case.
>
> **In the revised version, we have updated the statement of Assumption 3 in the main text and added the explanation right after Assumption 3**.
>
> *[1] Zheng-Chu Guo, Shao-Bo Lin, and Ding-Xuan Zhou. Learning theory of distributed spectral
> algorithms*.
>
> *[2] Junhong Lin and Volkan Cevher. Convergences of regularized algorithms and stochastic gradient
> methods with random projections*.

---

### Author Response · Authors · 2025-11-18
**To all Reviewers**

We sincerely thank all the reviewers for their insightful feedback and diligent evaluation of our paper. Their valuable input has significantly enriched the quality and clarity of our work. We have carefully considered all questions, concerns, and comments raised by the reviewers and provided detailed responses to each review separately. Our responses have been meticulously integrated into the revised manuscript, with particular attention given to the following key aspects:

(1) We have added a more detailed description and intuitive example to Assumption 2 and 3 (*Reviewer DTFG*), have modified the form of Assumption 4 and added an additional Assumption 5 (*Reviewer i6nj*), have relaxed the compactness and Mercer assumptions (*Reviewers 3MZ4 i6nj*).

(2) We have highlighted the contributions of this paper from the aspects of generalization of conditions (*Reviewer TgQw*), proof techniques (*Reviewers 3MZ4 i6nj*) and comparisons with the related work (*Reviewers TgQw 3MZ4 i6nj*);

(3) We have removed the incorrect claim on the saturation effect and clarified that this does not affect our main results and contributions (*Reviewers 3MZ4 i6nj*), have corrected the description on the lower bound and examples of benign RKHSs (*Reviewer i6nj*);

(4) We have expanded the theoretical and empirical discussion of the projection dimension
$m$ and the choice of the regularization parameter (Reviewers TgQw, 3MZ4);

(5) We have added a detailed proof of the general source condition and the key operator inequality, and improved the paper by correcting typos, clarifying notation, and outlining several promising directions for future work (Reviewer i6nj).

**Revised passages are marked in red, and the important parts we wish to emphasize are highlighted in blue in the updated manuscript.**

Once again, we extend our heartfelt gratitude for your time, expertise, and contribution to our work.

---

### Note · Authors · 2025-11-19

I have read and agree with the venue's withdrawal policy on behalf of myself and my co-authors.